# Provable Rich Observation Reinforcement Learning with Combinatorial Latent States

**Dipendra Misra**[*]
Microsoft Research

**Qinghua Liu**
Princeton University

**Chi Jin**
Princeton University

**John Langford**
Microsoft Research

## Abstract

We propose a novel setting for reinforcement learning that combines two common real-world difficulties: presence of observations (such as camera images) and factored states (such as location of objects). In our setting, the agent receives observations generated stochastically from a *latent* factored state. These observations are *rich enough* to enable decoding of the latent state and remove partial observability concerns. Since the latent state is combinatorial, the size of state space is exponential in the number of latent factors. We create a learning algorithm `FactoRL` (Fact-o-Rel) for this setting which uses noise-contrastive learning to identify latent structures in emission processes and discover a factorized state space. We derive polynomial sample complexity guarantees for `FactoRL` which polynomially depend upon the number factors, and very weakly depend on the size of the observation space. We also provide a guarantee of polynomial time complexity when given access to an efficient planning algorithm.

## 1 Introduction

Most reinforcement learning (RL) algorithms scale polynomially with the size of the state space, which is inadequate for many real world applications. Consider for example a simple navigation task in a room with furniture where the set of furniture pieces and their locations change from episode to episode. If we crudely approximate the room as a $10 \times 10$ grid and consider each element in the grid to contain a single bit of information about the presence of furniture, then we end up with a state space of size $2^{100}$, as each element of the grid can be filled independent of others. This is intractable for RL algorithms that depend polynomially on the size of state space.

The notion of *factorization* allows tractable solutions to be developed. For the above example, the room can be considered a state with 100 factors, where the next value of each factor is dependent on just a few other *parent factors* and the action taken by the agent. Learning in factored Markov Decision Processes (MDP) has been studied extensively (Kearns & Koller, 1999; Guestrin et al., 2003; Osband & Van Roy, 2014) with tractable solutions scaling linearly in the number of factors and exponentially in the number of parent factors whenever planning can be done efficiently.

However, factorization alone is inadequate since the agent may not have access to the underlying factored state space, instead only receiving a rich-observation of the world. In our room example, the agent may have access to an image of the room taken from a megapixel camera instead of the grid representation. Naively, treating each pixel of the image as a factor suggests there are over a million factors and a prohibitively large number of parent factors for each pixel. Counterintuitively, thinking of the observation as the state in this way leads to the conclusion that problems become *harder* as the camera resolution increases or other sensors are added. It is entirely possible, that these pixels (or more generally, observation *atoms*) are generated by a small number of latent factors with a small number of parent factors. This motivates us to ask: *can we achieve PAC RL guarantees that depend polynomially on the number of latent factors and very weakly (e.g., logarithmically) on the size of observation space?* Recent work has addressed this for a rich-observation setting with a non-factored latent state space when certain supervised learning problems are tractable (Du et al., 2019; Misra et al., 2020; Agarwal et al., 2020). However, addressing the rich-observation setting with a latent factored state space has remained elusive. Specifically, ignoring the factored structure in the latent space or treating observation atoms as factors yields intractable solutions.

---

[*]Correspondence at: `dimisra@microsoft.com`

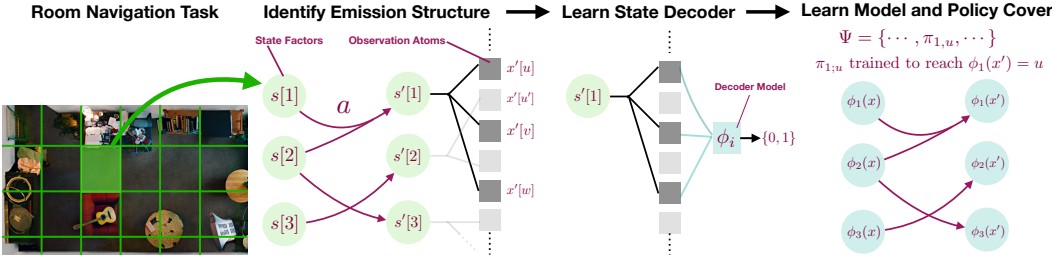

Figure 1: **Left:** A room navigation tasks as a Factored Block MDP setting showing atoms and factors. **Center and Right:** Shows the different stages executed by the FactoRL algorithm. We do not show the observation $x$ emitted by $s$ for brevity. In practice a factor would emit many more atoms.

**Contributions.** We combine two threads of research on rich-observation RL and factored MDP by proposing a new problem setup called *Factored Block MDP* (Section 2). In this setup, observations are emitted by latent states that obey the dynamics of a factored MDP. We assume observations to be composed of atoms (which can be pixels for an image) that are emitted by the latent factors. A single factor can emit a large number of atoms but no two factors can control the same atom. Following existing rich-observation RL literature, we assume observations are *rich enough* to decode the current latent state. We introduce an algorithm FactoRL that achieves the desired guarantees for a large class of Factored Block MDPs under certain computational and realizability assumptions (Section 4). The main challenge that FactoRL handles is to map atoms to the *parent factor* that emits them. We achieve this by reducing the identification problem to solving a set of independence test problems with distributions satisfying certain properties. We perform independence tests in a domain-agnostic setting using noise-contrastive learning (Section 3). Once we have mapped atoms to their parent factors, FactoRL then decodes the factors, estimates the model, recovers the latent structure in the transition dynamics, and learns a set of exploration policies. Figure 1 shows the different steps of FactoRL. This provides us with enough tools to visualize the latent dynamics, and plan for any given reward function. Due to the space limit, we defer the discussion of related work to Appendix B.

*To the best of our knowledge, our work represents the first provable solution to rich-observation RL with a combinatorially large latent state space.*

## 2 THE FACTORED BLOCK MDP SETTING

There are many possible ways to add rich observations to a factored MDP resulting in inapplicability or intractability. Our goal here is to define a problem setting that is tractable to solve and covers potential real-world problems. We start with the definition of Factored MDP (Kearns & Koller, 1999), but first review some useful notation that we will be using:

**Notations:** For any $n \in \mathbb{N}$, we use $[n]$ to denote the set $\{1, 2, \cdots, n\}$. For any ordered set (or a vector) $\mathcal{U}$ of size $n$, and an ordered index set $\mathcal{I} \subseteq [n]$ and length $k$, we use the notation $\mathcal{U}[\mathcal{I}]$ to denote the ordered set $(\mathcal{U}[\mathcal{I}[1]], \mathcal{U}[\mathcal{I}[2]], \cdots, \mathcal{U}[\mathcal{I}[k]])$.

**Definition 1.** *A **Factored MDP** $(\mathcal{S}, \mathcal{A}, T, R, H)$ consists of a d-dimensional discrete state space $\mathcal{S} \subseteq \{0,1\}^d$, a finite action space $\mathcal{A}$, an unknown transition function $T : \mathcal{S} \times \mathcal{A} \to \Delta(\mathcal{S})$, an unknown reward function $R : \mathcal{S} \times \mathcal{A} \to [0,1]$ and a time horizon $H$. Each state $s \in \mathcal{S}$ consists of d factors with the $i^{th}$ factor denoted as $s[i]$. The transition function satisfies $T(s' \mid s, a) = \prod_{i=1}^{d} T_i(s'[i] \mid s[pt(i)], a)$ for every $s, s' \in \mathcal{S}$ and $a \in \mathcal{A}$, where $T_i : \{0,1\}^{|pt(i)|} \times \mathcal{A} \to \Delta(\{0,1\})$ defines a factored transition distribution and a parent function $pt : [d] \to 2^{[d]}$ defines the set of parent factors that can influence a factor at the next timestep.*

We assume a deterministic start state. We also assume, without loss of generality, that each state and observation is reachable at exactly one time step. This can be easily accomplished by concatenating the time step information to state and observations. This allows us to write the state space as $\mathcal{S} = (\mathcal{S}_1, \mathcal{S}_2, \cdots, \mathcal{S}_H)$ where $\mathcal{S}_h$ is the set of states reachable at time step $h$.

A natural question to ask here is why we assume factored transition. In tabular MDPs, the lower bound for sample complexity scales linearly w.r.t. the size of the state set (Kakade, 2003). If we do not assume a factorized transition function then we can encode an arbitrary MDP with a state space of size $2^d$, which would yield a lower bound of $\Omega(2^d)$ rendering the setting intractable. Instead, we will prove sample complexity guarantees for FactoRL that scales in number of factors as $d^{\mathcal{O}(\kappa)}$ where $\kappa := \max_{i \in [d]} |pt(i)|$ is the size of the largest parent factor set. The dependence of $\kappa$ in the exponent is unavoidable as we have to find the parent factors from all possible $\binom{d}{\kappa}$ combinations, as well as learn the model for all possible values of the parent factor. However, for real-world problems we expect $\kappa$ to be a small constant such as 2. This yields significant improvement, for example, if $\kappa = 2$ and $d = 100$ then $d^\kappa = 100$ while $2^d \approx 10^{30}$.

Based on the definition of Factored MDP, we define the main problem setup of this paper, called Factored Block MDP, where the agent does not observe the state but instead receives an observation containing enough information to decode the latent state.

**Definition 2.** *A **Factored Block MDP** consists of an observation space $\mathcal{X} = \mathscr{X}^m$ and a latent state space $\mathcal{S} \subseteq \{0,1\}^d$. A single observation $x \in \mathcal{X}$ is made of $m$ atoms with the $k^{th}$ denoted by $x[k] \in \mathscr{X}$. Observations are generated stochastically given a latent state $s \in \mathcal{S}$ according to a factored emission function $q(x \mid s) = \prod_{i=1}^{d} q_i(x[ch(i)] \mid s[i])$ where $q_i : \{0,1\} \to \Delta(\mathscr{X}^{|ch(i)|})$ and $ch : [d] \to 2^{[m]}$ is a child function satisfying $ch(i) \cap ch(j) = \emptyset$ whenever $i \neq j$. The emission function satisfies the disjointness property: for every $i \in [d]$, we have $supp(q_i(\cdot \mid 0)) \cap supp(q_i(\cdot \mid 1)) = \emptyset$.[1] The dynamics of the latent state space follows a Factored MDP $(\mathcal{S}, \mathcal{A}, T, R, H)$, with parent function $pt$ and a deterministic start state.*

The notion of atoms generalizes commonly used abstractions. For example, if the observation is an image then atoms can be individual pixels or superpixels, and if the observation space is a natural language text then atoms can be individual letters or words. We make no assumption about the structure of the atom space $\mathscr{X}$ or its size, which can be infinite. An agent is responsible for mapping each observation $x \in \mathcal{X}$ to individual atoms $(x[1], \cdots, x[m]) \in \mathscr{X}^m$. For the two examples above, this mapping is routinely performed in practice. If observation is a text presented to the agent as a string, then it can use off-the-shelf tokenizer to map it to sequence of tokens (atoms). Similar to states, we assume the set of observations reachable at different time steps is disjoint. Additionally, we also allow the parent $(pt)$ and child function $(ch)$ to change across time steps. We denote these functions at time step $h$ by $pt_h$ and $ch_h$.

The disjointness property was introduced in Du et al. (2019) for Block MDPs—a class of rich-observation non-factorized MDPs. This property removes partial observability concerns and enables tractable learning. We expect this property to hold in real world problems whenever sufficient sensor data is available to decode the state from observation. For example, disjointness holds true for the navigation task with an overhead camera in Figure 1. In this case, the image provides us with enough information to locate all objects in the room, which describes the agent's state.. Disjointness allows us to define a decoder $\phi_i^\star : \mathscr{X}^{|ch(i)|} \to \{0,1\}$ for every factor $i \in [d]$, such that $\phi_i^\star(x[ch(i)]) = s[i]$ if $x[ch(i)] \in supp(q_i(. \mid s[i]))$. We define a shorthand $\phi_i^\star(x) = \phi_i^\star(x[ch(i)])$ whenever $ch$ is clear from the context. Lastly, we define the state decoder $\phi^\star : \mathcal{X} \to \{0,1\}^d$ where $\phi^\star(x)[i] = \phi_i^\star(x)$.

The agent interacts with the environment by taking actions according to a policy $\pi : \mathcal{X} \to \Delta(\mathcal{A})$. These interactions consist of episodes $\{s_1, x_1, a_1, r_1, s_2, x_2, a_2, r_2, \cdots, a_H, s_H\}$ with $s_1 = \vec{0}$, $x_h \sim q(. \mid s_h)$, $r_h = R(x_h, a_h)$, and $s_{h+1} \sim T(. \mid s_h, a_h)$. The agent never observes $\{s_1, \cdots, s_H\}$.

**Technical Assumptions.** We make two assumptions that are specific to the FactoRL algorithm. The first is a margin assumption on the transition dynamics that enables us to identify different values of a factor. This assumption was introduced by Du et al. (2019), and we adapt it to our setting.

**Assumption 1** (Margin Assumption). *For every $h \in \{2, 3, \cdots, H\}$, $i \in [d]$, let $u_i$ be the uniform distribution jointly over actions and all possible reachable values of $s_{h-1}[pt(i)]$. Then we assume: $\|\mathbb{P}_{u_i}(\cdot, \cdot \mid s_h[i] = 1) - \mathbb{P}_{u_i}(\cdot, \cdot \mid s_h[i] = 0)\|_{TV} \geq \sigma$ where $\mathbb{P}_{u_i}(s_{h-1}[pt(i)], a \mid s_h[i])$ is the* backward dynamics *denoting the probability over parent values and last action given $s_h[i]$ and roll-in distribution $u_i$, and $\sigma > 0$ is the margin.*

---

[1]The notation $supp(p)$ denotes the support of the distribution $p$. Formally, $supp(p) = \{z \mid p(z) > 0\}$.

Assumption 1 captures a large set of problems, including all deterministic problems for which the value of $\sigma$ is 1. Assumption 1 helps us identify the different values of a factor but it does not help with mapping atoms to the factors from which they are emitted. In order to identify if two atoms come from the same factor, we make the following additional assumption to measure their dependence.

**Assumption 2** (Atom Dependency Bound). *For any $h \in [H]$, $u, v \in [m]$ and $u \neq v$, if $ch^{-1}(u) = ch^{-1}(v)$, i.e., atoms $x_h[u]$ and $x_h[v]$ have the same factor. Then under any distribution $D \in \Delta(\mathcal{S}_h)$ we have $\|\mathbb{P}_D(x_h[u], x_h[v]) - \mathbb{P}_D(x_h[u])\mathbb{P}_D(x_h[v])\|_{TV} \geq \beta_{min}$.*

Dependence assumption states that atoms emitted from the same factor will be correlated. This is true for many real-world problems. For example, consider a toy grid-based navigation task. Each state factor $s[i]$ represents a cell in the grid which can be empty ($s[i] = 0$) or occupied ($s[i] = 1$). In the latter case, a randomly sampled box from the set {*red box*, *yellow box*, *black box*}, occupies its place. We expect Assumption 2 to hold in this case as pixels emitted from the same factor come from the same object and hence will be correlated. More specifically, if one pixel is red in color, then another pixel from the same cell will also be red as the object occupying the cell is a red box. This assumption does not remove the key challenge in identifying factors. As atoms from different factors can still be *dependent* due to actions and state distributions from previous time steps.

**Model Class.** We use two regressor classes $\mathcal{F}$ and $\mathcal{G}$. The first regressor class $\mathcal{F} : \mathcal{X} \times \mathcal{X} \to [0, 1]$ takes a pair of atoms and outputs a scalar in $[0, 1]$. To define the second class, we first define a decoder class $\Phi : \mathcal{X}^* \to \{0, 1\}$. We allow this class to be defined on any set of atoms. This is motivated by empirical research where commonly used neural network models operate on inputs of arbitrary lengths. For example, the LSTM model can operate on a text of arbitrary length (Sundermeyer et al., 2012). However, this is without loss of generality as we can define a different model class for different numbers of atom. We also define a model class $\mathcal{U} : \mathcal{X} \times \mathcal{A} \times \{0, 1\} \to [0, 1]$. Finally, we define the regressor class $\mathcal{G} : \mathcal{X} \times \mathcal{A} \times \mathcal{X}^* \to [0, 1]$ as $\{(x, a, \check{x}) \mapsto u(x, a, \phi(\check{x})) \mid u \in \mathcal{U}, \phi \in \Phi\}$. We assume $\mathcal{F}$ and $\mathcal{G}$ are finite classes and derive sample complexity guarantees which scale as $\log|\mathcal{F}|$ and $\log|\mathcal{G}|$. However, since we only use uniform convergence arguments extending the guarantees to other statistical complexity measures such as Rademacher complexity is straightforward. Let $\Pi_{\text{all}} : \mathcal{S} \to \mathcal{A}$ denote the set of all non-stationary policies of this form. We then define the class of policies $\Pi : \mathcal{X} \to \mathcal{A}$ by $\{x \mapsto \varphi(\phi^*(x)) \mid \forall \varphi \in \Pi_{\text{all}}\}$, which we use later to define our task. We use $\mathbb{P}_\pi[\mathcal{E}]$ to denote probability of an event $\mathcal{E}$ under the distribution over episodes induced by policy $\pi$.

**Computational Oracle.** We assume access to two regression oracles REG for model classes $\mathcal{F}$ and $\mathcal{G}$. Let $\mathcal{D}_1$ be a dataset of triplets $(x[u], x[v], y)$ where $u, v$ denote two different atoms and $y \in \{0, 1\}$. Similarly, let $\mathcal{D}_2$ be a dataset of quads $(x, a, x', y)$ where $x \in \mathcal{X}$, $a \in \mathcal{A}$, $\check{x} \in \mathcal{X}^*$, and $y \in \{0, 1\}$. Lastly, let $\widehat{\mathbb{E}}_D[\cdot]$ denote the empirical mean over dataset $D$. The two computational oracles compute:

$$\texttt{REG}(\mathcal{D}_1, \mathcal{F}) = \arg\min_{f \in \mathcal{F}} \widehat{\mathbb{E}}_{\mathcal{D}_1}\Big[\big(f(x[u], x[v]) - y\big)^2\Big], \quad \texttt{REG}(\mathcal{D}_2, \mathcal{G}) = \arg\min_{g \in \mathcal{G}_N} \widehat{\mathbb{E}}_{\mathcal{D}_2}\Big[\big(g(x, a, \check{x}) - y\big)^2\Big].$$

We also assume access to a $\Delta_{\text{pl}}$-optimal planning oracle $\texttt{planner}$. Let $\widehat{\mathcal{S}} = (\widehat{\mathcal{S}}_1, \cdots, \widehat{\mathcal{S}}_h)$ be a learned state space and $\widehat{T} = (\widehat{T}_1, \cdots, \widehat{T}_H)$ with $\widehat{T}_h : \widehat{\mathcal{S}}_{h-1} \times \mathcal{A} \to \Delta(\widehat{\mathcal{S}}_h)$ be the learned dynamics, and $\widehat{R} : \widehat{\mathcal{S}} \times \mathcal{A} \to [0, 1]$ be a given reward function. Let $\varphi : \widehat{\mathcal{S}} \to \mathcal{A}$ be a policy and $V(\varphi; \widehat{T}, \widehat{R})$ be the policy value. Then for any $\Delta_{\text{pl}} > 0$ the output of planner $\hat{\varphi} = \texttt{planner}(\widehat{T}, \widehat{R}, \Delta_{\text{pl}})$ satisfies $V(\hat{\varphi}; \widehat{T}, \widehat{R}) \geq \sup_\varphi V(\varphi; \widehat{T}, \widehat{R}) - \Delta_{\text{pl}}$, where supremum is taken over policies of type $\widehat{\mathcal{S}} \to \mathcal{A}$.

**Task Definition.** We focus on a reward-free setting with the goal of learning a state decoder and estimating the latent dynamics $T$. Since the state space is exponentially large, we cannot visit every state. However, the factorization property allows us to estimate the model by reaching factor values. In fact, we show that controlling the value of at most $2\kappa$ factors is sufficient for learning the model. Let $\mathscr{C}_{\leq k}(\mathcal{U})$ denote the space of all sets containing at most $k$ different elements selected from the set $\mathcal{U}$ including $\emptyset$. We define the reachability probability $\eta_h(\mathcal{K}, \mathcal{Z})$ for a given $h \in [H]$, $\mathcal{K} \subseteq [d]$, and $\mathcal{Z} \in \{0, 1\}^{|\mathcal{K}|}$, and the reachability parameter $\eta_{min}$ as:

$$\eta_h(\mathcal{K}, \mathcal{Z}) := \sup_{\pi \in \Pi_{\text{NS}}} \mathbb{P}_\pi(s_h[\mathcal{K}] = \mathcal{Z}), \qquad \eta_{min} := \inf_{h \in [H]} \inf_{s \in \mathcal{S}_h} \inf_{\mathcal{K} \in \mathscr{C}_{\leq 2\kappa}([d])} \eta_h(\mathcal{K}, s[\mathcal{K}]).$$

Our sample complexity scales polynomially with $\eta_{min}^{-1}$. Note that we only require that if $s_h[\mathcal{K}] = \mathcal{Z}$ is reachable, then it is reachable with at least $\eta_{min}$ probability, i.e., either $\eta_h(\mathcal{K}, \mathcal{Z}) = 0$ or it is at least $\eta_{min}$. These requirements are similar to those made by earlier work for non-factored state

space (Du et al., 2019; Misra et al., 2020). The key difference being that instead of requiring every state to be reachable with $\eta_{min}$ probability, we only require a small set of factor values to be reachable. For reference, if every policy induces a uniform distribution over $\mathcal{S} = \{0, 1\}^d$, then probability of visiting any state is $2^{-d}$ but the probability of two factors taking certain values is only $0.25$. This gives us a more practical value for $\eta_{min}$.

Besides estimating the dynamics and learning a decoder, we also learn an $\alpha$-policy cover to enable exploration of different reachable values of factors. We define this below:

**Definition 3** (Policy Cover). *A set of policies $\Psi$ is an $\alpha$-policy cover of $\mathcal{S}_h$ for any $\alpha > 0$ and $h$ if:*

$$\forall s \in \mathcal{S}_h, \mathcal{K} \in \mathscr{C}_{\leq 2\kappa}([d]), \qquad \sup_{\pi \in \Psi} \mathbb{P}_\pi(s_h[\mathcal{K}] = s[\mathcal{K}]) \geq \alpha \eta_h(\mathcal{K}, s[\mathcal{K}]).$$

## 3 DISCOVERING EMISSION STRUCTURE WITH CONTRASTIVE LEARNING

Directly applying the prior work (Du et al., 2019; Misra et al., 2020) to decode a factored state from observation results in failure, as the learned factored state need not obey the transition factorization. Instead, the key high-level idea of our approach is to first learn the latent emission structure $ch$, and then use it to decode each factor individually. We next discuss our approach for learning $ch$.

**Reducing Identification of Latent Emission Structure to Independence Tests.** Assume we are able to perfectly decode the latent state and estimate the transition model till time step $h - 1$. Our goal is to infer the latent emission structure $ch_h$ at time step $h$, which is equivalent to: given an arbitrary pair of atoms $u$ and $v$, determine if they are emitted from the same factor or not. This is challenging since we cannot observe or control the latent state factors at time step $h$.

Let $i = ch^{-1}(u)$ and $j = ch^{-1}(v)$ be the factors that emit $x[u]$ and $x[v]$. If $i = j$, then Assumption 2 implies that these atoms are dependent on each other for any roll-in distribution $D \in \Delta(\mathcal{S}_{h-1} \times \mathcal{A})$ over previous state and action. However, if $i \neq j$ then deterministically setting the previous action and values of the parent factors $pt(i)$ or $pt(j)$, makes $x[u]$ and $x[v]$ independent. For the example in Figure 1, fixing the value of $s[1], s[2]$ and $a$ would make $x[u]$ and $x[u']$ independent of each other.

This observation motivates us to reduce this identification problem to performing independence tests with different roll-in distributions $D \in \Delta(\mathcal{S}_{h-1} \times \mathcal{A})$. Naively, we can iterate over all subsets $\mathcal{K} \in \mathscr{C}_{\leq 2\kappa}([d])$ where for each $\mathcal{K}$ we create a roll-in distribution such that the values of $s_{h-1}[\mathcal{K}]$ and the action $a_{h-1}$ are fixed, and then perform independence test under this distribution. If two atoms are independent then there must exist a $\mathcal{K}$ that makes them independent. Otherwise, they should always be dependent by Assumption 2.

However, there are two problems with this approach. Firstly, we do not have access to the latent states but only a decoder at time step $h - 1$. Further, it may not even be possible to find a policy that can set the values of factors deterministically. We later show that our algorithm `FactoRL` can learn a decoder that induces a bijection between learned factors and values, and the real factors and values. Therefore, maximizing the probability of $\hat{\mathcal{E}}_{\mathcal{K};\mathcal{Z}} = \{\widehat{\phi}_{h-1}(x_{h-1})[\mathcal{K}] = \mathcal{Z}\}$ for a set of learned factors $\mathcal{K}$ and their values $\mathcal{Z}$, implicitly maximizes the probability of $\mathcal{E}_{\mathcal{K}';\mathcal{Z}'} = \{s_{h-1}[\mathcal{K}'] = \mathcal{Z}'\}$ for corresponding real factors $\mathcal{K}'$ and their values $\mathcal{Z}'$. Since the event $\hat{\mathcal{E}}_{\mathcal{K};\mathcal{Z}}$ is observable we can use rejection sampling to increase its probability sufficiently close to 1 which makes the probability of $\mathcal{E}_{\mathcal{K}';\mathcal{Z}'}$ close to 1.

The second problem is to perform independence tests in a domain agnostic setting. Directly estimating mutual information $\mathcal{I}(x[u]; x[v])$ can be challenging. Instead, we propose an *oraclized* independence test that reduces the problem to binary classification using noise-contrastive learning.

**Oraclized Independent Test.** Here, we briefly sketch the main idea of our independence test scheme and defer the details to Appendix C. We comment that the high-level idea of our independence testing subroutine is similar to Sen et al. (2017). Suppose we want to test if two random variables $Y$ and $Z$ are independent. Firstly, we construct a dataset in the following way: sample a Bernoulli random variable $w \sim \text{Bern}(1/2)$, and two pairs of independent realizations $(y^{(1)}, z^{(1)})$ and $(y^{(2)}, z^{(2)})$; if $w = 1$, add $(y^{(1)}, z^{(1)}, w)$ to the dataset, and $(y^{(1)}, z^{(2)}, w)$ otherwise. We repeat the sampling procedure $n$ times and obtain a dataset $\{(y_i, z_i, w_i)\}_{i=1}^n$. Then we can fit a classifier that predicts the value of $w_i$ using $(y_i, z_i)$. If $Y$ and $Z$ are independent, then $(y_i, z_i)$ will provide no information about $w_i$ and thus no classifier can do better than random guess. However, if $Y$ and $Z$ are dependent, then

the Bayes optimal classifier would perform strictly better than random guess. As a result, by looking at the training loss of the learned classifier, we can determine whether $Y$ and $Z$ are dependent or not.

## 4    FactoRL: Reinforcement Learning in Factored Block MDPs

In this section, we present the main algorithm FactoRL (Algorithm 1). It takes as input the model classes $\mathcal{F}, \mathcal{G}$, failure probability $\delta > 0$, and five hyperparameters $\sigma, \eta_{min}, \beta_{\min} \in (0, 1)$ and $d, \kappa \in \mathbb{N}$.[2] We use these hyperparamters to define three sample sizes $n_{\text{ind}}, n_{\text{abs}}, n_{\text{est}}$ and rejection sample frequency $k$. For brevity, we defer the exact values of these constants to Appendix D.7. FactoRL returns a learned decoder $\widehat{\phi}_h : \mathcal{X} \to \{0, 1\}^{d_h}$ for some $d_h \in [m]$, an estimated transition model $\widehat{T}_h$, learned parent $\widehat{pt}_h$ and child functions $\widehat{ch}_h$, and a $1/2$-policy cover $\Psi_h$ of $\mathcal{S}_h$ for every time step $h \in \{2, 3, \cdots, H\}$. We use $\hat{s}_h$ to denote the learned state at time step $h$. Formally, $\hat{s}_h = (\widehat{\phi}_{h1}(x_h), \cdots, \widehat{\phi}_{hd_h}(x_h))$. In the analysis of FactoRL, we show that $d_{h-1} = d$, and $\widehat{ch}_h$ is equivalent to $ch_h$ up to permutation with high probability. Further, we show that $\widehat{\phi}_h$ and $\widehat{ch}_h$ together learn a bijection between learned factors and their values and real factors and their values.

FactoRL operates inductively over the time steps (Algorithm 1, line 2-8). In the $h^{th}$ iteration, the algorithm performs four stages of learning: identifying the latent emission structure, decoding the factors, estimating the model, and learning a policy cover. We describe these below.

---

**Algorithm 1** FactoRL$(\mathcal{F}, \mathcal{G}, \delta, \sigma, \eta_{min}, \beta_{\min}, d, \kappa)$. RL in Factored Block MDPs.

1: Initialize $\Psi_h = \emptyset$ for every $h \in [H]$ and $\widehat{\phi}_1 = \mathcal{X} \to \{0\}$. Set global constants $n_{\text{ind}}, n_{\text{abs}}, n_{\text{est}}, k$.
2: **for** $h \in \{2, 3, \cdots, H\}$ **do**
3:     $\widehat{ch}_h = \texttt{FactorizeEmission}(\Psi_{h-1}, \widehat{\phi}_{h-1}, \mathcal{F})$ // stage 1: discover latent emission structure
4:     $\widehat{\phi}_h = \texttt{LearnDecoder}(\mathcal{G}, \Psi_{h-1}, \widehat{ch}_h)$                         // stage 2: learn a decoder for factors
5:     $\widehat{T}_h, \widehat{pt}_h = \texttt{EstModel}(\Psi_{h-1}, \widehat{\phi}_{h-1}, \widehat{\phi}_h)$         // stage 3: find latent $pt_h$ and estimate model
6:     **for** $\mathcal{I} \in \mathscr{C}_{\leq 2\kappa}([d]), \mathcal{Z} \in \{0, 1\}^{|\mathcal{I}|}$ **do**
7:         $\hat{\varphi}_{h\mathcal{IZ}} = \texttt{planner}(\widehat{T}, R_{h\mathcal{IZ}}, \Delta_{\text{pl}})$ where $R_{h\mathcal{IZ}} := \mathbf{1}\{\hat{s}_h[\mathcal{I}] = \mathcal{Z}\}$   // stage 4: planning
8:         **If** $V(\hat{\varphi}_{h\mathcal{IZ}}; \widehat{T}, R_{h\mathcal{IZ}}) \geq 3\eta_{min}/4$ **then** $\Psi_h \leftarrow \Psi_h \cup \{\hat{\varphi}_{h\mathcal{IZ}} \circ \widehat{\phi}_h\}$
    **return** $\{\widehat{ch}_h, \widehat{\phi}_h, \widehat{T}_h, \widehat{pt}_h, \Psi_h\}_{h=2}^H$

---

**Identifying Latent Emission Process.** The FactorizeEmission collects a dataset of observations for every policy in $\Psi_{h-1}$ and action $a \in \mathcal{A}$ (Algorithm 2, line 1-4). Policies in $\Psi_{h-1}$ are of the type $\pi_{\mathcal{I};\mathcal{Z}}$ where $\mathcal{I} \in \mathscr{C}_{\leq 2\kappa}([d_{h-1}])$ and $\mathcal{Z} \in \{0, 1\}^{|\mathcal{I}|}$. We can inductively assume $\pi_{\mathcal{I};\mathcal{Z}}$ to be maximizing the probability of $\mathcal{E}_{\mathcal{I};\mathcal{Z}} = \{\hat{s}_{h-1}[\mathcal{I}] = \mathcal{Z}\}$. If our decoder is accurate enough, then we hope that maximizing the probability of this event in turn maximizes the probability of fixing the values of a set of real factors. However, it is possible that $\mathbb{P}_{\pi_{\mathcal{I};\mathcal{Z}}}(\hat{s}_{h-1}[\mathcal{I}] = \mathcal{Z})$ is only $\mathcal{O}(\eta_{min})$. Therefore, as explained earlier, we use rejection sampling to drive the probability of this event close to 1. Formally, we define a procedure $\texttt{RejectSamp}(\pi_{\mathcal{I};\mathcal{Z}}, \mathcal{E}_{\mathcal{I};\mathcal{Z}}, k)$ which rolls-in at time step $h - 1$ with $\pi_{\mathcal{I};\mathcal{Z}}$ to observe $x_{h-1}$ (line 3). If the event $\mathcal{E}_{\mathcal{I};\mathcal{Z}}$ holds for $x_{h-1}$ then we return $x_{h-1}$, otherwise, we repeat the procedure. If we fail to satisfy the event $k$ times then we return the last sample. We use this to define our main sampling procedure $x_h \sim D_{\mathcal{I},\mathcal{Z},a} := \texttt{RejectSamp}(\pi_{\mathcal{I};\mathcal{Z}}, \mathcal{E}_{\mathcal{I};\mathcal{Z}}, k) \circ a$ which first samples $x_{h-1}$ using the rejection sampling procedure and then takes action $a$ to observe $x_h$. We collect a dataset of observation pairs $(x^{(1)}, x^{(2)})$ sampled independently from $D_{\mathcal{I},\mathcal{Z},a}$.

For every pair of atoms $u, v \in [m]$, we calculate if they are independent under the distribution induced by $D_{\mathcal{I},\mathcal{Z},a}$ using IndTest with dataset $\mathcal{D}_{\mathcal{I},\mathcal{Z},a}$ (line 5-7). We share the dataset across atoms for sample efficiency. If there exists at least one $(\mathcal{I}, \mathcal{Z}, a)$ triple such that we evaluate $x[u], x[v]$ to be independent, then we mark these atoms as coming from different factors. Intuitively, such an $\mathcal{I}$ would contain parent factors of at least $ch_h^{-1}(u)$ or $ch_h^{-1}(v)$. If no such $\mathcal{I}$ exists then we mark these atoms as being emitted from the same factor.

---

[2]Our analysis can use any non-zero lower bound on $\eta_{min}, \beta_{\min}, \sigma$ and an upper bound on $d$ and $\kappa$.

---

**Algorithm 2** $\texttt{FactorizeEmission}(\Psi_{h-1}, \widehat{\phi}_{h-1}, \mathcal{F})$.

---

1: **for** $(\pi_{\mathcal{I};\mathcal{Z}}, a) \in \Psi_{h-1} \times \mathcal{A}$ and $i \in [n_{\text{ind}}]$ **do**

2:      Define $\mathcal{E}_{\mathcal{I};\mathcal{Z}} := \mathbf{1}\{\widehat{\phi}_{h-1}(x_{h-1})[\mathcal{I}] = \mathcal{Z}\}$

3:      Sample $x_h^{(1)}, x_h^{(2)} \sim \texttt{RejectSamp}(\pi_{\mathcal{I};\mathcal{Z}}, \mathcal{E}_{\mathcal{I};\mathcal{Z}}, k) \circ a$      // rejection sampling procedure

4:      $\mathcal{D}_{\mathcal{I};\mathcal{Z};a} \leftarrow \mathcal{D}_{\mathcal{I};\mathcal{Z};a} \cup \{(x_h^{(1)}, x_h^{(2)})\}$      // initialize $\mathcal{D}_{\mathcal{I};\mathcal{Z};a} = \emptyset$

5: **for** $u \in \{1, 2, \cdots, m-1\}$ and $v \in \{u+1, \cdots, m\}$ **do**

6:      Mark $u, v$ as coming from the same factor, i.e., $\widehat{ch}_h^{-1}(u) = \widehat{ch}_h^{-1}(v)$ if $\forall (\mathcal{I}, \mathcal{Z}, a)$

7:      the oraclized independence test finds $x_h[u], x_h[v]$ as dependent using $\mathcal{D}_{\mathcal{I};\mathcal{Z};a}$ and $\mathcal{F}$

     **return** $\widehat{ch}_h$      // label ordering of parents does not matter.

---

**Algorithm 3** $\texttt{LearnDecoder}(\mathcal{G}, \Psi_{h-1}, \widehat{ch}_h)$.      Child function has type $\widehat{ch}_h : [d_h] \to 2^{[m]}$

---

1: **for** $i$ in $[d_h]$, define $\omega = \widehat{ch}_h(i), \mathcal{D} = \emptyset$ **do**

2:      **for** $n_{\text{abs}}$ times **do**      // collect a dataset of real $(y=1)$ and imposter $(y=0)$ transitions

3:          Sample $(x^{(1)}, a^{(1)}, x'^{(1)}), (x^{(2)}, a^{(2)}, x'^{(2)}) \sim \texttt{Unf}(\Psi_{h-1}) \circ \texttt{Unf}(\mathcal{A})$ and $y \sim \texttt{Bern}(\frac{1}{2})$

4:          **If** $y = 1$ **then** $\mathcal{D} \leftarrow \mathcal{D} \cup (x^{(1)}, a^{(1)}, x'^{(1)}[\omega], y)$ **else** $\mathcal{D} \leftarrow \mathcal{D} \cup (x^{(1)}, a^{(1)}, x'^{(2)}[\omega], y)$

5:      $\widehat{u}_i, \widehat{\phi}_i = \texttt{REG}(\mathcal{D}, \mathcal{G})$      // train the decoder using noise-contrastive learning

     **return** $\widehat{\phi} : \mathcal{X} \to \{0, 1\}^{d_h}$ where for any $x \in \mathcal{X}$ and $i \in [d_h]$ we have $\widehat{\phi}(x)[i] = \widehat{\phi}_i(x[\widehat{ch}_h(i)])$.

---

**Decoding Factors.** $\texttt{LearnDecoder}$ partitions the set of atoms into groups based on the learned child function $\widehat{ch}_h$ (Algorithm 3). For the $i^{th}$ group $\omega$, we learn a decoder $\widehat{\phi}_{hi} : \mathcal{X}^\star \to \{0, 1\}$ by adapting the prediction problem of Misra et al. (2020) to Factored Block MDP setting. We define a sampling procedure $(x, a, x') \sim \texttt{Unf}(\Psi_{h-1}) \circ \texttt{Unf}(\mathcal{A})$ where $x$ is observed after roll-in with a uniformly selected policy in $\Psi_{h-1}$ till time step $h-1$, action $a$ is taken uniformly, and $x' \sim T(\cdot \mid x, a)$ (line 3). We collect a dataset $\mathcal{D}$ of real and imposter transitions. A single datapoint in $\mathcal{D}$ is collected by sampling two independent transitions $(x^{(1)}, a^{(1)}, x'^{(1)}), (x^{(2)}, a^{(2)}, x'^{(2)}) \sim \texttt{Unf}(\Psi_{h-1}) \circ \texttt{Unf}(\mathcal{A})$ and a Bernoulli random variable $y \sim \texttt{Bern}(1/2)$. If $y = 1$ then we add the real transition $(x^{(1)}, a^{(1)}, x'^{(1)}[\omega], y)$ to $\mathcal{D}$, otherwise we add the imposter transition $(x^{(1)}, a^{(1)}, x'^{(2)}[\omega], y)$ (line 4). The key difference from Misra et al. (2020) is our use $x'[\omega]$ instead of $x'$ which allows us to decode a specific latent factor. We train a model to predict the probability that a given transition $(x, a, x'[\omega])$ is real by solving a regression task with model class $\mathcal{G}$ (line 5). The bottleneck structure of $\mathcal{G}$ allows us to recover a decoder $\widehat{\phi}_i$ from the learned model. The algorithm also checks for the special case where a factor takes a single value. If it does, then we return the decoder that always outputs 0, otherwise we stick with $\widehat{\phi}_i$. For brevity, we defer the details of this special case to Appendix D.2.2. The decoder for the $h^{th}$ timestep is given by composition of decoders for each group.

---

**Algorithm 4** $\texttt{EstModel}(\Psi_{h-1}, \widehat{\phi}_{h-1}, \widehat{\phi}_h)$.

---

1: Collect dataset $\mathcal{D}$ of $n_{\text{est}}$ triplets $(x, a, x') \sim \texttt{Unf}(\Psi_{h-1}) \circ \texttt{Unf}(\mathcal{A})$

2: **for** $\mathcal{I}, \mathcal{J} \in \mathscr{C}_{\leq \kappa}([d_{h-1}])$ satisfying $\mathcal{I} \cap \mathcal{J} = \emptyset$ **do**

3:      Estimate $\widehat{\mathbb{P}}(\hat{s}_h[k] \mid \hat{s}_{h-1}[\mathcal{I}], \hat{s}_{h-1}[\mathcal{J}], a)$ from $\mathcal{D}$ using $\widehat{\phi}$,    $\forall a \in \mathcal{A}, k \in [d_h]$.

4: For every $k$ define $\widehat{pt}_h(k)$ as solution to following: (where we bind $\hat{s}' = \hat{s}_h$ and $\hat{s} = \hat{s}_{h-1}$)

$$\underset{\mathcal{I}}{\text{argmin}} \ \underset{u, \mathcal{J}_1, \mathcal{J}_2, w_1, w_2, a}{\max} \left| \widehat{\mathbb{P}}(\hat{s}'[k] \mid \hat{s}[\mathcal{I}] = u, \hat{s}[\mathcal{J}_1] = w_1, a) - \widehat{\mathbb{P}}(\hat{s}'[k] \mid \hat{s}[\mathcal{I}] = u, \hat{s}[\mathcal{J}_2] = w_2, a) \right|_{\text{TV}}$$

5: Define $\widehat{T}_h(\hat{s}' \mid \hat{s}, a) = \prod_k \widehat{\mathbb{P}}(\hat{s}[k] \mid \hat{s}[\widehat{pt}_h(k)], a)$ and **return** $\widehat{T}_h, \widehat{pt}_h$

---

**Estimating the Model.** $\texttt{EstModel}$ routine first collects a dataset $\mathcal{D}$ of $n_{\text{est}}$ independent transitions $(x, a, x') \sim \texttt{Unf}(\Psi_{h-1}) \circ \texttt{Unf}(\mathcal{A})$ (Algorithm 4, line 1). We iterate over two disjoint sets of factors $\mathcal{I}, \mathcal{J}$ of size at most $\kappa$. We can view $\mathcal{I}$ as the control set and $\mathcal{J}$ as the variable set. For every

learned factor $k \in [d_h]$, factor set $\mathcal{I}, \mathcal{J}$ and action $a \in \mathcal{A}$, we estimate the model $\widehat{\mathbb{P}}(\hat{s}_h[k] \mid \hat{s}_{h-1}[\mathcal{I}], \hat{s}_{h-1}[\mathcal{J}], a)$ using count based statistics on dataset $\mathcal{D}$ (line 3).

Consider the case where the $\widehat{ch}_t = ch_t$ for every $t \in [h]$ and where we ignore the label permutation for brevity. If $\mathcal{I}$ contains the parent factors $pt(k)$, then we expect the value of $\widehat{\mathbb{P}}(\hat{s}'[k] \mid \hat{s}[\mathcal{I}], \hat{s}[\mathcal{J}], a) \approx T_k(\hat{s}'[k] \mid \hat{s}[pt(k)], a)$ to not change significantly on varying either the set $\mathcal{J}$ or its values. This motivates us to define the learned parent set as the $\mathcal{I}$ which achieves the minimum value of this *gap* (line 4). When computing the gap, we take $\max$ only over those values of $\hat{s}[\mathcal{I}]$ and $\hat{s}[\mathcal{J}]$ which can be reached jointly using a policy in $\Psi_{h-1}$. This is important since we can only reliably estimate the model for reachable factor values. The learned parent function $\widehat{pt}_h$ need not be identical to $pt_h$ even upto relabeling. However, finding the exact parent factor is not necessary for learning an accurate model, and may even be impossible. For example, two factors may always take the same value making it impossible to distinguish between them. We use the learned parent function $\widehat{pt}_h$ to define $\widehat{T}_h$ similar to the structure of $T$ (line 5).

**Learning a Policy Cover.** We plan in the latent space using the estimated model $\{\widehat{T}_t\}_{t=1}^h$, to find a policy cover for time step $h$. Formally, for every $\mathcal{I} \in \mathscr{C}_{\leq 2\kappa}([d_h])$ and $\mathcal{Z} \in \{0,1\}^{|\mathcal{I}|}$, we find a policy $\hat{\varphi}_{h\mathcal{I}\mathcal{Z}}$ to reach $\{\hat{s}_h[\mathcal{I}] = \mathcal{Z}\}$ using the planner (Algorithm 1, line 7). This policy acts on the learned state space and is easily lifted to act on observations by composition with the learned decoder. We add every policy that achieves a return of at least $\mathcal{O}(\eta_{min})$ to $\Psi_h$ (line 8).

## 5 THEORETICAL ANALYSIS AND DISCUSSION

In this section, we present theoretical guarantees for FactoRL. For technical reasons, we make the following realizability assumption on the function classes $\mathcal{F}$ and $\mathcal{G}$.

**Assumption 3** (Realizability). *For any $h \in [H]$, $i \in [d]$ and distribution $\rho \in \Delta(\{0,1\})$, there exists $g_{ih\rho} \in \mathcal{G}$, such that for all $\forall (x,a,x') \in \mathcal{X}_{h-1} \times \mathcal{A} \times \mathcal{X}_h$ and $\check{x} = x'[ch_h(i)]$ we have:*

$$g_{ih\rho}(x, a, \check{x}) = \frac{T_i(\phi_i^\star(\check{x}) \mid \phi^\star(x), a)}{T_i(\phi_i^\star(\check{x}) \mid \phi^\star(x), a) + \rho(\phi_i^\star(\check{x}))}.$$

*For any $h \in [H]$, $u, v \in [m]$ with $u \neq v$, and any $D \in \Delta(S_h)$, there exists $f_{uvD} \in \mathcal{F}$ satisfying:*

$$\forall s \in supp(D), x \in supp(q(\cdot \mid s)), \qquad f_{uvD}(x[u], x[v]) = \frac{D(x[u], x[v])}{D(x[u], x[v]) + D(x[u])D(x[v])}.$$

Assumption 3 requires the function classes to be expressive enough to represent optimal solutions for our regression tasks. Realizability assumptions are common in literature and are in practice satisfied by using deep neural networks (Sen et al., 2017; Misra et al., 2020).

**Theorem 1** (Main Theorem). *For any $\delta > 0$, FactoRL returns a transition function $\widehat{T}_h$, a parent function $\widehat{ch}_h$, a decoder $\widehat{\phi}_h$, and a set of policies $\Psi_h$ for every $h \in \{2, 3, \cdots, H\}$, that with probability at least $1 - \delta$ satisfies: (i) $\widehat{ch}_h$ is equal to $ch_h$ upto permutation, (ii) $\Psi_h$ is a $1/2$ policy cover of $\mathcal{S}_h$, (iii) For every $h \in [H]$, there exists a permutation mapping $\theta_h : \{0,1\}^d \to \{0,1\}^d$ such that for every $s \in \mathcal{S}_{h-1}, a \in \mathcal{A}, s' \in \mathcal{S}_h$ and $x' \in \mathcal{X}_h$ we have:*

$$\mathbb{P}(\widehat{\phi}_h(x') = \theta_h^{-1}(s') \mid s') \geq 1 - \mathcal{O}\left(\frac{\eta_{min}^2}{\kappa H}\right), \quad \left\| T(s' \mid s, a) - \widehat{T}_h(\theta_h^{-1}(s') \mid \theta_{h-1}^{-1}(s), a) \right\|_{TV} \leq \frac{\eta_{min}}{8H},$$

*and the sample complexity is $\texttt{poly}(d^{16\kappa}, |\mathcal{A}|, H, \frac{1}{\eta_{min}}, \frac{1}{\delta}, \frac{1}{\beta_{min}}, \frac{1}{\sigma}, \ln m, \ln |\mathcal{F}|, \ln |\mathcal{G}|)$.*

**Discussion.** The proof and the detailed analysis of Theorem 1 has been deferred to Appendix C-D. Our guarantees show that FactoRL is able to discover the latent emission structure, learn a decoder, estimate the model, and learn a policy cover for every timestep. We set the hyperparmeters in order to learn a $1/2$-policy cover, however, they can also be set to achieve a desired accuracy for the decoder or the transition model. This will give a polynomial sample complexity that depends on this desired accuracy. It is straightforward to plan a near-optimal policy for a given reward function in our learned latent space, using the estimated model and the learned decoder. This incurs zero sample cost apart from the samples needed to learn the reward function.

Our results show that we depend polynomially on the number of factors and only logarithmic in the number of atoms. This appeals to real-world problem where $d$ and $m$ can be quite large. We also depend logarithmically on the size of function classes. This allows us to use exponentially large function classes, further, as stated before, our results can also be easily extended to Rademacher complexity. Our algorithm only makes a polynomial number of calls to computational oracles. Hence, if these oracles can be implemented efficiently then our algorithm has a polynomial computational complexity. The squared loss oracles are routinely used in practice, but planning in a fully-observed factored MDP is EXPTIME-complete (see Theorem 2.24 of Mausam (2012)). However, various approximation strategies based on linear programming and dynamic programming have been employed succesfully (Guestrin et al., 2003). These assumptions provide a black-box mechanism to leverage such efforts. Note that all computational oracles incur no additional sample cost and can be simply implemented by enumeration over the search space.

**Comparison with Block MDP Algorithms.**   Our work is closely related to algorithms for Block MDPs, which can be viewed as a non-factorized version of our setting. Du et al. (2019) proposed a model-based approach for Block MDPs. They learn a decoder for a given time step by training a classifier to predict the decoded state and action at the last time step. In our case, this results in a classification problem over exponentially many classes which can be practically undesirable. In contrast, Misra et al. (2020) proposed a model-free approach that learns a decoder by training a classifier to distinguish between real and imposter transitions. As optimal policies for factored MDPs do not factorize, therefore, a model-free approach is unlikely to succeed (Sun et al., 2019). Feng et al. (2020) proposed another approach for solving Block MDPs. They assume access to a purely unsupervised learning oracle, that can learn an accurate decoder using a set of observations. This oracle assumption is significantly stronger than those made in Du et al. (2019) and Misra et al. (2020), and reduces the challenge of learning the decoder. Crucially, these three approaches have a sample complexity guarantee which depends polynomially on the size of latent state space. This yields an exponential dependence on $d$ when applied to our setting. It is unclear if these approaches can be extended to give polynomial dependence on $d$. For general discussion of related work see Appendix B.

**Proof of Concept Experiments.**   We empirically evaluate `FactoRL` to support our theoretical results, and to provide implementation details. We consider a problem with $d$ factors each emitting 2 atoms. We generate atoms for factor $s[i]$, by first defining a vector $z_i = [1, 0]$ if $s[i] = 0$, and $z_i = [0, 1]$ otherwise. We then sample a scalar Gaussian noise $g_i$ with 0 mean and 0.1 standard deviation, and add it to both component of $z_i$. Atoms emitted from each factor are concatenated to generate a vector $z \in \mathbb{R}^{2d}$. The observation $x$ is generated by applying a fixed time-dependent permutation to $z$ to shuffle atoms from different factors. This ensures that an algorithm cannot figure out the children function by relying on the order in which atoms are presented. We consider an action space $\mathcal{A} = \{a_1, a_2, \cdots, a_d\}$ and non-stationary dynamics. For each time step $t \in [H]$, we define $\sigma_t$ as a fixed permutation of $\{1, 2, \cdots, d\}$. Dynamics at time step $t$ are given by: $T_t(s_{t+1} \mid s_t, a) = \prod_{i=1}^{d} T_{ti}(s_{t+1}[i] \mid s_t[i], a)$, where $T_{ti}(s_{t+1}[i] = s_t[i] \mid s_t[i], a) = 1$ for all $a \neq a_{\sigma_t(i)}$, and $T_{ti}(s_{t+1}[i] = 1 - s_t[i] \mid s_t[i], a_{\sigma_t(i)}) = 1$. We evaluate on the setting $d = 10$ and $H = 10$.

We implement model classes $\mathcal{F}$ and $\mathcal{G}$ using feed-forward neural networks. Specifically, for $\mathcal{G}$ we apply the Gumbel-softmax trick to model the bottleneck following Misra et al. (2020). We train the models using cross-entropy loss instead of squared loss that we use for theoretical analysis.[3] For the independence test task, we declare two atoms to be independent, if the best log-loss on the validation set is greater than $c$. We train the model using Adam optimization and perform model selection using a held-out set. We defer the full model and training details to Appendix F.

For each time step, we collect 20,000 samples and share them across all routines. This gives a sample complexity of $20,000 \times H$. We repeat the experiment 3 times and found that each time, the model was able to perfectly detect the latent child function, learn a $1/2$-policy cover, and estimate the model with error $< 0.01$. This is in sync with our theoretical findings and demonstrates the empirical use of `FactoRL`. We will make the code available at: `https://github.com/cereb-rl`.

**Conclusion.**   We introduce Factored Block MDPs that model the real-world difficulties of rich-observation environments with exponentially large latent state spaces. We also propose a provable RL algorithm called `FactoRL` for solving a large class of Factored Block MDPs. We hope the setting and ideas in `FactoRL` will stimulate both theoretical and empirical work in this important area.

---

[3]We can also easily modify our proof to use cross-entropy loss by using generalization bounds for log-loss (see Appendix E in Agarwal et al. (2020))

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

APPENDIX ORGANIZATION

This appendix is organized as follows.

- Appendix A provides a list of notations used in this paper.
- Appendix B covers related work
- Appendix C describes the independence test algorithm and its sample complexity guarantees
- Appendix D provides sample complexity guarantees for `FactoRL`
- Appendix E provides list of supporting results
- Appendix F provides details of the experimental setup and optimization

## A  NOTATIONS

We present notations and their definition in Table 1. In general, calligraphic notations represent sets. All logarithms are with respect to base $e$.

## B  RELATED WORK

There is a rich literature on sample-efficient reinforcement learning in tabular MDPs with a small number of observed states (Brafman & Tennenholtz, 2002; Strehl et al., 2006; Kearns & Singh, 2002; Jaksch et al., 2010; Azar et al., 2017; Jin et al., 2018). While recent state-of-the-art results along this line achieve near-optimal sample complexity, these algorithms do not exploit the latent structure in the environment, and therefore, cannot scale to many practical settings such as rich-observation environments with possibly a large number of factored latent states.

In order to overcome this challenge, one line of research has been focusing on factored MDPs (Kearns & Koller, 1999; Guestrin et al., 2002; 2003; Strehl et al., 2010), which allow a combinatorial number of observed states with a factorized structure. Planning in factored MDPs is EXPTIME-complete (Mausam, 2012) yet often tractable in practice, with factored MDPs statistically learnable with polynomial samples in the number of parameters that encode the factored MDP (Osband & Van Roy, 2014; Li et al., 2011). There has also been several empirical works that either focus on the factored state space setting (e.g., Kim & Mnih (2018); Thomas et al. (2018); Laversanne-Finot et al. (2018); Miladinović et al. (2019)), or the factored action space setting (e.g., He et al. (2016); Sharma et al. (2017)). However, these works do not directly address our problem and do not provide sample complexity guarantees.

Another line of work focuses on exploration in a rich observation environment. Empirical results (Tang et al., 2017; Chen et al., 2017; Bellemare et al., 2016; Pathak et al., 2017) have achieved inspiring performance on several RL benchmarks, while theoretical works (Krishnamurthy et al., 2016; Jiang et al., 2017) show that it is information-theoretically possible to explore these environments. As discussed before, recent works of Du et al. (2019), Misra et al. (2020) and Feng et al. (2020) provide computationally and sample efficient algorithms for Block MDP which is a rich-observation setting with a latent non-factored state space. Nevertheless, this line of results crucially relies on the number of latent states being relatively small.

Finally, we comment that the contrastive learning technique used in this paper has been used by other reinforcement learning algorithms for learning feature representation (e.g., Kim et al. (2019); Nachum et al. (2019); Srinivas et al. (2020)) without theoretical guarantee.

## C  INDEPENDENCE TESTING USING NOISE CONTRASTIVE ESTIMATION

In this section, we introduce the independence testing algorithm, Algorithm 5 and provide its theoretic guarantees. Algorithm 5 will be used in Algorithm 2 as a subroutine for determining if two atoms are emitted from the same latent factor. We comment that the high-level idea of Algorithm 5 is similar to Sen et al. (2017), which reduces independence testing to regression by adding imposter samples.

| Notation | Description |
|---|---|
| $\mathbb{N}$ | Space of natural numbers $\{1, 2, \cdots, \}$ |
| $\mathbb{Z}$ | Space of integers $\{\cdots, -2, -1, 0, 1, 2, \cdots, \}$ |
| $\mathbb{Z}_{\geq 0}$ | Space of positive integers $\{0, 1, 2, \cdots, \}$, equivalent to $\mathbb{N} \cup \{0\}$. |
| $[N]$ | Given $n \in \mathbb{N}$, this notation denotes the set $\{1, 2, \cdots, N\}$. |
| $\mathscr{C}_{\leq k}(\mathcal{U})$ | Given an ordered set $\mathcal{U}$ and $k \in \mathbb{Z}_{\geq 0}$, this denotes the set of all ordered subsets of $\mathcal{U}$ with at most $k$ elements including the empty set. |
| $u[\mathcal{I}]$ | For a given $n$-dimensional vector $u$ and $\mathcal{I} = (i_1, i_2, \cdots, i_k) \in 2^{[n]}$, $u[\mathcal{I}]$ denotes the $k$-dimensional vector $(u[i_1], u[i_2], \cdots, u[i_k])$ |
| $[\mathcal{I}; \mathcal{J}]$ | Denotes concatenation of two ordered sets $\mathcal{I}$ and $\mathcal{J}$ |
| $\Delta(\mathcal{U})$ | Set of all possible distributions over the set $\mathcal{U}$. |
| $\mathscr{X}$ | Set of atoms |
| $\mathcal{X}$ | Set of all observations |
| $\mathcal{X}_h$ | Set of all observations reachable at time step $h$ |
| $m$ | Number of atoms in the observation |
| $x$ | Observation consisting of $m$ atoms denoted by $(x[1], \cdots, x[m])$. |
| $\mathcal{S}$ | Set of latent states |
| $\mathcal{S}_h$ | Set of latent states reachable at time step $h$ |
| $d$ | Number of latent factors |
| $s$ | Latent state represented by a vector in $\{0, 1\}^d$. |
| $\phi^\star$ | Decoder function which maps an observation $x \in \mathcal{X}$ to latent state $s \in \mathcal{S}$. |
| $\phi_i^\star$ | Decoder function which maps $x \in \mathcal{X}$ to its $i^{th}$ latent state value. Formally, $\forall x \in \mathcal{X}$ and $i \in [d]$, $\phi_i^\star(x) = \phi^\star(x)[i]$. |
| $\mathcal{A}$ | Action space |
| $T : \mathcal{S} \times \mathcal{A} \to \Delta(\mathcal{S})$ | Transition function on the latent dynamics |
| $H$ | Horizon of the problem denoting the maximum number of actions an agent takes in a single episode. |
| $pt_h : [d] \to 2^{[d]}$ | Parent function for time step $h$. We drop $h$ when it is clear. |
| $ch_h : [d] \to 2^{[m]}$ | Child function for time step $h$. By definition, for any $i, j \in [d]$ and $i \neq j$ we have $ch_h(i) \cap ch_h(j) = \emptyset$. We drop $h$ when clear. |
| $q : \mathcal{S} \to \Delta(\mathcal{X})$ | Emission function that generates $x \sim q(\cdot \mid s)$ given $s \in \mathcal{S}$. |
| $T_i$ | $i^{th}$ product term in the transition function. Formally, for $s, s' \in \mathcal{S}$ and $a \in \mathcal{A}$ we have $T(s' \mid s, a) = \prod_{i=1}^d T_i(s'[i] \mid s[pt(i)], a)$. |
| $q_i$ | $i^{th}$ product term in emission function. Formally for $x \in \mathcal{X}$ with $\phi^\star(x) = s$ we have $q(x \mid s) = \prod_i^d q_i(x[ch(i)] \mid s[i])$ |
| $\mathcal{F}, \mathcal{G}$ | Regressor classes. We will denote individual member of the class as $f \in \mathcal{F}$ and $g \in \mathcal{G}$ |
| $\Phi : \mathcal{X} \to \{0, 1\}$ | Decoder class. |

Table 1: List of notations and their definition.

## C.1 Algorithm description

Let $D \in \Delta(\mathcal{S}_{h-1} \times \mathcal{A})$ be our roll-in distribution that induces a probability distribution $\mathbb{P}_D \in \Delta(\mathcal{X}_h)$ over observations at time step $h$. Let $u, v \in [m]$ be two different atoms, and $\mathbb{P}_D(x[u], x[v]), \mathbb{P}_D(x_h[u])$ and $\mathbb{P}_D(x_h[v])$ be the joint and marginal distributions over $x_h[u], x_h[v]$ with respect to roll-in distribution $D$. The goal of our algorithm to determine if $x_h[u]$ and $x_h[v]$ are independent under $\mathbb{P}_D$.

---

**Algorithm 5** $\texttt{IndTest}(\mathcal{F}, \mathcal{D}, u, v, \beta)$ Oraclized Independency Test. We initialize $\mathcal{D}_{\text{train}} = \emptyset$.

1: Initialize $\mathcal{D}_{\text{train}} = \emptyset$ and sample $z_1, z_2, \cdots, z_n \sim \texttt{Bern}(\frac{1}{2})$.

2: **for** $i \in [n]$ **do**

3:     **if** $z_i = 1$ **then**

4:         $\mathcal{D}_{\text{train}} \leftarrow \mathcal{D}_{\text{train}} \cup \{(x^{(i,1)}[u], x^{(i,1)}[v], 1)\}$.

5:     **else**

6:         $\mathcal{D}_{\text{train}} \leftarrow \mathcal{D}_{\text{train}} \cup \{(x^{(i,1)}[u], x^{(i,2)}[v], 0)\}$.

7: Compute $\hat{f} := \arg\min_{f \in \mathcal{F}} L(\mathcal{D}_{\text{train}}, f)$, where

$$L(\mathcal{D}_{\text{train}}, f) := \frac{1}{n} \sum_{(x[u], x[v], z) \in \mathcal{D}_{\text{train}}} \{f(x[u], x[v]) - z\}^2.$$

8: **return** Independent **if** $L(\mathcal{D}_{\text{train}}, \hat{f}) > 0.25 - \beta^2/10^3$ **else return** Dependent.

---

We solve this task using the $\texttt{IndTest}$ algorithm (Algorithm 5) which takes as input a dataset of observation pairs $\mathcal{D} = \{(x^{(i,1)}, x^{(i,2)})\}_{i=1}^n$ where $x^{(i,1)}, x^{(i,2)} \sim \mathbb{P}_D(\cdot, \cdot)$, and a scalar $\beta \in (0, 1)$. We use $\mathcal{D}$ to create a dataset $\mathcal{D}_{\text{train}}$ of real and imposter atom pairs $(x[u], x[v])$. This is done by taking every datapoint in $\mathcal{D}$ and sampling a Bernoulli random variable $z_i \sim \texttt{Bern}(1/2)$ (line 1). If $z_i = 1$ then we add the *real pair* $(x^{(i,1)}[u], x^{(i,1)}[v], 1)$ to $\mathcal{D}_{\text{train}}$ (line 4), otherwise we add the *imposter pair* $(x^{(i,1)}[u], x^{(i,2)}[v], 0)$ (line 6). We train a classifier to predict if a given atom pair $(x[u], x[v])$ is real or an imposter (line 7). The Bayes optimal classifier for this problem is given by:

$$\forall x \in \text{supp}(\mathbb{P}_D), \quad f_D^\star(x[u], x[v]) := \mathbb{P}_D(z = 1 \mid x[u], x[v]) = \frac{\mathbb{P}_D(x[u], x[v])}{\mathbb{P}_D(x[u], x[v]) + \mathbb{P}_D(x[u])\mathbb{P}(x[v])}.$$

If $x[u]$ and $x[v]$ are independent then we have $\mathbb{P}_D(x[u], x[v]) = \mathbb{P}_D(x[u])\mathbb{P}_D(x[v])$ everywhere on the support of $\mathbb{P}_D$. This implies $f_D^\star(x) = \frac{1}{2}$ and its training loss will concentrate around 0.25. Intuitively, this can be interpreted as the classifier having no information to tell real samples from imposter samples. However, if $x[u], x[v]$ are dependent and $\|\mathbb{P}_D(x[u], x[v]) - \mathbb{P}_D(x[u])\mathbb{P}_D(x[v])\|_{\text{TV}} \geq \beta$ then we can show the training loss of $f^\star$ is less than $0.25 - \mathcal{O}(\beta^2)$ with high probability. The remainder of this section is devoted to a rigorous proof for this argument.

## C.2 Analysis of Algorithm 5

Before analyzing Algorithm 5, we want to slightly simplify the problem in terms of notations. We introduce two simple notations $X$ and $Y$ which represents the random variables $x[u]$ and $x[v]$, respectively. We will simply use $D$ to denote the joint distribution of $X$ and $Y$. Define $D_{\text{train}}$ to be the distribution of the training data $(X_{\text{train}}, Y_{\text{train}}, z)$ produced in Algorithm 5. It's easy to verify that

$$D_{\text{train}}(X_{\text{train}}, Y_{\text{train}}, z) = \frac{1}{2} \left[ z D(X_{\text{train}}, Y_{\text{train}}) + (1 - z) D(X_{\text{train}}) D(Y_{\text{train}}) \right]. \tag{1}$$

Suppose the distribution $D$ is specially designed such that at least one of the following hypothesis holds (which can be guaranteed when we invoke Algorithm 5):

$$H_0 : \|D(X, Y) - D(X)D(Y)\|_1 \geq \frac{\beta}{2}$$

$$v.s. \quad H_1 : \|D(X, Y) - D(X)D(Y)\|_1 \leq \frac{\beta^2}{10^3}.$$

In the remaining part, we will prove that Algorithm 5 can correctly distinguish between $H_0$ and $H_1$ with high probability.

### C.2.1 Two properties of the Bayes optimal classifier

Our first lemma shows that the Bayes optimal classifier for the optimization problem in line 7 is a constant function equal to $1/2$ if $X$ and $Y$ are independent.

**Lemma 1** (Bayes Optimal Classifier for Independent Case). *In Algorithm 5, if $X$ and $Y$ (atoms $u$ and $v$) are independent under distribution D, then for the optimization problem in line 7 , the Bayes optimal classifier is given by:*

$$\forall (X_{\text{train}}, Y_{\text{train}}) \qquad f^\star(X_{\text{train}}, Y_{\text{train}}) = \frac{1}{2}.$$

*Proof.* From Bayes rule we have:

$$
\begin{aligned}
&f^\star(X_{\text{train}}, Y_{\text{train}})\\
=&D_{\text{train}}(z = 1 \mid X_{\text{train}}, Y_{\text{train}})\\
=&\frac{D_{\text{train}}(X_{\text{train}}, Y_{\text{train}} \mid z = 1)D_{\text{train}}(z = 1)}{D_{\text{train}}(X_{\text{train}}, Y_{\text{train}} \mid z = 1)D_{\text{train}}(z = 1) + D_{\text{train}}(X_{\text{train}}, Y_{\text{train}} \mid z = 0)D_{\text{train}}(z = 0)}\\
=&\frac{D_{\text{train}}(X_{\text{train}}, Y_{\text{train}} \mid z = 1)}{D_{\text{train}}(X_{\text{train}}, Y_{\text{train}} \mid z = 1) + D_{\text{train}}(X_{\text{train}}, Y_{\text{train}} \mid z = 0)},
\end{aligned}
$$

where the last identity uses $D_{\text{train}}(z = 1) = D_{\text{train}}(z = 0) = 1/2$.

When $z = 0$, we collect $X_{\text{train}}$ and $Y_{\text{train}}$ from two independent samples. Therefore, we have $D_{\text{train}}(X_{\text{train}}, Y_{\text{train}} \mid z = 0) = D(X_{\text{train}})D(Y_{\text{train}})$.

When $z = 1$, using the fact that $X_{\text{train}}$ and $Y_{\text{train}}$ are independent under distribution $D$, we also have $D_{\text{train}}(X_{\text{train}}, Y_{\text{train}} \mid z = 1) = D(X_{\text{train}}, Y_{\text{train}}) = D(X_{\text{train}})D(Y_{\text{train}})$.

Consequently, $f^\star(X_{\text{train}}, Y_{\text{train}}) \equiv 1/2$. $\qquad\square$

Our second lemma provides an upper bound for the expected training loss of the Bayes Optimal Classifier. Later we will use this lemma to show the training loss is less than $0.25 - \mathcal{O}(\beta^2)$ with high probability when $H_0$ holds.

**Lemma 2** (Square Loss of the Bayes Optimal Classifier). *In Algorithm 5 line 7, the Bayes optimal classifier has expected square loss*

$$\mathbb{E}_{D_{\text{train}}} L(f^\star, \mathcal{D}_{\text{train}}) \leq \frac{1}{4} - \left( \frac{1}{2} \mathbb{E}_D \left[ \left| \frac{1}{2} - \frac{D(X,Y)}{D(X)D(Y) + D(X,Y)} \right| \right] \right)^2,$$

*Proof.* Recall the formula of the Bayes optimal classifier in Lemma 1,

$$f^\star(X, Y) = \frac{D(X,Y)}{D(X)D(Y) + D(X,Y)}.$$

Plugging it into the square loss, we obtain

$$
\begin{aligned}
&\mathbb{E}_{D_{\text{train}}} \left[ (f^\star(X,Y) - y)^2 \right]\\
=&\mathbb{E}_{D_{\text{train}}} \left[ f^\star(X,Y) \left( f^\star(X,Y) - 1 \right)^2 + (1 - f^\star(X,Y)) \left( f^\star(X,Y) \right)^2 \right]\\
=&\mathbb{E}_{D_{\text{train}}} \left[ f^\star(X,Y) \left( 1 - f^\star(X,Y) \right) \right]\\
=&\frac{1}{4} - \mathbb{E}_{D_{\text{train}}} \left[ \left( \frac{1}{2} - \frac{D(X,Y)}{D(X)D(Y) + D(X,Y)} \right)^2 \right]\\
\leq&\frac{1}{4} - \left( \mathbb{E}_{D_{\text{train}}} \left[ \left| \frac{1}{2} - \frac{D(X,Y)}{D(X)D(Y) + D(X,Y)} \right| \right] \right)^2\\
\leq&\frac{1}{4} - \left( \frac{1}{2} \mathbb{E}_D \left[ \left| \frac{1}{2} - \frac{D(X,Y)}{D(X)D(Y) + D(X,Y)} \right| \right] \right)^2. \qquad\square
\end{aligned}
$$

### C.2.2 THREE USEFUL LEMMAS

To proceed, we need to take a detour and prove three useful technical lemmas.

**Lemma 3.** *Let $\mu$ and $\nu$ be two probability measures defined on a countable set $\mathcal{X}$. If $\|\mu - \nu\|_{\mathrm{TV}} \geq c$, then*

$$\mathbb{E}_{x \sim \mu} \left| \frac{\mu(x)}{\mu(x) + \nu(x)} - \frac{1}{2} \right| \geq \frac{c}{4}.$$

*Proof.*

$$
\begin{aligned}
\mathbb{E}_{x \sim \mu} \left[ \left| \frac{\mu(x)}{\mu(x) + \nu(x)} - \frac{1}{2} \right| \right] &= \mathbb{E}_{x \sim \mu} \left[ \frac{1}{2} \left| \frac{\mu(x) - \nu(x)}{\mu(x) + \nu(x)} \right| \right] \\
&\geq \frac{1}{2} \mathbb{E}_{x \sim \mu} \left[ \mathbf{1}\{\mu(x) > \nu(x)\} \left| \frac{\mu(x) - \nu(x)}{\mu(x) + \nu(x)} \right| \right] \\
&= \frac{1}{2} \sum_{x \in \mathcal{X}} \mu(x) \left[ \mathbf{1}\{\mu(x) > \nu(x)\} \frac{\mu(x) - \nu(x)}{\mu(x) + \nu(x)} \right] \\
&\geq \frac{1}{4} \sum_{x \in \mathcal{X}} \left[ \mathbf{1}\{\mu(x) > \nu(x)\} (\mu(x) - \nu(x)) \right] \\
&\geq \frac{c}{4}.
\end{aligned}
$$
$\qquad\square$

**Lemma 4.** *Fix $\delta \in (0, 1)$. Then with probability at least $1 - \delta$, we have*

$$\left| L(\hat{f}, \mathcal{D}_{\mathrm{train}}) - \mathbb{E}_{D_{\mathrm{train}}} L(f^\star, \mathcal{D}_{\mathrm{train}}) \right| \leq 10 \sqrt{\frac{C(\mathcal{F}, \delta)}{n}},$$

*where $\mathcal{D}_{\mathrm{train}}$ is the training set consisting of $n$ i.i.d. samples sampled from $D_{\mathrm{train}}$, $\hat{f}$ is the empirical minimizer of $L(f, \mathcal{D}_{\mathrm{train}})$ over $\mathcal{F}$, $f^\star$ is the population minimizer, and $C(\mathcal{F}, \delta) := \ln \frac{|\mathcal{F}|}{\delta}$ is the complexity measure of function class $\mathcal{F}$.*

*Proof.* By Hoeffding's inequality and union bound, with probability at least $1 - \delta$, for every $f \in \mathcal{F}$, we have

$$|L(f, \mathcal{D}_{\mathrm{train}}) - \mathbb{E}_{D_{\mathrm{train}}} L(f, \mathcal{D}_{\mathrm{train}})| \leq 10 \sqrt{\frac{C(\mathcal{F}, \delta)}{n}}.$$

Because $\hat{f}$ is the empirical optimizer,

$$L(\hat{f}, \mathcal{D}_{\mathrm{train}}) \leq L(f^\star, \mathcal{D}_{\mathrm{train}}) \leq \mathbb{E}_{D_{\mathrm{train}}} L(f^\star, \mathcal{D}_{\mathrm{train}}) + 10 \sqrt{\frac{C(\mathcal{F}, \delta)}{n}}.$$

Because $f^\star$ is the population optimizer,

$$L(\hat{f}, \mathcal{D}_{\mathrm{train}}) \geq \mathbb{E}_{D_{\mathrm{train}}} L(\hat{f}, \mathcal{D}_{\mathrm{train}}) - \sqrt{\frac{C(\mathcal{F}, \delta)}{n}} \geq \mathbb{E}_{D_{\mathrm{train}}} L(f^\star, \mathcal{D}_{\mathrm{train}}) - 10 \sqrt{\frac{C(\mathcal{F}, \delta)}{n}}.$$

Combining the two inequalities above, we finish the proof. $\qquad\square$

For notational convenience, we introduce the following factor distribution $D_{\mathrm{factor}}$ defined on the same domain of $(X, Y, z)$:

$$D_{\mathrm{factor}}(X, Y, z) = \frac{1}{2} D(X) D(Y).$$

**Lemma 5.** *Suppose $\mathcal{F}$ contains the constant function $f \equiv 1/2$. Then with probability at least $1 - \delta$, we have*

$$\left| L(\hat{f}, \mathcal{D}_{\mathrm{train}}) - \frac{1}{4} \right| \leq 10 \sqrt{\frac{C(\mathcal{F}, \delta)}{n}} + 2 \|D_{\mathrm{train}} - D_{\mathrm{factor}}\|_{\mathrm{TV}}.$$

*Proof.* By Lemma 4, with probability at least $1 - \delta$, we have

$$\left| L(\hat{f}, \mathcal{D}_{\text{train}}) - \mathbb{E}_{D_{\text{train}}} L(f^\star, \mathcal{D}_{\text{train}}) \right| \leq 10 \sqrt{\frac{C(\mathcal{F}, \delta)}{n}}. \tag{2}$$

Noticing that $L$ is bounded by 1, we have for every $f \in \mathcal{F}$

$$|\mathbb{E}_{D_{\text{factor}}} L(f, \mathcal{D}_{\text{train}}) - \mathbb{E}_{D_{\text{train}}} L(f, \mathcal{D}_{\text{train}})| \leq 2\|D_{\text{train}} - D_{\text{factor}}\|_{\text{TV}}, \tag{3}$$

where $\mathbb{E}_{D_{\text{factor}}} L(f, \mathcal{D}_{\text{train}})$ defines the expected loss of $f$ over $\mathcal{D}_{\text{train}}$ where $\mathcal{D}_{\text{train}}$ consists of samples i.i.d. sampled from $D_{\text{factor}}$.

Since $y$ is a symmetric Bernoulli r.v. independent of $(X, Y)$ under distribution $D_{\text{factor}}$, we have

$$\min_{f \in \mathcal{F}} \mathbb{E}_{D_{\text{factor}}} L(f, \mathcal{D}_{\text{train}}) = \frac{1}{4}. \tag{4}$$

Using the inequality $|\min_f L_1(f) - \min_f L_2(f)| \leq \max_f |L_1(f) - L_2(f)|$ for any functionals $L_1, L_2$, along with (4) and (3) we bound the minimum loss under distribution $D_{\text{train}}$ as:

$$\left| \min_{f \in \mathcal{F}} \mathbb{E}_{D_{\text{train}}} L(f, \mathcal{D}_{\text{train}}) - \frac{1}{4} \right| \leq 2\|D_{\text{train}} - D_{\text{factor}}\|_{\text{TV}}. \tag{5}$$

Combining (5) with (2) completes the proof. $\qquad\square$

### C.2.3 MAIN THEOREM FOR ALGORITHM 5

Finally, we are ready to state and prove the main theorem for Algorithm 5.

**Theorem 2.** *Under the realizability assumption and $n \geq \Omega(\frac{C(\mathcal{F}, \delta)}{\beta^4})$, Algorithm 5 can distinguish*

$$H_0 : \|D(X, Y) - D(X)D(Y)\|_1 \geq \frac{\beta}{2}$$

$$v.s. \quad H_1 : \|D(X, Y) - D(X)D(Y)\|_1 \leq \frac{\beta^2}{10^3}$$

*correctly with probability at least $1 - \delta$.*

*Proof.* If $H_1$ holds, by Lemma 5, we have the following lower bound for the training loss of the empirical minimizer,

$$L(\hat{f}, \mathcal{D}) \geq \frac{1}{4} - 10\sqrt{\frac{C(\mathcal{F}, \delta)}{n}} - \frac{\beta^2}{10^3}. \tag{6}$$

In contrast, if $H_0$ is true, applying Lemma 4, we obtain

$$L(\hat{f}, \mathcal{D}) \leq \mathbb{E}_{D_{\text{train}}} L(f^\star, \mathcal{D}) + 10\sqrt{\frac{C(\mathcal{F}, \delta)}{n}}.$$

Invoke Lemma 2 and Lemma 3,

$$\mathbb{E}_{D_{\text{train}}} L(f^\star, \mathcal{D}_{\text{train}}) \leq \frac{1}{4} - \left( \frac{1}{2} \mathbb{E}_D \left[ \left| \frac{1}{2} - \frac{D(X, Y)}{D(X)D(Y) + D(X, Y)} \right| \right] \right)^2$$

$$\leq \frac{1}{4} - \frac{\beta^2}{256}.$$

Therefore, under $H_0$, the training loss of the empirical minimizer is upper bounded as below

$$L(\hat{f}, \mathcal{D}) \leq \frac{1}{4} - \frac{\beta^2}{256} + 10\sqrt{\frac{C(\mathcal{F}, \delta)}{n}}. \tag{7}$$

Plugging $n \geq O(\frac{C(\mathcal{F}, \delta)}{\beta^4})$ into (6) and (7), we complete the proof. $\qquad\square$

## D    THEORETICAL ANALYSIS OF FactoRL

In this section, we provide a detailed theoretical analysis of FactoRL. The structure of the algorithm is iterative making an inductive case argument appealing. We will, therefore, make an induction hypothesis for each time step that we will guarantee at the end of the time step.

**Induction Hypothesis.**    We make the following induction assumption for FactoRL under Assumption 1-3 and across all time steps. For all $t \in \{2, 3, \cdots, H\}$, at the end of time step $t$ (Algorithm 1, line 8), FactoRL finds a child function $\widehat{ch}_t : [d] \to 2^{[m]}$, a decoder $\widehat{\phi}_t : \mathcal{X} \to \{0, 1\}^d$, a transition function $\widehat{T}_t : \{0, 1\}^d \times \mathcal{A} \to \{0, 1\}^d$, and a set of policies $\Psi_t$ satisfying the following:

IH.1  $\widehat{ch}_t : [d] \to 2^m$ and $ch_t : [d] \to 2^m$ are same upto relabeling, i.e, for all $u, v \in [m]$ we have $\widehat{ch}_t^{-1}(u) = \widehat{ch}_t^{-1}(v)$ if and only if $ch_t^{-1}(u) = ch_t^{-1}(v)$. Note that a child function is invertible by definition. We can ignore this label permutation and assume $\widehat{ch}_t = ch_t$ for cleaner expressions. This can be done without any effect. We will assume $\widehat{ch}_t = ch_t$ when stating the next three induction hypothesis.

IH.2  There exists a permutation mapping $\theta_t : \{0, 1\}^d \to \{0, 1\}^d$ and $\varrho \in (0, \frac{1}{2d})$ such that for every $i \in [d]$ and $s \in \mathcal{S}_t$ we have:

$$\mathbb{P}(\widehat{\phi}_t(x_t)[i] = \theta_t^{-1}(s)[i] \mid s[i]) \geq 1 - \varrho,$$
$$\mathbb{P}(\widehat{\phi}_t(x_t) = \theta_t^{-1}(s) \mid s) \geq 1 - d\varrho \geq \frac{1}{2}$$

The two distributions are independent of the roll-in distribution at time step $t$. The first one holds as $\widehat{\phi}_t(x_t)[i]$ only depends upon the value $x_t[\widehat{ch}_t(i)] = x_t[ch_t(i)]$ which only depends on $s[i]$. The second one holds as $x_t$ is independent of everything else given $s_t$. The form of $\varrho$ will become clear the end of analysis.

IH.3  For every $s \in \mathcal{S}_{t-1}, s' \in \mathcal{S}_t$ and $a \in \mathcal{A}$ we have:

$$\left\| \widehat{T}_t(\theta_t^{-1}(s') \mid \theta_{t-1}^{-1}(s), a) - T(s' \mid s, a) \right\|_{\text{TV}} \leq 3d(\Delta_{\text{est}} + \Delta_{\text{app}}),$$

where $\Delta_{\text{est}}, \Delta_{\text{app}} > 0$ denote estimation and approximation errors whose form will become clear at the end of analysis.

IH.4  For every $s \in \mathcal{S}_t$ and $\mathcal{K} \in \mathscr{C}_{\leq 2\kappa}([d])$, let $\mathcal{Z} = s[\mathcal{K}]$ and $\widehat{\mathcal{Z}} = \theta_t^{-1}(s)[\mathcal{K}]$, then there exists a policy $\pi_{\mathcal{K}\widehat{\mathcal{Z}}} \in \Psi_t$ such that:

$$\mathbb{P}_{\pi_{\mathcal{K}\widehat{\mathcal{Z}}}}(s_t[\mathcal{K}] = \mathcal{Z}) \geq \alpha\eta_t(\mathcal{K}, \mathcal{Z}) \geq \alpha\eta_{min}.$$

**Base Case.**    In the base case ($t = 1$), we have a deterministic start state. Therefore, we can without loss of generality assume a single factor and define $\widehat{ch}_1[1] = m$. As we can also define $ch_1[1] = [m]$ without loss of generality, therefore, this trivially satisfies the induction hypothesis 1. We define $\widehat{\phi}_1 : \mathcal{X} \to [0]^d$ (Algorithm 1, line 1). This satisfies induction hypothesis 2 with $\theta_1$ being the identity map. The induction hypothesis 3 is vacuous since there is no transition function before time step 1. For the last condition, we have for any $\mathcal{K}$, $\mathcal{Z} = [0]^{|\mathcal{K}|}$ and $\widehat{\mathcal{Z}} = [0]^{|\mathcal{K}|}$. For any policy $\pi$ we have $\mathbb{P}_\pi(s_1[\mathcal{K}] = \mathcal{Z}) = \mathbb{P}_\pi(\widehat{\phi}_1(x_1)[\mathcal{K}] = \widehat{\mathcal{Z}}) = 1 \geq \frac{\eta_{min}}{2}$. Note that we never take any action from this policy, therefore, we can simply define $\Psi_1 = \emptyset$.

### D.1    GRAPH STRUCTURE IDENTIFICATION

In this section, we analyze the performance of Algorithm 2, given as input $\Psi_{h-1}, \widehat{\phi}_{h-1}, \mathcal{F}, \beta$ and $n$.

We will analyze the performance a fixed pair of atoms $u, v \in [m]$ and then apply the full result using union bound. We first state the result for the rejection sampling.

**Lemma 6.** *For policy $\pi_{\mathcal{I};\widehat{\mathcal{Z}}} \in \Psi_{h-1}$, event $\mathcal{E}_{\mathcal{I};\widehat{\mathcal{Z}}} = \{\hat{s}_{h-1}[\mathcal{I}] = \widehat{\mathcal{Z}}\}$ and $k \in \mathbb{N}$, let $D_{\mathcal{I};\widehat{\mathcal{Z}}}^{reject} :=$* RejectSamp$(\pi_{\mathcal{I};\widehat{\mathcal{Z}}}, \mathcal{E}_{\mathcal{I};\widehat{\mathcal{Z}}}, k)$ *be the distribution induced by our rejection sampling procedure. Let*

$\mathcal{Z} = \theta(\widehat{\mathcal{Z}})$ *denote the real factor values corresponding* $\widehat{\mathcal{Z}}$. *Then we have:*

$$\mathbb{P}_{D^{reject}_{\mathcal{I};\widehat{\mathcal{Z}}}}(s_{h-1}[\mathcal{I}] = \mathcal{Z}) \geq 1 - \varrho - \left(1 - \frac{\eta_{min}}{4}\right)^k. \tag{8}$$

*Proof.* From IH.4 we have $\mathbb{P}_{\pi_{\mathcal{I};\widehat{\mathcal{Z}}}}(s_{h-1}[\mathcal{I}] = \mathcal{Z}) \geq \frac{\eta_{min}}{2}$. This implies:

$$\mathbb{P}_{\pi_{\mathcal{I};\widehat{\mathcal{Z}}}}(\hat{s}_{h-1}[\mathcal{I}] = \widehat{\mathcal{Z}}) \geq \mathbb{P}_{\pi_{\mathcal{I};\widehat{\mathcal{Z}}}}(\hat{s}_{h-1}[\mathcal{I}] = \widehat{\mathcal{Z}} \mid s_{h-1}[\mathcal{I}] = \mathcal{Z})\mathbb{P}_{\pi_{\mathcal{I};\widehat{\mathcal{Z}}}}(s_{h-1}[\mathcal{I}] = \mathcal{Z})$$
$$\geq \frac{(1-d\varrho)\eta_{min}}{2} \geq \frac{\eta_{min}}{4}, \qquad \text{(using IH.2 and IH.4)}.$$

Let $a = \mathbb{P}_{\pi_{\mathcal{I};\widehat{\mathcal{Z}}}}(\hat{s}_{h-1}[\mathcal{I}] = \widehat{\mathcal{Z}})$ be the acceptance probability of event $\mathcal{E}_{\mathcal{I};\widehat{\mathcal{Z}}}$. then it is easy to see that the probability of the event occurring under $D^{reject}_{\mathcal{I};\widehat{\mathcal{Z}}}$ is:

$$\mathbb{P}_{D^{reject}_{\mathcal{I};\widehat{\mathcal{Z}}}}\left(\mathcal{E}_{\mathcal{I};\widehat{\mathcal{Z}}}\right) = a + (1-a)a + (1-a)^2 a + \cdots (1-a)^{k-1}a = 1 - (1-a)^k \geq 1 - \left(1 - \frac{\eta_{min}}{4}\right)^k.$$

We express the desired failure probability as shown:

$$\mathbb{P}_{D^{reject}_{\mathcal{I};\widehat{\mathcal{Z}}}}(s_{h-1}[\mathcal{I}] \neq \mathcal{Z}) = \mathbb{P}_{D^{reject}_{\mathcal{I};\widehat{\mathcal{Z}}}}\left(s_{h-1}[\mathcal{I}] \neq \mathcal{Z}, \hat{s}_{h-1}[\mathcal{I}] \neq \widehat{\mathcal{Z}}\right) + \mathbb{P}_{D^{reject}_{\mathcal{I};\widehat{\mathcal{Z}}}}\left(s_{h-1}[\mathcal{I}] \neq \mathcal{Z}, \hat{s}_{h-1}[\mathcal{I}] = \widehat{\mathcal{Z}}\right) \tag{9}$$

We bound the two terms below:

$$\mathbb{P}_{D^{reject}_{\mathcal{I};\widehat{\mathcal{Z}}}}\left(s_{h-1}[\mathcal{I}] \neq \mathcal{Z}, \hat{s}_{h-1}[\mathcal{I}] \neq \widehat{\mathcal{Z}}\right) \leq \mathbb{P}_{D^{reject}_{\mathcal{I};\widehat{\mathcal{Z}}}}\left(\hat{s}_{h-1}[\mathcal{I}] \neq \widehat{\mathcal{Z}}\right) \leq \left(1 - \frac{\eta_{min}}{4}\right)^k, \tag{10}$$

$$\mathbb{P}_{D^{reject}_{\mathcal{I};\widehat{\mathcal{Z}}}}\left(s_{h-1}[\mathcal{I}] \neq \mathcal{Z}, \hat{s}_{h-1}[\mathcal{I}] = \widehat{\mathcal{Z}}\right) \leq \mathbb{P}_{D^{reject}_{\mathcal{I};\widehat{\mathcal{Z}}}}\left(\hat{s}_{h-1}[\mathcal{I}] = \widehat{\mathcal{Z}} \mid s_{h-1}[\mathcal{I}] \neq \mathcal{Z},\right) \leq \varrho \tag{11}$$

Combining Equation 9, Equation 10 and Equation 11 we get:

$$\mathbb{P}_{D^{reject}_{\mathcal{I};\widehat{\mathcal{Z}}}}(s_{h-1}[\mathcal{I}] = \mathcal{Z}) = 1 - \mathbb{P}_{D^{reject}_{\mathcal{I};\widehat{\mathcal{Z}}}}(s_{h-1}[\mathcal{I}] \neq \mathcal{Z}) \geq 1 - \varrho - \left(1 - \frac{\eta_{min}}{4}\right)^k. \tag{12}$$

$\square$

We now analyze the situation for a given pair of atoms. Recall for any distribution $D \in \Delta(\mathcal{S}_{h-1})$ and $a \in \mathcal{A}$, we denote $D \circ a$ as the distribution over $\mathcal{S}_h$ where $s' \sim D \circ a$ is sampled by sampling $s \sim D$ and then $s' \sim T(. \mid s, a)$. We want to derive roll-in distributions at time step $h$, such that atoms coming from the same parent satisfy hypothesis $H_0$ and atoms coming from different parents satisfy hypothesis $H_1$ under this roll-in distribution. This will allow us to use independence test to identify the parent structure in the emission process. Specifically, we consider the roll-in distributions induced by $D^{reject}_{\mathcal{I};\widehat{\mathcal{Z}}} \circ a$ for some sets $\mathcal{I}, \mathcal{J}$ and action $a$. Instantiating the definition of these hypothesis from Appendix C, with these roll-in distributions and setting $\beta = \beta_{\min}$ gives us:

$$H_0 : \|\mathbb{P}_{D^{reject}_{\mathcal{I};\widehat{\mathcal{Z}}} \circ a}(x[u], x[v]) - \mathbb{P}_{D^{reject}_{\mathcal{I};\widehat{\mathcal{Z}}} \circ a}(x[u])\mathbb{P}_{D^{reject}_{\mathcal{I};\widehat{\mathcal{Z}}} \circ a}(x[v])\|_1 \geq \frac{\beta_{\min}}{2}$$

$$v.s. \quad H_1 : \|\mathbb{P}_{D^{reject}_{\mathcal{I};\widehat{\mathcal{Z}}} \circ a}(x[u], x[v]) - \mathbb{P}_{D^{reject}_{\mathcal{I};\widehat{\mathcal{Z}}} \circ a}(x[u])\mathbb{P}_{D^{reject}_{\mathcal{I};\widehat{\mathcal{Z}}} \circ a}(x[v])\|_1 \leq \frac{\beta^2_{\min}}{100}$$

**Lemma 7** (Same Factors). *If for two atoms $u, v$ we have $ch_h^{-1}(u) = ch_h^{-1}(v)$, i.e., they are from the same factor then the hypothesis $H_0$ is true for $D \circ a$ for any $D \in \Delta(\mathcal{S}_{h-1})$ and $a \in \mathcal{A}$. In particular, this is true for $D^{reject}_{\mathcal{I};\widehat{\mathcal{Z}}} \circ a$ for any choice of sets $\mathcal{I}, \mathcal{J}$ and action $a$.*

*Proof.* Follows trivially from Assumption 2. $\square$

**Lemma 8** (Different Factors). *If for two atoms $u, v$ we have $ch_h^{-1}(u) = i$ and $ch_h^{-1}(v) = j$ and $i \neq j$, then if $\mathcal{I}$ contains $pt_h(i) \cup pt_h(j)$, then for $\varrho \leq \frac{\beta^2_{min}}{1200}$ and $k \geq \frac{8}{\eta_{min}} \ln\left(\frac{30}{\beta_{min}}\right)$, the hypothesis $H_1$ holds for $D^{reject}_{\mathcal{I};\widehat{\mathcal{Z}}} \circ a$ for any $\widehat{\mathcal{Z}}$ such that $\pi_{\mathcal{I};\widehat{\mathcal{Z}}} \in \Psi_{h-1}$ and $a \in \mathcal{A}$.*

*Proof.* Let $D' \in \Delta(\mathcal{S}_{h-1})$ be a distribution that deterministically sets $s_{h-1}[\mathcal{I}] = \mathcal{Z}$. Then it is easy to verify that $\mathbb{P}_{D' \circ a}(x_h[u], x_h[v]) = \mathbb{P}_{D' \circ a}(x_h[u])\mathbb{P}_{D' \circ a}(x_h[v])$ for any $a \in \mathcal{A}$ and $x_h \in \mathcal{X}_h$.

Then for any $\widehat{\mathcal{Z}}$ and action $a \in \mathcal{A}$ we have using triangle inequality:

$$\left| \mathbb{P}_{D^{\text{reject}}_{\mathcal{I};\widehat{\mathcal{Z}}} \circ a}(x_h[u], x_h[v]) - \mathbb{P}_{D^{\text{reject}}_{\mathcal{I};\widehat{\mathcal{Z}}} \circ a}(x_h[u])\mathbb{P}_{D^{\text{reject}}_{\mathcal{I};\widehat{\mathcal{Z}}} \circ a}(x_h[v]) \right|_1$$

$$\leq \left| \mathbb{P}_{D^{\text{reject}}_{\mathcal{I};\widehat{\mathcal{Z}}} \circ a}(x_h[u], x_h[v]) - \mathbb{P}_{D' \circ a}(x_h[u], x_h[v]) \right|_1 +$$

$$\left| \mathbb{P}_{D' \circ a}(x_h[u]) - \mathbb{P}_{D^{\text{reject}}_{\mathcal{I};\widehat{\mathcal{Z}}} \circ a}(x_h[u]) \right|_1 + \left| \mathbb{P}_{D' \circ a}(x_h[v]) - \mathbb{P}_{D^{\text{reject}}_{\mathcal{I};\widehat{\mathcal{Z}}} \circ a}(x_h[v]) \right|_1$$

As $x_h[u]$ and $x_h[v]$ come from different factors, therefore, we have

$$\mathbb{P}(x_h[u], x_h[v] \mid s_{h-1}, a) = \mathbb{P}(x_h[u] \mid s_{h-1}[\mathcal{I}], a)\mathbb{P}(x_h[v] \mid s_{h-1}[\mathcal{I}], a).$$

We use this to bound the three terms in the summation above.

$$\left| \mathbb{P}_{D^{\text{reject}}_{\mathcal{I};\widehat{\mathcal{Z}}} \circ a}(x_h[u], x_h[v]) - \mathbb{P}_{D' \circ a}(x_h[u], x_h[v]) \right|_1$$

$$= \sum_{x_h[u], x_h[v]} \left| \sum_{s_{h-1}[\mathcal{I}]} \mathbb{P}(x_h[u] \mid s_{h-1}[\mathcal{I}], a)\mathbb{P}(x_h[v] \mid s_{h-1}[\mathcal{I}], a) \left\{ \mathbb{P}_{D^{\text{reject}}_{\mathcal{I};\widehat{\mathcal{Z}}}}(s_{h-1}[\mathcal{I}]) - \mathbb{P}_{D'}(s_{h-1}[\mathcal{I}]) \right\} \right|$$

$$\leq \sum_{s_{h-1}[\mathcal{I}]} \sum_{x_h[u], x_h[v]} \mathbb{P}(x_h[u] \mid s_{h-1}[\mathcal{I}], a)\mathbb{P}(x_h[v] \mid s_{h-1}[\mathcal{I}], a) \left| \mathbb{P}_{D^{\text{reject}}_{\mathcal{I};\widehat{\mathcal{Z}}}}(s_{h-1}[\mathcal{I}]) - \mathbb{P}_{D'}(s_{h-1}[\mathcal{I}]) \right|$$

$$\leq \sum_{s_{h-1}[\mathcal{I}]} \left| \mathbb{P}_{D^{\text{reject}}_{\mathcal{I};\widehat{\mathcal{Z}}}}(s_{h-1}[\mathcal{I}]) - \mathbb{P}_{D'}(s_{h-1}[\mathcal{I}]) \right|$$

$$= \left| 1 - \mathbb{P}_{D^{\text{reject}}_{\mathcal{I};\widehat{\mathcal{Z}}}}(s_{h-1}[\mathcal{I}] = \mathcal{Z}) \right| + \sum_{s_{h-1}[\mathcal{I}] \neq \mathcal{Z}} \mathbb{P}_{D^{\text{reject}}_{\mathcal{I};\widehat{\mathcal{Z}}}}(s_{h-1}[\mathcal{I}])$$

$$= 2 \left( 1 - \mathbb{P}_{D^{\text{reject}}_{\mathcal{I};\widehat{\mathcal{Z}}}}(s_{h-1}[\mathcal{I}] = \mathcal{Z}) \right) \leq 2\varrho + 2 \left( 1 - \frac{\eta_{min}}{4} \right)^k.$$

The other two terms are bounded similarly which gives us:

$$\left| \mathbb{P}_{D^{\text{reject}}_{\mathcal{I};\widehat{\mathcal{Z}}} \circ a}(x_h[u], x_h[v]) - \mathbb{P}_{D^{\text{reject}}_{\mathcal{I};\widehat{\mathcal{Z}}} \circ a}(x_h[u])\mathbb{P}_{D' \circ a}(x_h[v]) \right|_1 \leq 6\varrho + 6 \left( 1 - \frac{\eta_{min}}{4} \right)^k.$$

We want this quantity to be less than $\frac{\beta_{\min}^2}{100}$ to satisfy hypothesis $H_1$. We distribute the errors equally and use $\ln(1 + a) \leq a$ for all $a > -1$ to get:

$$\varrho \leq \frac{\beta_{\min}^2}{1200}, \qquad k \geq \frac{8}{\eta_{min}} \ln \left( \frac{30}{\beta_{\min}} \right). \tag{13}$$

$\square$

**Theorem 3** (Learning $\widehat{ch}_h$). *Fix $\delta_{ind} \in (0, 1)$. If $\varrho \leq \frac{\beta_{min}^2}{1200}$ and $k \geq \frac{8}{\eta_{min}} \ln \left( \frac{30}{\beta_{min}} \right)$ and $n_{ind} \geq \mathcal{O}\left( \frac{1}{\beta_{min}^4} \ln \frac{m^2 |\mathcal{A}||\mathcal{F}|(2ed)^{2\kappa+1}}{\delta_{ind}} \right)$, then learned $\widehat{ch}_h$ is equivalent to $ch_h$ upto label permutation with probability at least $1 - \delta_{ind}$.*

*Proof.* For any pair of atom $u, v$, if they are from the same factor then $H_0$ holds from Lemma 7 and `IndTest` mark them dependent with probability at least $1 - \delta$. This holds for every triplet of $\mathcal{I}, \mathcal{Z}, a$ and there are at most $|\mathcal{A}|(2ed)^{2\kappa+1}$, of them. Hence, from union bound we mark $u, v$ correctly as coming from different factors with probability at least $1 - |\mathcal{A}|(2ed)^{2\kappa+1}\delta$.

If $u$ and $v$ have different factors then for any $\mathcal{I}$ containing the parents of both of them, and any value of $\mathcal{Z}$ and $a$, $H_1$ always holds from Lemma 8 and `IndTest` marks them as independent. Note that

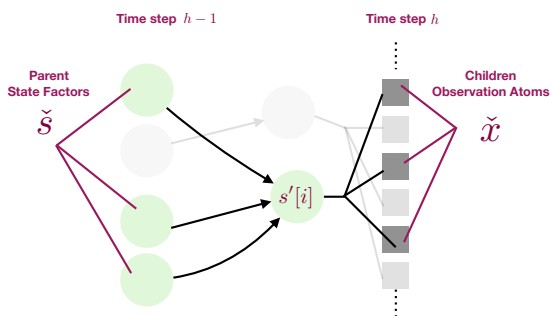

Figure 2: Scheme showing the important variables for the decoding step.

such an $\mathcal{I}$ will exists since the we iterate over all possible sets of size upto $2\kappa$. Hence, with probability at least $1 - |\mathcal{A}|2^\kappa\delta$, we find $u$ and $v$ to be independent for every $\mathcal{Z}$ and $a$. Hence, our algorithm correctly will mark them as coming from different factors.

For a given $u$ and $v$, we correctly predict their output with probability at least $1 - |\mathcal{A}|(2ed)^{2\kappa+1}\delta$. Therefore, using union bound we correctly output right result for each $u$ and $v$ with probability at least $1 - |\mathcal{A}|m^2(2ed)^{2\kappa+1}\delta$. From Theorem 2, we require $n_{\mathrm{ind}} \geq \mathcal{O}\left(\frac{1}{\beta_{\min}^4}\ln\frac{|\mathcal{F}|}{\delta}\right)$. Binding $|\mathcal{A}|(2ed)^{2\kappa+1}\delta$ to $\delta_{\mathrm{ind}}$ then gives us the required value of $n_{\mathrm{ind}}$ to achieve a success probability of at least $1 - \delta_{\mathrm{ind}}$. If we correctly assess the dependence for every pair of atoms correctly, then trivially partitioning them using the dependence equivalence relation gives us $\widehat{ch}_h$ which is same as $ch_h$ upto label permutation. $\quad\square$

### D.2 Learning a State Decoder

We focus on the task of learning an abstraction at time step $h$ using Algorithm 3. We have access to $\hat{ch}_h$ which is same as $ch_h$ upto label permutation. We showed how to do this in Appendix C. We will ignore the label permutation to avoid having to complicate our notations. This would essentially mean that we will recover a backward decoder $\widehat{\phi}_h = \left(\widehat{\phi}_{h1}, \cdots, \widehat{\phi}_{hd}\right)$, where there is a bijection between $\left\{\widehat{\phi}_{hi}\right\}_i$ and $\left\{\phi_j^\star\right\}_j$.

As we learn each decoder $\{\widehat{\phi}_{hi}\}$ independently of each other, therefore, we will focus on learning the decoder $\widehat{\phi}_{hi}$ for a fixed $i$. The same analysis will hold for other decoder and with application of union bound, we will establish guarantees for all decoders. Further, since we are learning the decoder at a fixed time step $h$, therefore, we will drop the $h$ from the subscript for brevity. We use additional shorthand described below and visualize some of them in Figure 2.

- $s$ and $x$ denote a state $s_{h-1}$ and an observation $x_{h-1}$ at time step $h - 1$
- $s'$ and $x'$ denotes state $s_h$ and observation $x_h$ at time step $h$
- $s'[i]$ denotes $i^{th}$ factor of state at time step $h$
- $\check{s}$ denotes $s[pt_h(i)]$ which is the set of parent factors of $s'[i]$. Recall that from the factorization assumption, we have $T(s'[i] \mid s, a) = T_i(s'[i] \mid \check{s}, a)$ for any $s, a$.
- $\widehat{\phi}_i$ denotes $\widehat{\phi}_{hi}$ decoder for $i^{th}$ factor at time step $h$
- $\omega$ denotes $pt(i)$ which is the set of indices of atoms emitted by $s'[i]$.
- $\check{x}$ denotes $x'[ch_h(i)]$ which is the collection of atoms generated by $s'[i]$.
- $N = |\Psi_{h-1}|$ is the size of policy cover for previous time step.

Let $\mathcal{D} = \{(x^{(k)}, a^{(k)}, \check{x}^{(k)}, y^{(k)})\}_{k=1}^{n_{\mathrm{abs}}}$ be a dataset of $n_{\mathrm{abs}}$ real transitions $(y = 1)$ and imposter transitions $(y = 0)$ collected in Algorithm 3, line 2-4. We define the expected risk minimizer (ERM) solution as:

$$\hat{g}_i = \arg\min_{g \in \mathcal{G}} \frac{1}{n_{\mathrm{abs}}} \sum_{k=1}^{n_{\mathrm{abs}}} \left\{g(x^{(k)}, a^{(k)}, \check{x}^{(k)}) - y^{(k)}\right\}^2 \tag{14}$$

Recall that by the structure of $\mathcal{G}$, we have $\hat{g}_i = (\hat{u}_i, \widehat{\phi}_i)$ where $\hat{w}_i \in \mathcal{W}_2$ and $\widehat{\phi}_i \in \Phi : \mathscr{X}^* \to \{0, 1\}$ is the learned decoder. Our algorithm only cares about the properties of the decoder and we throw away the regressor $\hat{u}_i$.

Let $D(x, a, \check{x})$ be the marginal distribution over transitions. We get the marginal distribution by marginalizing out the real ($y = 1$) and imposter transition ($y = 0$). We also define $D(x, a)$ as the marginal distribution over $x, a$. We have $D(x, a) = \mu_{h-1}(x) \frac{1}{|\mathcal{A}|}$ as both real and imposter transitions involve sampling $x \sim \mu_{h-1}$ and taking action uniformly. Recall that $\mu_{h-1}$ is generated by roll-in with a uniformly selected policy in $\Psi_{h-1}$ till time step $h - 1$. Let $P(x, a, \check{x} \mid y = 1)$ be the probability of a transition being real and $P(x, a, \check{x} \mid y = 0)$ be the probability of the transition being imposter. We can express these probabilities as:

$$P(x, a, \check{x} \mid y = 1) = D(x, a)T(\check{x} \mid x, a), \qquad P(x, a, \check{x} \mid y = 0) = D(x, a)\rho(\check{x}), \qquad (15)$$

where $\rho(\check{x}) = \mathbb{E}_{(x,a) \sim D}[T(\check{x} \mid x, a)]$ is the marginal distribution over $\check{x}$. We will overload the notation $\rho$ to also define $\rho(x') = \mathbb{E}_{(x,a) \sim D}[T(x' \mid x, a)]$. Lastly, we can express the marginal distribution over transition as:

$$D(x, a, \check{x}) = P(x, a, \check{x} \mid y = 1)P(y = 1) + P(x, a, \check{x} \mid y = 0)P(y = 0)$$

$$= \frac{\mu_{h-1}(x)}{2|\mathcal{A}|} \{T(\check{x} \mid x, a) + \rho(\check{x})\}$$

We start by expressing the Bayes optimal classifier for problem in Equation 14.

**Lemma 9** (Bayes Optimal Classifier). *Bayes optimal classifier $g^\star$ for problem in Equation 14 is given by:*

$$\forall (x, a, \check{x}) \in \text{supp}\, D, \qquad g^\star(x, a, \check{x}) = \frac{T_i(\phi_i^\star(\check{x}) \mid \phi^\star(x)[pt(i)], a)}{T_i(\phi_i^\star(\check{x}) \mid \phi^\star(x)[pt(i)], a) + \rho(\phi_i^\star(\check{x}))} \qquad (16)$$

*Proof.* The Bayes optimal classifier is given by $g^\star(x, a, \check{x}) = P(y = 1 \mid x, a, \check{x})$ which can be expressed using Bayes rule as:

$$P(y = 1 \mid x, a, \check{x}) = \frac{P(x, a, \check{x} \mid y = 1)P(y = 1)}{P(x, a, \check{x} \mid y = 1)P(y = 1) + P(x, a, \check{x} \mid y = 0)P(y = 0)}$$

$$= \frac{P(x, a, \check{x} \mid y = 1)}{P(x, a, \check{x} \mid y = 1) + P(x, a, \check{x} \mid y = 0)}, \quad \text{using } p(y) = \text{Bern}(1/2)$$

$$= \frac{D(x, a)T(\check{x} \mid x, a)}{D(x, a)T(\check{x} \mid x, a) + D(x, a)\rho(\check{x})}$$

$$= \frac{T(\check{x} \mid x, a)}{T(\check{x} \mid x, a) + \rho(\check{x})}$$

$$= \frac{q_i(\check{x} \mid \phi_i^\star(\check{x}))T_i(\phi_i^\star(\check{x}) \mid x, a)}{q_i(\check{x} \mid \phi_i^\star(\check{x}))T_i(\phi_i^\star(\check{x}) \mid x, a) + q_i(\check{x} \mid \phi_i^\star(\check{x}))\rho(\phi_i^\star(\check{x}))}$$

$$= \frac{T_i(\phi_i^\star(\check{x}) \mid x, a)}{T_i(\phi_i^\star(\check{x}) \mid x, a) + \rho(\phi_i^\star(\check{x}))} = \frac{T_i(\phi_i^\star(\check{x}) \mid \phi^\star(x)[pt(i)], a)}{T_i(\phi_i^\star(\check{x}) \mid \phi^\star(x)[pt(i)], a) + \rho(\phi_i^\star(\check{x}))}.$$

$\square$

**Theorem 4** (Decoder Regression Guarantees). *For any given $\delta_{abs} \in (0, 1)$ and $n_{abs} \in \mathbb{N}$ we have the following with probability at least $1 - \delta_{abs}$:*

$$\mathbb{E}_{x, a, \check{x} \sim D} \left[ (\hat{g}_i(x, a, \check{x}) - g^\star(x, a, \check{x}))^2 \right] \leq \Delta(n_{abs}, \delta_{abs}, |\mathcal{G}|),$$

*where $\Delta(n_{abs}, \delta_{abs}, |\mathcal{G}|) := \frac{c}{n_{abs}} \ln \frac{|\mathcal{G}|}{\delta_{abs}}$ and $c$ is a universal constant.*

*Proof.* This is a standard regression guarantee derived using Bernstein's inequality with realizability (3). For example, see Proposition 11 in Misra et al. (2020) for proof. $\square$

**Corollary 5.** *For any given $\delta_{abs} \in (0,1)$ and $n_{abs} \in \mathbb{N}$ we have the following with probability at least $1 - \delta_{abs}$:*

$$\mathbb{E}_{x,a,\check{x} \sim D} \left[ |\hat{g}_i(x,a,\check{x}) - g^\star(x,a,\check{x})| \right] \leq \sqrt{\Delta(n_{abs}, \delta_{abs}, |\mathcal{G}|)} \tag{17}$$

*Proof.* Applying Jensen's inequality ($\mathbb{E}[\sqrt{Z}] \leq \sqrt{\mathbb{E}[Z]}$) to Theorem 4 gives us:

$$\mathbb{E}_{x,a,\check{x} \sim D} \left[ |\hat{g}(x,a,\check{x}) - g^\star(x,a,\check{x})| \right] = \mathbb{E}_{x,a,\check{x} \sim D} \left[ \sqrt{|\hat{g}(x,a,\check{x}) - g^\star(x,a,\check{x})|^2} \right]$$

$$\leq \sqrt{\mathbb{E}_{x,a,\check{x} \sim D} \left[ (\hat{g}(x,a,\check{x}) - g^\star(x,a,\check{x}))^2 \right]}$$

$$\leq \sqrt{\Delta(n_{\text{abs}}, \delta_{\text{abs}}, |\mathcal{G}|)}.$$

$\square$

**Coupling Distribution**   We introduce a coupling distribution following Misra et al. (2020).

$$D_{\text{coup}}(x,a,\check{x}_1,\check{x}_2) = D(x,a)\rho(\check{x}_1)\rho(\check{x}_2). \tag{18}$$

We also define the following quantity which will be useful for stating our results:

$$\xi(\check{x}_1, \check{x}_2, x, a) = \frac{T(\check{x}_1 \mid x, a)}{\rho(\check{x}_1)} - \frac{T(\check{x}_2 \mid x, a)}{\rho(\check{x}_2)}. \tag{19}$$

**Lemma 10.** *For any fixed $\delta_{abs} \in (0,1)$ we have the following with probability at least $1 - \delta_{abs}$:*

$$\mathbb{E}_{x,a,\check{x}_1,\check{x}_2 \sim D_{coup}} \left[ \mathbf{1}\{\widehat{\phi}_i(\check{x}_1) = \widehat{\phi}_i(\check{x}_2)\} |\xi(\check{x}_1, \check{x}_2, x, a)| \right] \leq 8\sqrt{\Delta(n_{abs}, \delta_{abs}, |\mathcal{G}|)}.$$

*Proof.* We define a shorthand notation $\mathcal{E} = \mathbf{1}\{\widehat{\phi}_i(\check{x}_1) = \widehat{\phi}_i(\check{x}_2)\}$ for brevity. We also define a different coupled distribution $D'_{\text{coup}}$ given below:

$$D'_{\text{coup}}(x,a,\check{x}_1,\check{x}_2) = D(x,a)D(\check{x}_1 \mid x,a)D(\check{x}_2 \mid x,a) \tag{20}$$

where $D(\check{x} \mid x,a) = \frac{1}{2}\{T(\check{x} \mid x,a) + \rho(\check{x})\}$. It is easy to see that marginal distribution of $D'_{\text{coup}}$ over $x, a, \check{x}_1$ is same as $D(x, a, \check{x}_1)$.

We first use the definition of $\xi$ (Equation 19) and $g^\star$ (Equation 9) to express their relation:

$$|g^\star(x,a,\check{x}_1) - g^\star(x,a,\check{x}_2)| = \frac{\rho(\check{x}_1)\rho(\check{x}_2) |\xi(\check{x}_1, \check{x}_2, x, a)|}{(T(\check{x}_1 \mid x,a) + \rho(\check{x}_1))(T(\check{x}_1 \mid x,a) + \rho(\check{x}_2))}$$

$$= \frac{\rho(\check{x}_1)\rho(\check{x}_2)}{4D(\check{x}_1 \mid x,a)D(\check{x}_2 \mid, x, a)} |\xi(\check{x}_1, \check{x}_2, x, a)|. \tag{21}$$

The second line uses the definition of $D(\check{x} \mid x,a)$. We can view $\frac{\rho(\check{x}_1)}{D(\check{x}_1|x,a)}$ and $\frac{\rho(\check{x}_2)}{D(\check{x}_2|x,a)}$ as importance weight terms. Multiplying both sides by $\mathcal{E}$ and taking expectation with respect to $D'_{\text{coup}}$ then gives us:

$$\mathbb{E}_{D'_{\text{coup}}} \left[ \mathcal{E} |g^\star(x,a,\check{x}_1) - g^\star(x,a,\check{x}_2)| \right] = \frac{1}{4}\mathbb{E}_{D_{\text{coup}}} \left[ \mathcal{E}|\xi(\check{x}_1, \check{x}_2, x, a)| \right] \tag{22}$$

We bound the left hand side of Equation 22 as shown below:

$$\mathbb{E}_{D'_{\text{coup}}} \left[ \mathcal{E} |g^\star(x,a,\check{x}_1) - g^\star(x,a,\check{x}_2)| \right]$$
$$\leq \mathbb{E}_{D'_{\text{coup}}} \left[ \mathcal{E} |g^\star(x,a,\check{x}_1) - \hat{g}_i(x,a,\check{x}_1)| \right] + \mathbb{E}_{D'_{\text{coup}}} \left[ \mathcal{E} |\hat{g}_i(x,a,\check{x}_1) - g^\star(x,a,\check{x}_2)| \right]$$
$$= \mathbb{E}_{D'_{\text{coup}}} \left[ \mathcal{E} |g^\star(x,a,\check{x}_1) - \hat{g}_i(x,a,\check{x}_1)| \right] + \mathbb{E}_{D'_{\text{coup}}} \left[ \mathcal{E} |\hat{g}_i(x,a,\check{x}_2) - g^\star(x,a,\check{x}_2)| \right]$$
$$= 2\mathbb{E}_{D'_{\text{coup}}} \left[ \mathcal{E} |g^\star(x,a,\check{x}_1) - \hat{g}_i(x,a,\check{x}_1)| \right] = 2\mathbb{E}_D \left[ \mathcal{E} |g^\star(x,a,\check{x}) - \hat{g}(x,a,\check{x})| \right]$$
$$\leq 2\sqrt{\Delta(n_{\text{abs}}, \delta_{\text{abs}}, \mathcal{G})}$$

Here the first inequality follows from triangle inequality. The second step is key where we use $\hat{g}_i(x, a, \check{x}_1) = \hat{g}_i(x, a, \check{x}_2)$ whenever $\mathcal{E} = 1$. This itself follows from the bottleneck structure of $\mathcal{G}$ where $\hat{g}_i(x, a, \check{x}_i) = \hat{w}_i(x, a, \widehat{\phi}_i(\check{x}))$. The third step uses the symmetry of $\check{x}_1$ and $\check{x}_2$ in $D'_{\text{coup}}$ whereas the fourth step uses the fact that marginal distribution of $D'_{\text{coup}}$ is same as $D$. Lastly, final inequality uses $\mathcal{E} \leq 1$ and the result of Corollary 5. Combining the derived inequality with Equation 22 proves our result. $\qquad\square$

We define the quantity $\mathbb{P}(s'[i] = z \mid D') := \mathbb{E}_{(s,a)\sim D'}[\mathbf{1}\{s'[i] = z\}]$ for any distribution $D' \in \Delta(\mathcal{S}_{h-1} \times \mathcal{A})$. From the definition of $\rho$, we have $\rho(s'[i] = z) = \mathbb{P}(s'[i] = z \mid D)$. Intuitively, as we have policy cover at time step $h - 1$ and we take actions uniformly, therefore, we expect to have good lower bound on $\mathbb{P}(s'[i] = z \mid D)$ for every $i \in [d]$ and reachable $z \in \{0, 1\}$. Note that if $z = 0$ ($z = 1$) is not reachable then it means we always have $s_h[i] = 1$ ($s_h[i] = 0$) from our reachability assumption (see Section 2). We formally prove this next which will be useful later.

**Lemma 11.** *For any $z \in \{0, 1\}$ such that $s'[i] = z$ is reachable, we have:*

$$\rho(s'[i] = z) = \mathbb{P}(s'[i] = z \mid D) \geq \frac{\alpha\eta_{min}}{N|\mathcal{A}|}$$

*Proof.* Fix $z$ in $\{0, 1\}$. As $s'[i] = z$ is reachable, therefore, from the definition of $\eta_{min}$ we have:

$$\eta_{min} \leq \sup_{\pi\in\Pi} \mathbb{P}_\pi(s'[i] = z) \leq \sup_\pi \sum_{\check{s},a} \mathbb{P}_\pi(\check{s})T(s'[i] = z \mid \check{s}, a)$$

$$\leq \sum_{\check{s},a} \sup_{\pi\in\Pi} \mathbb{P}_\pi(\check{s})T(s'[i] \mid \check{s}, a) = \sum_{\check{s},a} \eta(\check{s})T(s'[i] \mid \check{s}, a)$$

We use the derived inequality to bound $\mathbb{P}(s'[i] = z \mid D)$ as shown:

$$\mathbb{P}(s'[i] = z \mid D) = \sum_{\check{s},a} \frac{\mu_{h-1}(\check{s})}{|\mathcal{A}|}T(s'[i] = z \mid \check{s}, a) \geq \frac{\alpha}{N|\mathcal{A}|} \sum_{\check{s},a} \eta(\check{s})T(s'[i] = z \mid \check{s}, a) \geq \frac{\alpha\eta_{min}}{N|\mathcal{A}|}.$$

The first inequality uses the fact that $\mu_{h-1}$ is created by roll-in with a uniformly selected policy in $\Psi_{h-1}$ which is an $\alpha$ policy cover. Recall that $N = |\Psi_{h-1}|$. The second inequality uses the derived result above. $\qquad\square$

**Lemma 12.** *For any $\check{x}_1, \check{x}_2$ such that $\phi_i^\star(\check{x}_1)$ and $\phi_i^\star(\check{x}_2)$ is reachable, we have:*

$$\mathbb{E}_{x,a\sim D}\left[|\xi(\check{x}_1, \check{x}_2, x, a)|\right] \geq \mathbf{1}\{\phi_i^\star(\check{x}_1) \neq \phi_i^\star(\check{x}_2)\}\frac{\alpha\eta_{min}\sigma}{2N}.$$

*Proof.* For any $x, a, \check{x}_1, \check{x}_2$ we can express $\xi$ (Equation 19) as:

$$|\xi(\check{x}_1, \check{x}_2, x, a)| = \left|\frac{T(\phi_i^\star(\check{x}_1) \mid \phi^\star(x)[pt(i)], a)}{\rho(\phi_i^\star(\check{x}_1))} - \frac{T(\phi_i^\star(\check{x}_2) \mid \phi^\star(x)[pt(i)], a)}{\rho(\phi^\star(\check{x}_2))}\right|$$

where we use the factorization assumption and decodability assumption. Note that we are implicitly assuming $\phi_i^\star(\check{x}_1)$ and $\phi_i^\star(\check{x}_2)$ are reachable, for the quantity $\xi(\check{x}_1, \check{x}_2, x, a)$ to be well defined.

We define $D_i$ to be the marginal distribution over $\mathcal{S}[pt(i)] \times \mathcal{A}$. Taking expectation on both side gives us:

$$\mathbb{E}_{x,a\sim D}\left[|\xi(\check{x}_1, \check{x}_2, x, a)|\right] = \mathbb{E}_{\check{s},a\sim D_i}\left[\left|\frac{T(\phi_i^\star(\check{x}_1) \mid \check{s}, a)}{\rho(\phi_i^\star(\check{x}_1))} - \frac{T(\phi_i^\star(\check{x}_2) \mid \check{s}, a)}{\rho(\phi_i^\star(\check{x}_2))}\right|\right]$$

$$= \sum_{\check{s},a} |\mathbb{P}_{D_i}(\check{s}, a \mid \phi_i^\star(\check{x}_1)) - \mathbb{P}_{D_i}(\check{s}, a \mid \phi_i^\star(\check{x}_2))|$$

$$= 2\left\|\mathbb{P}_{D_i}(., . \mid \phi_i^\star(\check{x}_1)) - \mathbb{P}_{D_i}(., . \mid \phi_i^\star(\check{x}_2))\right\|_{\text{TV}}$$

The second equality uses the definition of backward dynamics $\mathbb{P}_{D_i}$ over $\mathcal{S}[pt(i)] \times \mathcal{A}$ and the identity $\rho(s'[i]) = \mathbb{P}(s'[i] \mid D)$. If $\phi_i^\star(\check{x}_1) = \phi_i^\star(\check{x}_2)$ then the quantity on the right is 0. Otherwise, this quantity is given by $2\left\|\mathbb{P}_{D_i}(., . \mid s'[i] = 1) - \mathbb{P}_{D_i}(., . \mid s'[i] = 0)\right\|_{\text{TV}}$. In the later case, both

$s'[i] = 1$ and $s'[i] = 0$ configurations are reachable, and without loss of generality we can assume $\mathbb{P}(s'[i] = 0 \mid D) \geq 1/2$. Our goal is to bound this term using the margin assumption (Assumption 1). We do so using importance weight as shown below:

$$2 \left\| \mathbb{P}_{D_i}(., . \mid s'[i] = 1) - \mathbb{P}_{D_i}(., . \mid s'[i] = 0) \right\|_{\text{TV}}$$

$$= \sum_{\check{s}, a} \left| \mathbb{P}_{u_i}(\check{s}, a \mid s'[i] = 1) \frac{\mathbb{P}_{D_i}(\check{s}, a \mid s'[i] = 1)}{\mathbb{P}_{u_i}(\check{s}, a \mid s'[i] = 1)} - \mathbb{P}_{u_i}(\check{s}, a \mid s'[i] = 0) \frac{\mathbb{P}_{D_i}(\check{s}, a \mid s'[i] = 0)}{\mathbb{P}_{u_i}(\check{s}, a \mid s'[i] = 0)} \right|$$

$$= \sum_{\check{s}, a} \frac{D_i(\check{s}, a)}{u_i(\check{s}, a)} \left| \mathbb{P}_{u_i}(\check{s}, a \mid s'[i] = 1) \frac{\mathbb{P}(s'[i] = 1 \mid u_i)}{\mathbb{P}(s'[i] = 1 \mid D_i)} - \mathbb{P}_{u_i}(\check{s}, a \mid s'[i] = 0) \frac{\mathbb{P}(s'[i] = 0 \mid u_i)}{\mathbb{P}(s'[i] = 0 \mid D_i)} \right|$$

$$\geq \min_{\check{s}, a} \frac{D_i(\check{s}, a)}{u_i(\check{s}, a)} \frac{\mathbb{P}(s'[i] = 0 \mid D_i)}{\mathbb{P}(s'[i] = 0 \mid u_i)} \left\| P_{u_i}(., . \mid s'[i] = 1) - \mathbb{P}_{u_i}(., . \mid s'[i] = 0) \right\|_{\text{TV}}$$

$$\geq \min_{\check{s}, a} \frac{D_i(\check{s}, a)}{u_i(\check{s}, a)} \frac{\mathbb{P}(s'[i] = 0 \mid D_i)}{\mathbb{P}(s'[i] = 0 \mid u_i)} \sigma$$

The first step applies importance weight. As $u_i$ has support over all reachable configurations $\check{s}$ and actions $a \in \mathcal{A}$, hence, we can apply importance weight. The second step uses the definition of backward dynamics ($\mathbb{P}_D, \mathbb{P}_{u_i}$). The third step uses Lemma H.1 of Du et al. Du et al. (2019) (see Lemma 24 in Appendix E for statement). Finally, the last step uses Assumption 1. We bound the two multiplicative terms below:

We have $D(\check{s}, a) = \mu_{h-1}(\check{s}) \frac{1}{|\mathcal{A}|} \geq \frac{\alpha \eta_{min}}{N|\mathcal{A}|}$. The first equality uses the fact that actions are taken uniformly and second inequality uses the fact that $\mu_{h-1}$ is an $\alpha$-policy cover. As $u_i$ is the uniform distribution over $\mathcal{S}[pt(i)] \times \mathcal{A}$, therefore, we have $u_i(\check{s}, a) = \frac{1}{2^{|pt(i)|}|\mathcal{A}|}$. This gives us $\frac{D(\check{s}, a)}{u_i(\check{s}, a)} \geq \frac{\alpha \eta_{min}}{N} 2^{|pt(i)|}$. We bound the other multiplicative term as shown below:

$$\frac{\mathbb{P}(s'[i] = 0 \mid D_i)}{\mathbb{P}(s'[i] = 0 \mid u_i)} \geq \mathbb{P}(s'[i] = 0 \mid D_i) \geq \frac{1}{2}.$$

Combining the lower bounds for the two multiplicative terms and using $2^{|pt(i)|} \geq 1$ we get:

$$2 \left\| \mathbb{P}_{D_i}(., . \mid s'[i] = 1) - \mathbb{P}_{D_i}(., . \mid s'[i] = 0) \right\|_{\text{TV}} \geq \frac{\alpha \eta_{min} \sigma}{2N}. \tag{23}$$

Lastly, recall that our desired result is given by $2 \left\| \mathbb{P}_{D_i}(., . \mid s'[i] = 1) - \mathbb{P}_{D_i}(., . \mid s'[i] = 0) \right\|_{\text{TV}}$ whenever $\phi_i^\star(\tilde{x}_1) \neq \phi_i^\star(\tilde{x}_2)$ and 0 otherwise. Therefore, using the derived lower bound multiplied by $\mathbf{1}\{\phi_i^\star(\tilde{x}_1) \neq \phi_i^\star(\tilde{x}_2)\}$ gives us the desired result. □

**Corollary 6.** *We have the following with probability at least* $1 - \delta_{abs}$:

$$\mathbb{E}_{\tilde{x}_1, \tilde{x}_2 \sim \rho} \left[ \mathbf{1}\{\widehat{\phi}_i(\tilde{x}_1) = \widehat{\phi}_i(\tilde{x}_2)\} \mathbf{1}\{\phi_i^\star(\tilde{x}_1) \neq \phi_i^\star(\tilde{x}_2)\} \right] \leq \frac{16N}{\alpha \eta_{min} \sigma} \sqrt{\Delta(n_{abs}, \delta_{abs}, |\mathcal{G}|)}.$$

*Proof.* The proof trivially follows from applying the bound in Lemma 12 to Lemma 10 as shown below:

$$\mathbb{E}_{x, a, \tilde{x}_1, \tilde{x}_2 \sim D_{\text{coup}}} \left[ \mathbf{1}\{\widehat{\phi}_i(\tilde{x}_1) = \widehat{\phi}_i(\tilde{x}_2)\} |\xi(\tilde{x}_1, \check{x}_2, x, a)| \right]$$

$$= \mathbb{E}_{\tilde{x}_1, \tilde{x}_2 \sim \rho} \left[ \mathbf{1}\{\widehat{\phi}_i(\tilde{x}_1) = \widehat{\phi}_i(\tilde{x}_2)\} \mathbb{E}_{x, a, \sim D} \left[ |\xi(\tilde{x}_1, \check{x}_2, x, a)| \right] \right]$$

$$\geq \frac{\alpha \eta_{min} \sigma}{2N} \mathbb{E}_{\tilde{x}_1, \tilde{x}_2 \sim \rho} \left[ \left[ \mathbf{1}\{\widehat{\phi}_i(\tilde{x}_1) = \widehat{\phi}_i(\tilde{x}_2)\} \mathbf{1}\{\phi_i^\star(\tilde{x}_1) \neq \phi_i^\star(\tilde{x}_2)\} \right] \right]$$

The inequality here uses Lemma 12. The left hand side is bounded by $8\sqrt{\Delta(n_{\text{abs}}, \delta_{\text{abs}}, |\mathcal{G}|)}$ using Lemma 10. Combining the two bounds and rearranging the terms proves the result. □

At this point we analyze the two cases separately. In the first case, $s'[i]$ can be set to both 0 and 1. In the second case, $s'[i]$ can only be set to one of the values and we will call $s'[i]$ as *degenerate* at time step $h$. We will show how we can detect the second case, at which point we just output a decoder that always outputs 0. We analyze the first case below.

### D.2.1 CASE A: WHEN $s'[i]$ CAN TAKE ALL VALUES

Corollary 6 allows us to define a correspondence between learned state (i.e., output of $\widehat{\phi}_i$) and the actual state (i.e., output of $\phi_i^\star$). We show this correspondence in the result.

**Theorem 7** (Correspondence Theorem). *For any state factor $s'[i]$, there exists exists $\hat{u}_0 \in \{0,1\}$ and $\hat{u}_1 = 1 - \hat{u}_0$ with probability at least $1 - \delta_{abs}$ such that:*

$$\mathbb{P}(\widehat{\phi}_i(\check{x}) = \hat{u}_0 \mid s'[i] = 0) \geq 1 - \varrho,$$
$$\mathbb{P}(\widehat{\phi}_i(\check{x}) = \hat{u}_1 \mid s'[i] = 1) \geq 1 - \varrho,$$

*where* $\varrho := \frac{16N^2|\mathcal{A}|}{\alpha^2 \eta_{min}^2 \sigma} \sqrt{\Delta(n_{abs}, \delta_{abs}, |\mathcal{G}|)}$ *and* $\check{x} \sim \rho$*, provided* $\varrho \in (0, \frac{1}{2})$.

*Proof.* For any $u, z \in \{0, 1\}$ we define the following quantities:

$$P_z := \mathbb{E}_{\check{x} \sim \rho}[\mathbf{1}\{\phi_i^\star(\check{x}) = z\}], \qquad P_{uz} := \mathbb{E}_{\check{x} \sim \rho}[\mathbf{1}\{\widehat{\phi}_i(\check{x}) = u\}\mathbf{1}\{\phi_i^\star(\check{x}) = z\}].$$

It is easy to see that these quantities are related by: $P_z = P_{uz} + P_{(1-u)z}$. We define $\hat{u}_0 = \arg\max_{u \in \{0,1\}} P_{u0}$ and $\hat{u}_1 = 1 - \hat{u}_0$. This can be viewed as the learned bit value which is in most correspondence with $s[i] = 0$. We will derive lower bound on $P_{\hat{u}_0 0}/P_0$ and $P_{\hat{u}_1 1}/P_1$ which gives us the desired result. We first derive the following lower bound on $P_{\hat{u}_0 0}$:

$$P_{\hat{u}_0 0} \geq \frac{P_{\hat{u}_0 0} + P_{\hat{u}_1 0}}{2} \geq \frac{P_0}{2}, \tag{24}$$

where we use the fact that max is greater than average. Further, for any $u, z \in \{0, 1\}$ we have:

$$\mathbb{E}_{\check{x}_2, \check{x}_2 \sim \rho}\left[\mathbf{1}\{\widehat{\phi}_i(\check{x}_1) = \widehat{\phi}_i(\check{x}_2)\}\mathbf{1}\{\phi_i^\star(\check{x}_1) \neq \phi_i^\star(\check{x}_2)\}\right]$$
$$\geq \mathbb{E}_{\check{x}_1, \check{x}_2 \sim \rho}\left[\mathbf{1}\{\widehat{\phi}_i(\check{x}_1) = u\}\mathbf{1}\{\widehat{\phi}_i(\check{x}_2) = u\}\mathbf{1}\{\phi_i^\star(\check{x}_1) = z\}\mathbf{1}\{\phi_i^\star(\check{x}_2) = 1 - z\}\right]$$
$$= P_{uz}P_{u(1-z)}$$

We define a shorthand notation $\Delta' := \frac{16N}{\alpha \eta_{min} \sigma} \sqrt{\Delta(n_{abs}, \delta_{abs}, |\mathcal{G}|)}$. Then from Corollary 6 we have proven that $P_{uz}P_{u(1-z)} \leq \Delta'$ for any $u, z \in \{0, 1\}$. This allows us to write:

$$P_{\hat{u}_1 1} = P_1 - P_{\hat{u}_0 1} \geq P_1 - \frac{\Delta'}{P_{\hat{u}_0 0}} \quad \Rightarrow \quad \frac{P_{\hat{u}_1 1}}{P_1} \geq 1 - \frac{\Delta'}{P_1 P_{\hat{u}_0 0}} \geq 1 - \frac{2\Delta'}{P_0 P_1}$$

where the last inequality uses Equation 24. We will derive the same result for $P_{\hat{u}_0 0}/P_0$.

$$P_{\hat{u}_0 0} = P_0 - P_{\hat{u}_1 0} \geq P_0 - \frac{\Delta'}{P_{\hat{u}_1 1}} \quad \Rightarrow \quad \frac{P_{\hat{u}_0 0}}{P_0} \geq 1 - \frac{\Delta'}{P_0 P_{\hat{u}_1 1}} \geq 1 - \frac{\Delta'}{P_0 P_1 - 2\Delta'},$$

where the last inequality uses derived bound for $P_{\hat{u}_1 1}/P_1$. If we assume $\Delta' \leq \frac{P_0 P_1}{4}$ then we get $\frac{P_{\hat{u}_0 0}}{P_0} \geq 1 - \frac{2\Delta'}{P_0 P_1}$.

As $P_0 + P_1 = 1$, therefore, we get $P_0 P_1 = P_0 - P_0^2 = P_1 - P_1^2$. If $P_0 \leq \frac{1}{2}$ then $P_0 - P_0^2 \geq \frac{P_0}{2}$. Otherwise, $P_0 > \frac{1}{2}$ which implies $P_1 \leq \frac{1}{2}$ and $P_1 - P_1^2 \geq \frac{P_1}{2}$. This gives us $P_0 P_1 \geq \min\{\frac{P_0}{2}, \frac{P_1}{2}\}$. Using lower bounds for $P_0$ and $P_1$ from Lemma 11 gives us $P_0 P_1 \geq \frac{\alpha \eta_{min}}{2N|\mathcal{A}|}$, and allows us to write:

$$\frac{P_{\hat{u}_1 1}}{P_1} \geq 1 - \frac{4N|\mathcal{A}|\Delta'}{\alpha \eta_{min}}, \qquad \frac{P_{\hat{u}_0 0}}{P_0} \geq 1 - \frac{4N|\mathcal{A}|\Delta'}{\alpha \eta_{min}}.$$

It is easy to verify that $\varrho = \frac{4N|\mathcal{A}|\Delta'}{\alpha \eta_{min}}$. As $\frac{P_{\hat{u}_0 0}}{P_0} = \mathbb{P}(\widehat{\phi}_i(x') = \hat{u}_0 \mid s[i] = 0)$ and $\frac{P_{\hat{u}_1 1}}{P_1} = \mathbb{P}(\widehat{\phi}_i(x') = \hat{u}_1 \mid s[i] = 1)$, therefore, we prove our result. The only requirement we used is that $\Delta' \leq \frac{P_0 P_1}{4}$ which is ensured if $\varrho \in (0, \frac{1}{2})$. □

### D.2.2 Case B: when $s'[i]$ takes a single value

We want to be able to detect this case with high probability so that we can learn a degenerate decoder that only takes value 0. This would trivially give us a correspondence result similar to Theorem 7.

We describe the general form of the `LearnDecoder` subroutine in Algorithm 6. The key difference from the case we covered earlier is line 6-10. For a given factor $i$, we first learn the model $\hat{g}_i$ containing the decoder $\widehat{\phi}_i$, as before using noise contrastive learning. We then sample $n_{\text{deg}}$ iid triplets $\mathcal{D}_{deg} = \{(x_j, a_j, \check{x}_j)\}_{j=1}^{n_{\text{deg}}}$ where $(x_j, a_j) \sim D$ and $\check{x}_j \sim \rho$ (line 6-8). Recall $\check{x} = x[ch_h(i)]$. Next, we compute the *width* of prediction values over $\mathcal{D}_{deg}$ as defined below:

$$\max_{j,k \in [n_{\text{deg}}]} |\hat{g}(x_j, a_j, \check{x}_j) - \hat{g}(x_k, a_k, \check{x}_k)| \tag{25}$$

If the width is smaller than a certain value then we determine the factor to be degenerate and output a degenerate decoder $\widehat{\phi}_i := 0$, otherwise, we stick to the decoder learned by our regressor task. The form of sample size $n_{\text{deg}}$ will become clear at the end of analysis, and we will determine the reason for the choice of threshold for width in line 9. Intuitively, if the latent factor only takes one value then the optimal classifier will always output $1/2$ and so our prediction values should be close to one another. However, if the latent factor takes two values then the model prediction should be distinct.

---

**Algorithm 6** `LearnDecoder`$(\mathcal{G}, \Psi_{h-1}, \widehat{ch}_h)$. $\qquad\qquad$ Child function has type $\widehat{ch}_h : [d_h] \to 2^{[m]}$

---

1: **for** $i$ in $[d_h]$, define $\omega = \widehat{ch}_h(i), \mathcal{D} = \emptyset, \mathcal{D}_{deg} = \emptyset$ **do**

2: $\quad$ **for** $n_{\text{abs}}$ times **do** $\qquad$ // collect a dataset of real $(y=1)$ and imposter $(y=0)$ transitions

3: $\qquad$ Sample $(x^{(1)}, a^{(1)}, x'^{(1)}), (x^{(2)}, a^{(2)}, x'^{(2)}) \sim \text{Unf}(\Psi_{h-1}) \circ \text{Unf}(\mathcal{A})$ and $y \sim \text{Bern}(\frac{1}{2})$

4: $\qquad$ **If** $y = 1$ **then** $\mathcal{D} \leftarrow \mathcal{D} \cup (x^{(1)}, a^{(1)}, x'^{(1)}[\omega], y)$ **else** $\mathcal{D} \leftarrow \mathcal{D} \cup (x^{(1)}, a^{(1)}, x'^{(2)}[\omega], y)$

5: $\quad$ $\hat{g}_i := \widehat{u}_i, \widehat{\phi}_i = \text{REG}(\mathcal{D}, \mathcal{G})$ $\qquad$ // train the decoder using noise-contrastive learning

6: $\quad$ **for** $n_{\text{deg}}$ times **do** $\qquad\qquad\qquad\qquad\qquad\qquad\qquad$ // detect degenerate factors

7: $\qquad$ Sample $(x^{(1)}, a^{(1)}, x'^{(1)}), (x^{(2)}, a^{(2)}, x'^{(2)}) \sim \text{Unf}(\Psi_{h-1}) \circ \text{Unf}(\mathcal{A})$

8: $\qquad$ $\mathcal{D}_{deg} \leftarrow \mathcal{D}_{deg} \cup \{(x^{(1)}, a^{(1)}, x'^{(2)})\}$

9: $\quad$ **if** $\max_{j,k \in [n_{\text{deg}}]} |\hat{g}_i(x_j, a_j, x'_j[\omega]) - \hat{g}_i(x_k, a_k, x'_k[\omega])| \le \frac{\alpha^2 n_{min}^2 \sigma}{40 |\Psi_{h-1}|^2 |\mathcal{A}|}$ **then** // max over $\mathcal{D}_{deg}$

10: $\qquad$ $\widehat{\phi}_i := 0$ $\qquad\qquad\qquad\qquad\qquad\qquad$ // output a decoder that always returns 0

$\quad$ **return** $\widehat{\phi} : \mathcal{X} \to \{0,1\}^{d_h}$ where for any $x \in \mathcal{X}$ and $i \in [d_h]$ we have $\widehat{\phi}(x)[i] = \widehat{\phi}_i(x[\widehat{ch}_h(i)])$.

---

For convenience, we define $D'(x, a, \check{x}) = D(x, a)\rho(\check{x})$, and so $(x_j, a_j, \check{x}_j) \sim D'$. For brevity reasons, we do not add additional qualifiers to differentiate $x_j, a_j, \check{x}_j$ from the dataset of real and imposter transitions, we used in the previous section for the regression task. In this part alone, we will use $x_j, a_j, \check{x}_j$ to refer to the transitions collected for the purpose of detecting degenerate factors.

**Lemma 13** (Markov Bound). *Let $\{(x_j, a_j, \check{x}_j)\}_{j=1}^{n_{deg}}$ be a dataset of iid transitions sampled from $D'$. Fix $a > 0$. Then with probability at least $1 - \delta_{abs} - \frac{2n_{deg}\sqrt{\Delta(n_{abs}, \delta_{abs}, |\mathcal{G}|)}}{a}$ we have:*

$$\forall j \in [n_{deg}], \qquad |\hat{g}(x_j, a_j, \check{x}_j) - g^\star(x_j, a_j, \check{x}_j| \le a.$$

*Proof.* It is straightforward to verify that for any $(x_j, a_j, \check{x}_j)$ we have $D(x_j, a_j, \check{x}_j) \ge {}^{D'(x_j, a_j, \check{x}_j)}/_2$. Using Corollary 5 we get:

$$\mathbb{E}_{x,a,\check{x} \sim D'} [|\hat{g}(x, a, \check{x}) - g^\star(x, a, \check{x})|] \le 2\sqrt{\Delta(n_{\text{abs}}, \delta_{\text{abs}}, |\mathcal{G}|)}$$

Let $\mathcal{E}_j$ denote the event $\{|\hat{g}(x_j, a_j, \check{x}_j) - g^\star(x_j, a_j, \check{x}_j)| \le a\}$ and $\overline{\mathcal{E}_j}$ be its negation, then:

$$\mathbb{P}(\cap_{j=1}^{n_{\text{deg}}} \mathcal{E}_j) \ge 1 - \sum_{j=1}^{n_{\text{deg}}} \mathbb{P}(\overline{\mathcal{E}_j}) \ge 1 - \frac{2n_{\text{deg}}\sqrt{\Delta(n_{\text{abs}}, \delta_{\text{abs}}, |\mathcal{G}|)}}{a},$$

where the first inequality uses union bound and the second inequality uses Markov's inequality. As Corollary 5 holds with probability $\delta_{\text{abs}}$, our overall failure probability is at most $\delta_{\text{abs}} + \frac{2n_{\text{deg}}\sqrt{\Delta(n_{\text{abs}}, \delta_{\text{abs}}, |\mathcal{G}|)}}{a}$. $\qquad\square$

**Lemma 14.** *For any reachable parent factor values $\check{s}$, action $a \in \mathcal{A}$ and reachable $s'[i] \in \{0,1\}$, we have $D'(\check{s}, a, s'[i]) \geq \frac{\alpha^2 \eta_{min}^2}{N^2 |\mathcal{A}|^2}$.*

*Proof.* We have $D'(\check{s}, a, s'[i]) = \frac{\mu_{h-1}(\check{s})}{|\mathcal{A}|} \rho(s'[i]) \geq \frac{\alpha^2 \eta_{min}^2}{N^2 |\mathcal{A}|^2}$, where used the induction hypothesis IH.4 that $\Psi$ is an $\alpha$-policy cover of $\mathcal{S}_{h-1}$ and Lemma 11. $\qquad\square$

**Lemma 15** (Degenerate Factors). *Fix $a > 0$. If $s'[i]$ only takes a single value then with probability at least $1 - \delta_{abs} - \frac{2 n_{deg} \sqrt{\Delta(n_{abs}, \delta_{abs}, |\mathcal{G}|)}}{a}$ we have:*

$$\max_{j,k \in [n_{deg}]} |\hat{g}(x_j, a_j, \check{x}_j) - \hat{g}(x_k, a_k, \check{x}_k)| \leq 2a$$

*Proof.* When $s'[i]$ takes a single value then $g^\star$ is the constant function $\frac{1}{2}$. For any $j$ and $k$ we get the following using Lemma 13 and triangle inequality.

$$|\hat{g}(x_j, a_j, \check{x}_j) - \hat{g}(x_k, a_k, \check{x}_k)| \leq |\hat{g}(x_j, a_j, \check{x}_j) - g^\star(x_j, a_j, \check{x}_j)| +$$
$$|g^\star(x_k, a_k, \check{x}_k) - \hat{g}(x_k, a_k, \check{x}_k)| \leq 2a.$$

$\square$

**Lemma 16** (Non Degenerate Factors). *Fix $a > 0$ and assume $n_{deg} \geq \frac{N^2 |\mathcal{A}|^2}{\alpha^2 \eta_{min}^2}$, then we have:*

$$\max_{j,k \in [n_{deg}]} |\hat{g}(x_j, a_j, \check{x}_j) - \hat{g}(x_k, a_k, \check{x}_k)| \geq \frac{\alpha^2 \eta_{min}^2 \sigma}{16 N^2 |\mathcal{A}|} - 2a$$

*with probability at least $1 - \delta_{abs} - \frac{2 n_{deg} \sqrt{\Delta(n_{abs}, \delta_{abs}, |\mathcal{G}|)}}{a} - 4 \exp\left(-\frac{\alpha^2 \eta_{min}^2 n_{deg}}{3 N^2 |\mathcal{A}|^2}\right)$.*

*Proof.* Equation 23 implies that there exists $\check{s}, a$ such that

$$\left| \frac{T(s'[i] = 1 \mid \check{s}, a)}{\rho(s'[i] = 1)} - \frac{T(s'[i] = 0 \mid \check{s}, a)}{\rho(s'[i] = 0)} \right| \geq \frac{\alpha \eta_{min} \sigma}{2N}$$

Combining this with Equation 21 we get:

$$|g^\star(\check{s}, a, s'[i] = 1) - g^\star(\check{s}, a, s'[i] = 0)| \geq \frac{\rho(s'[i] = 1)\rho(s'[i] = 0)}{4 D(s'[i] = 1 \mid \check{s}, a) D(s'[i] = 0 \mid \check{s}, a)} \frac{\alpha \eta_{min} \sigma}{2N}$$
$$\geq \frac{\alpha^2 \eta_{min}^2 \sigma}{16 N^2 |\mathcal{A}|} \qquad (26)$$

where Equation 26 uses $\rho(s'[i] = 1)\rho(s'[i] = 0) \geq \frac{\alpha \eta_{min}}{2N|\mathcal{A}|}$, as one of the terms is at least $1/2$ and other can be bounded using Lemma 11.

Say we have two examples in our dataset, say $\{(x_1, a_1, \check{x}_1), (x_2, a_2, \check{x}_2)\}$ without loss of generality, such that $\phi^\star(x_1)[\omega] = \phi^\star(x_2)[\omega] = \check{s}$, action $a_1 = a_2 = a$, $\phi_i^\star(\check{x}_1) = 1$, and $\phi_i^\star(\check{x}_2) = 0$. Then we have:

$$\max_{j,k \in [n_{deg}]} |\hat{g}(x_j, a_j, \check{x}_j) - \hat{g}(x_k, a_k, \check{x}_k)| \geq |\hat{g}(x_1, a_1, \check{x}_1) - \hat{g}(x_2, a_2, \check{x}_2)|$$
$$\geq |g^\star(\check{s}, a, 1) - g^\star(\check{s}, a, 0)|$$
$$- |\hat{g}(x_1, a_1, \check{x}_1) - g^\star(x_1, a_1, \check{x}_1)|$$
$$- |\hat{g}(x_2, a_2, \check{x}_2) - g^\star(x_2, a_2, \check{x}_2)|$$
$$\geq \frac{\alpha^2 \eta_{min}^2 \sigma}{16 N^2 |\mathcal{A}|} - 2a \quad \text{(using Equation 26 and Lemma 13)}$$

We use Lemma 13 which has a failure probability of $\delta_{abs} + \frac{2 n_{deg} \sqrt{\Delta(n_{abs}, \delta_{abs}, |\mathcal{G}|)}}{a}$. Further, we also assume that our dataset contains both $(\check{s}, a, s'[i] = 1)$ and $(\check{s}, a, s'[i] = 0)$. Probability of one of these

events is given by Lemma 14. Therefore, if $n_{\text{deg}} \geq \frac{N^2|\mathcal{A}|^2}{\alpha^2 \eta_{min}^2}$ then from Lemma 25 and union bound, the probability that at least one of these transitions does not occur is given by $4\exp\left(-\frac{\alpha^2 \eta_{min}^2 n_{\text{deg}}}{3N^2|\mathcal{A}|^2}\right)$.

The total failure probability is given by union bound and computes to:

$$\delta_{\text{abs}} + \frac{2n_{\text{deg}}\sqrt{\Delta(n_{\text{abs}}, \delta_{\text{abs}}, |\mathcal{G}|)}}{a} + 4\exp\left(-\frac{\alpha^2 \eta_{min}^2 n_{\text{deg}}}{3N^2|\mathcal{A}|^2}\right).$$

$\square$

If we fix $a = \frac{\alpha^2 \eta_{min}^2 \sigma}{80N^2|\mathcal{A}|}$ then in the two case we have:

$$\text{(Degenerate Factor)} \qquad \max_{j,k \in [n_{\text{deg}}]} |\hat{g}(x_j, a_j, \check{x}_j) - \hat{g}(x_k, a_k, \check{x}_k)| \leq \frac{\alpha^2 \eta_{min}^2 \sigma}{40N^2|\mathcal{A}|}$$

$$\text{(Non-Degenerate Factor)} \qquad \max_{j,k \in [n_{\text{deg}}]} |\hat{g}(x_j, a_j, \check{x}_j) - \hat{g}(x_k, a_k, \check{x}_k)| \geq \frac{3\alpha^2 \eta_{min}^2 \sigma}{80N^2|\mathcal{A}|}$$

**Theorem 8** (Detecting Degenerate Case). *We correctly predict if $s'[i]$ is a degenerate factor or not when using $n_{deg} = \frac{3N^2|\mathcal{A}|^2}{\alpha^2 \eta_{min}^2} \log\left(\frac{4}{\delta_{abs}}\right)$ and $\varrho \leq \frac{\alpha^2 \eta_{min}^2 \delta_{abs}}{30N^2|\mathcal{A}|^2} \log^{-1}\left(\frac{4}{\delta_{abs}}\right)$, with probability at least $1 - 3\delta_{abs}$.*

*Proof.* The result follows by combining Lemma 16 and Lemma 15, and using the value of $a$ described above. These two results hold with probability at least:

$$1 - \delta_{\text{abs}} - \frac{2n_{\text{deg}}\sqrt{\Delta(n_{\text{abs}}, \delta_{\text{abs}}, |\mathcal{G}|)}}{a} - 4\exp\left(-\frac{\alpha^2 \eta_{min}^2 n_{\text{deg}}}{3N^2|\mathcal{A}|^2}\right)$$

Setting the hyperparameters to satisfy the following:

$$n_{\text{deg}} = \frac{3N^2|\mathcal{A}|^2}{\alpha^2 \eta_{min}^2} \log\left(\frac{4}{\delta_{\text{abs}}}\right), \qquad \Delta(n_{\text{abs}}, \delta_{\text{abs}}, |\mathcal{G}|)^{-1/2} \geq \frac{480N^4|\mathcal{A}|^3}{\alpha^4 \eta_{min}^4 \delta_{\text{abs}} \sigma} \log\left(\frac{4}{\delta_{\text{abs}}}\right),$$

gives a failure probability of at most $3\delta_{\text{abs}}$. The later condition can be expressed in terms of $\varrho$ which gives us the desired bounds (see Theorem 7 for definition of $\varrho$). Lastly, note that setting $n_{\text{deg}}$ this way also satisfies the requirement in Lemma 16. Lastly, note that the resultant bound on $\varrho$ is much stronger than required for Theorem 7. Therefore, we can significantly improve the complexity bounds in the setting where there are no degenerate state factors. $\square$

### D.2.3 COMBINING CASE A AND CASE B

Theorem 8 shows that we can detect degenerate state factors with high probability. If we have a degenerate state factor and we detect it, then correspondence theorem holds trivially. However, if we don't have degeneracy and we correctly predict it, then we stick our learned decoder and Theorem 7 holds true. These two results allows us to define a bijection between real states and learned states that we explain below.

**Bijective Mapping between real and learned states**    For a given time step $h$ and state bit $s[i] = z$, we will define $\hat{u}_{hiz}$ as the corresponding learned state bit. When $h$ and $i$ will be clear from the context then we will express this as $\hat{u}_z$. We will use the notation $\hat{s}$ to denote a learned state at time step $h-1$ and $\hat{s}'$ to denote learned state at time step $h$. Let $pt(i) = (i_1, \cdots, i_l)$ and $s[pt(i)] = w := (w_1, w_2, \cdots w_l)$, then we define $\hat{w} = (\hat{u}_{(h-1)i_1 w_1} \cdots \hat{u}_{(h-1)i_l w_l})$ as the learned state bits corresponding to $w$. More generally, for a given set $\mathcal{K} \in 2^d$, we denote the real state factors as $s[\mathcal{K}]$ (or $s'[\mathcal{K}]$) and the corresponding learned state factors as $\hat{s}[\mathcal{K}]$ (or $\hat{s}'[\mathcal{K}]$).

We define a mapping $\theta_h : \{0,1\}^d \rightarrow \{0,1\}^d$ from learned state to real state. We will drop the subscript $h$ when the time step is clear from the context. We denote the domain of $\theta_h$ by $\widehat{\mathcal{S}}_h$ which is a subset of $\{0,1\}^d$. Note that every real state may not be reachable at time step $h$. E.g., maybe our decoder outputs $\hat{s}' = (0,0)$ but that the corresponding real state is not reachable at time step $h$. Figure 3 visualizes the mapping.

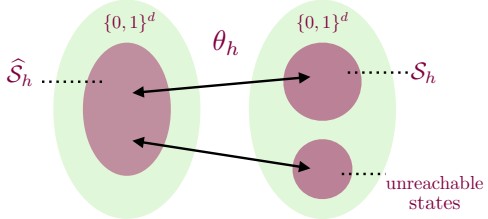

Figure 3: Bijection between learned state space $\widehat{\mathcal{S}}_h$ and the state space at time step $h$. The bijection maps every state in $\mathcal{S}_h$ to a unique state in $\widehat{\mathcal{S}}_h$. However $\widehat{\mathcal{S}}_h$ also contains learned states that do not correspond to a reachable state at this time step. This happens due to error in the decoder.

For a learned state $\hat{s}$ we have $s = \theta(\hat{s})$ if $s = (z_1, \cdots, z_d)$ and $\hat{s} = (u_{h1z_1}, \cdots, u_{hdz_d})$. We would also overload our notation to write $w = \theta(\hat{w})$ for a given $\hat{s}[\mathcal{K}] = \hat{w}$ where $w = \theta(\hat{s})[\mathcal{K}]$, whenever factor set $\mathcal{K}$ is clear from the context.

We call $s'$ (or $s$) as *reachable* if $s' \in \mathcal{S}_h$ (or $s \in \mathcal{S}_{h-1}$). Similarly, we call $\hat{s}'$ (or $\hat{s}$) as reachable if $\theta_h(\hat{s}')$ (or $\theta_{h-1}(\hat{s})$) are reachable. For a given set of factors $\mathcal{K}$, we call $w \in \{0,1\}^{|\mathcal{K}|}$ as *reachable for* $\mathcal{K}$ if there exists a reachable state with factors $\mathcal{K}$ taking on value $w$. Similarly, we define $\hat{w} \in \{0,1\}^{|\mathcal{K}|}$ as reachable for a given $\mathcal{K}$ if $\theta(\hat{w})$ is reachable for $\mathcal{K}$.

We use the mapping to state the correspondence theorem for the whole state.

**Corollary 9** (General Case). *If* $n_{deg} = \frac{3N^2|\mathcal{A}|^2}{\alpha^2\eta_{min}^2} \log\left(\frac{4}{\delta_{abs}}\right)$ *and* $\varrho \leq \frac{\alpha^2\eta_{min}^2\delta_{abs}}{30N^2|\mathcal{A}|^2} \log^{-1}\left(\frac{4}{\delta_{abs}}\right)$ *holds, then with probability at least* $1 - 3d\delta_{abs}$, *we have:*

$$\forall s' \in \mathcal{S}_h, \qquad \mathbb{P}(\hat{s}'[i] = \theta^{-1}(s')[i] \mid s'[i]) \geq 1 - \varrho, \qquad \mathbb{P}(\hat{s}' = \theta^{-1}(s') \mid s') \geq 1 - d\varrho.$$

*Proof.* The first result directly follows from being able to detect if we are in degenerate setting or not and if not, then applying Theorem 7, and if yes then result holds trivially. This holds with probability at least $1 - 3\delta_{abs}$ from Theorem 8. Applying union bound over all $d$ learned factors gives us success probability across all factors of at least $1 - 3d\delta_{abs}$. The second one follows from union bound.

$$\mathbb{P}(\hat{s} \neq \theta^{-1}(s) \mid s) = \mathbb{P}(\exists i : \hat{s}[i] \neq \theta^{-1}(s)[i] \mid s[i]) \leq \sum_{i=1}^{d} \mathbb{P}(\hat{s}[i] \neq \theta^{-1}(s)[i] \mid s[i]) \leq d\varrho.$$

$\square$

As we noticed before, our bounds can be significantly improved in the case of no degenerate factors. This prevents application of expensive Markov inequality. Therefore, we also state bounds for the special case below.

**Corollary 10** (Degenerate Factors Absent). *If* $\varrho \in \left(0, \frac{1}{2}\right)$, *then with probability at least* $1 - d\delta_{abs}$, *we have:*

$$\forall s' \in \mathcal{S}_h, \qquad \mathbb{P}(\hat{s}'[i] = \theta^{-1}(s')[i] \mid s'[i]) \geq 1 - \varrho, \qquad \mathbb{P}(\hat{s}' = \theta^{-1}(s') \mid s') \geq 1 - d\varrho.$$

*Proof.* Same as Corollary 9 except we can directly apply Theorem 7 as we don't need to do any expensive check for a degenerate factor. $\square$

### D.3 MODEL ESTIMATION

Our next goal is to estimate a model $\widehat{T}_h : \widehat{\mathcal{S}}_{h-1} \times \mathcal{A} \to \Delta(\widehat{\mathcal{S}}_h)$ and latent parent structure $\widehat{pt}_h$, given roll-in distribution $D \in \Delta(\mathcal{S}_{h-1})$, and the learned decoders $\{\widehat{\phi}_t\}_{t \leq h}$. Recall that our approach estimates the model by count-based estimation. Let $\mathcal{D} = \{(x^{(k)}, a^{(k)}, x'^{(k)})\}_{k=1}^{n_{est}}$ be the sample collected for model estimation.

Recall that we estimate $\widehat{pt}_h(i)$ be using a set of learned factors $\mathcal{I}$ that we believe is $\widehat{pt}_h(i)$ and varying a disjoint set of learned factors $\mathcal{J}$. If $\widehat{pt}(i) \subseteq \mathcal{I}$ then we expect the learned model to behave the same

irrespective of how we vary the factors $\mathcal{J}$. However, if $\widehat{pt}(i) \not\subseteq \mathcal{I}$ then there exists a parent factor that on varying will different values for the learned dynamics. Let $\mathcal{K} = \mathcal{I} \cup \mathcal{J}$ be the set of all factors in the control group ($\mathcal{I}$) and variable group ($\mathcal{J}$).

We will first analyze the case for a fixed $i \in \widehat{\mathcal{S}}_h$ and control group $\mathcal{I}$ and variable group $\mathcal{J}$. For a given $\hat{v} \in \{0,1\}$, $\hat{w} \in \{0,1\}^{|\mathcal{K}|}$ and $a \in \mathcal{A}$, we have $\widehat{\mathbb{P}}_D(\hat{s}'[i] = \hat{v} \mid \hat{s}[\mathcal{K}] = \hat{w}, a)$ denoting the estimate probability derived from our count based estimation (Algorithm 4, line 3). Let $\mathbb{P}_D(\hat{s}'[i] = \hat{v} \mid \hat{s}[\mathcal{K}] = \hat{w}, a)$ be the probabilities that are being estimated. It is important to note that we use subscript $D$ for these notations as the learned states $\hat{s}$ are not Markovian and $\mathcal{K}$ may not contain $pt(i)$, therefore, the estimated probabilities $\widehat{\mathbb{P}}$ and expected probabilities $\mathbb{P}$ will be dependent on the roll-in distribution $D$.

In order to estimate $\widehat{\mathbb{P}}_D(. \mid \hat{s}_{h-1}[\mathcal{K}] = \hat{w}, a)$ we want good lower bounds on $\mathbb{P}_D(\hat{s}_{h-1}[\mathcal{K}] = \hat{w}, a)$ for every $a \in \mathcal{A}$ and $\hat{w}$ reachable for $\mathcal{K}$. Our roll-in distribution $D$ only guarantees lower bound on $\mathbb{P}_D(s_{h-1}[\mathcal{K}], a)$. However, we can use IH.2 to bound the desired quantity below.

**Lemma 17** (Model Estimation Coverage). *If $\varrho \leq \frac{1}{2}$ then for all $\mathcal{K} \in \mathscr{C}_{\leq 2\kappa}([d])$, $a \in \mathcal{A}$ and $\hat{w} \in \{0,1\}^{|\mathcal{K}|}$ reachable for $\mathcal{K}$, we have:*

$$\mathbb{P}_D(\hat{s}_{h-1}[\mathcal{K}] = \hat{w}, a) \geq \frac{\alpha \eta_{min}}{4^\kappa N |\mathcal{A}|}$$

*Proof.* We can express $\mathbb{P}_D(\hat{s}_{h-1}[\mathcal{K}] = \hat{w}, a) = \frac{1}{|\mathcal{A}|}\mathbb{P}_D(\hat{s}_{h-1}[\mathcal{K}] = \hat{w})$ as actions are taken uniformly. Let $s_{h-1} = \theta_{h-1}(\hat{s}_{h-1})$ and $w = s_{h-1}[\mathcal{K}]$. We bound $\mathbb{P}_D(\hat{s}_{h-1}[\mathcal{K}] = \hat{w})$ as shown:

$$\begin{aligned}
\mathbb{P}_D(\hat{s}_{h-1}[\mathcal{K}] = \hat{w}) &\geq \mathbb{P}_D\left(\hat{s}_{h-1}[\mathcal{K}] = \hat{w}, s_{h-1}[\mathcal{K}] = w\right) \\
&= \mathbb{P}_D\left(\hat{s}_{h-1}[\mathcal{K}] = \hat{w} \mid s_{h-1}[\mathcal{K}] = w\right) \mathbb{P}_D\left(s_{h-1}[\mathcal{K}] = w\right) \\
&= \prod_{k \in \mathcal{K}} \mathbb{P}_D\left(\hat{s}_{h-1}[k] = \hat{w}_k \mid s_{h-1}[k] = w_k\right) \mathbb{P}_D\left(s_{h-1}[\mathcal{K}] = w\right) \\
&\geq (1 - \varrho)^{2\kappa} \frac{\alpha \eta_{min}}{N} \geq \frac{\alpha \eta_{min}}{4^\kappa N},
\end{aligned}$$

where the third step uses the fact that value of learned state $\hat{s}_{h-1}[k]$ is independent of other decoders given the real state bit $s_{h-1}[k]$. The fourth step uses IH.2 and the fact that we have good coverage over all sets of state factors of size at most $2\kappa$. Last inequality uses $|\mathcal{K}| \leq 2\kappa$ and $\varrho \leq \frac{1}{2}$. $\qquad\square$

We now show that our count-based estimator $\widehat{\mathbb{P}}_D$ converges to $\mathbb{P}_D$ and derive the rate of convergence.

**Lemma 18** (Model Estimation Error). *Fix $\delta_{est} \in (0,1)$. Then with probability at least $1 - \delta_{est}$ for every $\mathcal{K} \in \mathscr{C}_{\leq 2\kappa}([d])$, $\hat{w} \in \{0,1\}^{|\mathcal{K}|}$ reachable for $\mathcal{K}$, and $a \in \mathcal{A}$ we have the following:*

$$\sum_{\hat{v} \in \{0,1\}} \left| \widehat{\mathbb{P}}_D(\hat{s}_h[i] = \hat{v} \mid \hat{s}_{h-1}[\mathcal{K}] = \hat{w}, a) - \mathbb{P}_D(\hat{s}_h[i] = \hat{v} \mid \hat{s}_{h-1}[\mathcal{K}] = \hat{w}, a) \right| \leq 2\Delta_{est}(n_{est}, \delta_{est}),$$

*where* $\Delta_{est}(n_{est}, \delta_{est}) := \frac{1}{2}\left(\frac{2^{\kappa+5}N|\mathcal{A}|}{\alpha \eta_{min} n_{est}}\right)^{1/2} \ln\left(\frac{4e|\mathcal{A}|(ed)^{2\kappa}}{\delta_{est}}\right)$.

*Proof.* We sample $n_{est}$ samples by roll-in at time step $h-1$ with distribution $D$ and taking actions uniformly. We first analyze the failure probability for a given $\mathcal{K}, \hat{w}, a$. Let $\mathcal{E}(\mathcal{K}, \hat{w}, a)$ denote the event $\{\hat{s}_{h-1}[\mathcal{K}] = \hat{w}, a_{h-1} = a\}$. If $\mathcal{E}(\mathcal{K}, \hat{w}, a)$ occurs in our dataset at least $m$ times for some $m \geq \frac{16}{\epsilon^2} \ln(1/\delta)$ then from Corollary 15 we have

$$\sum_{\hat{v} \in \{0,1\}} \left| \widehat{\mathbb{P}}_D(\hat{s}_h[i] = \hat{v} \mid \hat{s}_{h-1}[\mathcal{K}] = \hat{w}, a) - \mathbb{P}_D(\hat{s}_h[i] = \hat{v} \mid \hat{s}_{h-1}[\mathcal{K}] = \hat{w}, a) \right| \leq \epsilon,$$

with probability at least $1 - \delta$. Lemma 17 shows that probability of $\mathcal{E}(\mathcal{K}, \hat{w}, a)$ is at least $\frac{\alpha \eta_{min}}{4^\kappa N |\mathcal{A}|}$. Therefore, from Lemma 28 if $n_{est} \geq \frac{2^{2\kappa+1}mN|\mathcal{A}|}{\alpha \eta_{min}} \ln\left(\frac{e}{\delta}\right)$ then we get at least $m$ samples of event $\mathcal{E}(\mathcal{K}, \hat{w}, a)$ with probability at least $1 - \delta$. Therefore, the total failure probability is at most $2\delta$: $\delta$ due to not getting at least $m$ samples and $\delta$ due to Corollary 15 on getting $m$ samples. This holds for every triplet $(\mathcal{K}, \hat{w}, a)$ and Lemma 23 shows that there are at most $2(ed)^{2\kappa}|\mathcal{A}|$ such triplets. Hence, with application of union bound we get the desired result. $\qquad\square$

**Lemma 19** (Model Approximation Error). *For any $i \in [d], \mathcal{K} \in \mathscr{C}_{\leq 2\kappa}([d]), s \in \mathcal{S}_{h-1}, a \in \mathcal{A}, s' \in \mathcal{S}_h$, let $\hat{s} = \theta_{h-1}^{-1}(s)$ and $\hat{s}' = \theta_h^{-1}(s')$. Then we have:*

$$|\mathbb{P}_D(\hat{s}'[i] \mid \hat{s}[\mathcal{K}], a) - \mathbb{P}_D(s'[i] \mid s[\mathcal{K}], a)| \leq \Delta_{app} := \frac{5\kappa\varrho N}{\alpha\eta_{min}}.$$

*Proof.* We will first bound $|\mathbb{P}_D(s'[i] \mid \hat{s}[\mathcal{K}], a) - \mathbb{P}_D(s'[i] \mid s[\mathcal{K}], a)|$ and then use correspondence result (Corollary 9) to prove the desired result. We start by expressing our conditional probabilities as ratio of joint probabilities.

$$\mathbb{P}_D(s'[i] \mid \hat{s}[\mathcal{K}], a) = \frac{\mathbb{P}_D(s'[i], \hat{s}[\mathcal{K}], a)}{\mathbb{P}_D(\hat{s}[\mathcal{K}], a)}, \qquad \mathbb{P}_D(s'[i] \mid s[\mathcal{K}], a) = \frac{\mathbb{P}_D(s'[i], s[\mathcal{K}], a)}{\mathbb{P}_D(s[\mathcal{K}], a)}.$$

From Lemma 29 we have:

$$|\mathbb{P}_D(s'[i] \mid \hat{s}[\mathcal{K}], a) - \mathbb{P}_D(s'[i] \mid s[\mathcal{K}], a)| \leq \frac{\varepsilon + \varepsilon_2}{\mathbb{P}_D(s[\mathcal{K}], a)}, \text{ where} \tag{27}$$

$\varepsilon_1 := |\mathbb{P}_D(s'[i], \hat{s}[\mathcal{K}], a) - \mathbb{P}_D(s'[i], s[\mathcal{K}], a)|$ and $\varepsilon_2 := |\mathbb{P}_D(\hat{s}[\mathcal{K}], a) - \mathbb{P}_D(s[\mathcal{K}], a)|$. We bound these two quantities below:

$$\varepsilon_1 = \left| \sum_{s_{h-1}[\mathcal{K}]} \mathbb{P}_D(s'[i], \hat{s}[\mathcal{K}], s_{h-1}[\mathcal{K}], a) - \sum_{\hat{s}_{h-1}[\mathcal{K}]} \mathbb{P}_D(s'[i], \hat{s}_{h-1}[\mathcal{K}], s[\mathcal{K}], a) \right|$$

$$= \left| \sum_{s_{h-1}[\mathcal{K}] \neq s[\mathcal{K}]} \mathbb{P}_D(s'[i], \hat{s}[\mathcal{K}], s_{h-1}[\mathcal{K}], a) - \sum_{\hat{s}_{h-1}[\mathcal{K}] \neq \hat{s}[\mathcal{K}]} \mathbb{P}_D(s'[i], \hat{s}_{h-1}[\mathcal{K}], s[\mathcal{K}], a) \right|$$

$$\leq \max \left\{ \underbrace{\sum_{s_{h-1}[\mathcal{K}] \neq s[\mathcal{K}]} \mathbb{P}_D(s'[i], \hat{s}[\mathcal{K}], s_{h-1}[\mathcal{K}], a)}_{\text{Term 1}}, \underbrace{\sum_{\hat{s}_{h-1}[\mathcal{K}] \neq \hat{s}[\mathcal{K}]} \mathbb{P}_D(s'[i], \hat{s}_{h-1}[\mathcal{K}], s[\mathcal{K}], a)}_{\text{Term 2}} \right\},$$

Where the first inequality uses $|a - b| \leq \max\{a, b\}$ for $a, b > 0$.

We bound Term 1 below:

$$\text{Term 1:} \quad \frac{1}{|\mathcal{A}|} \sum_{s_{h-1}[\mathcal{K}] \neq s[\mathcal{K}]} \mathbb{P}_D(s'[i] \mid \hat{s}[\mathcal{K}], s_{h-1}[\mathcal{K}], a) \mathbb{P}(\hat{s}[\mathcal{K}] \mid s_{h-1}[\mathcal{K}]) \mathbb{P}_D(s_{h-1}[\mathcal{K}])$$

$$\leq \frac{\varrho}{|\mathcal{A}|} \sum_{s_{h-1}[\mathcal{K}] \neq s[\mathcal{K}]} \mathbb{P}_D(s'[i] \mid \hat{s}[\mathcal{K}], s_{h-1}[\mathcal{K}], a) \mathbb{P}_D(s_{h-1}[\mathcal{K}])$$

$$\leq \frac{\varrho}{|\mathcal{A}|} \sum_{s_{h-1}[\mathcal{K}] \neq s[\mathcal{K}]} \mathbb{P}_D(s_{h-1}[\mathcal{K}]) \leq \frac{\varrho}{|\mathcal{A}|}$$

The key inequality here is $\mathbb{P}(\hat{s}[\mathcal{K}] \mid s_{h-1}[\mathcal{K}]) = \prod_{k \in \mathcal{K}} \mathbb{P}(\hat{s}[k] \mid s_{h-1}[k]) \leq \varrho$, as there exist at least one $j \in \mathcal{K}$ such that $s_{h-1}[j] \neq s[j]$ and for this $j$ we have $\mathbb{P}(\hat{s}[j] \mid s_{h-1}[j]) \leq \varrho$.

We bound Term 2 similarly:

$$\text{Term 2:} \quad \frac{1}{|\mathcal{A}|} \sum_{\hat{s}_{h-1}[\mathcal{K}] \neq \hat{s}[\mathcal{K}]} \mathbb{P}_D(s'[i] \mid \hat{s}_{h-1}[\mathcal{K}], s[\mathcal{K}], a) \mathbb{P}_D(\hat{s}_{h-1}[\mathcal{K}] \mid s[\mathcal{K}]) \mathbb{P}_D(s[\mathcal{K}])$$

$$\leq \frac{1}{|\mathcal{A}|} \sum_{\hat{s}_{h-1}[\mathcal{K}] \neq \hat{s}[\mathcal{K}]} \mathbb{P}_D(\hat{s}_{h-1}[\mathcal{K}] \mid s[\mathcal{K}]) = \frac{1}{|\mathcal{A}|} \{1 - \mathbb{P}_D(\hat{s}[\mathcal{K}] \mid s[\mathcal{K}])\}$$

$$\leq \frac{1}{|\mathcal{A}|}(1 - (1 - \varrho)^{|\mathcal{K}|}) \leq \frac{2\kappa\varrho}{|\mathcal{A}|}$$

where we use $\mathbb{P}_D(\hat{s}[\mathcal{K}] \mid s[\mathcal{K}]) = \prod_{k \in \mathcal{K}} \mathbb{P}(\hat{s}[k] \mid s[k]) \geq (1 - \varrho)^{|\mathcal{K}|}$ and $|\mathcal{K}| \leq 2\kappa$.

This gives us $\varepsilon_1 \leq \frac{2\kappa\varrho}{|\mathcal{A}|}$. The proof for $\varepsilon_2$ is similar.

$$
\varepsilon_2 = \left| \sum_{s_{h-1}[\mathcal{K}]} \mathbb{P}_D(\hat{s}[\mathcal{K}], s_{h-1}[\mathcal{K}], a) - \sum_{\hat{s}_{h-1}[\mathcal{K}]} \mathbb{P}_D(\hat{s}_{h-1}[\mathcal{K}], s[\mathcal{K}], a) \right|
$$

$$
= \left| \sum_{s_{h-1}[\mathcal{K}] \neq s[\mathcal{K}]} \mathbb{P}_D(\hat{s}[\mathcal{K}], s_{h-1}[\mathcal{K}], a) - \sum_{\hat{s}_{h-1}[\mathcal{K}] \neq \hat{s}[\mathcal{K}]} \mathbb{P}_D(\hat{s}_{h-1}[\mathcal{K}], s[\mathcal{K}], a) \right|
$$

$$
\max \left\{ \underbrace{\sum_{s_{h-1}[\mathcal{K}] \neq s[\mathcal{K}]} \mathbb{P}_D(\hat{s}[\mathcal{K}], s_{h-1}[\mathcal{K}], a)}_{\text{Term 3}}, \underbrace{\sum_{\hat{s}_{h-1}[\mathcal{K}] \neq \hat{s}[\mathcal{K}]} \mathbb{P}_D(\hat{s}_{h-1}[\mathcal{K}], s[\mathcal{K}], a)}_{\text{Term 4}} \right\}
$$

We bound Term 3 below similar to Term 1:

$$
\text{Term 3:} \quad \frac{1}{|\mathcal{A}|} \sum_{s_{h-1}[\mathcal{K}] \neq s[\mathcal{K}]} \mathbb{P}_D(\hat{s}[\mathcal{K}]|s_{h-1}[\mathcal{K}])\mathbb{P}_D(s_{h-1}[\mathcal{K}])
$$

$$
\leq \frac{\varrho}{|\mathcal{A}|} \sum_{s_{h-1}[\mathcal{K}] \neq s[\mathcal{K}]} \mathbb{P}_D(s_{h-1}[\mathcal{K}]) \leq \frac{\varrho}{|\mathcal{A}|}
$$

and Term 4 is bounded similar to Term 2 below:

$$
\text{Term 4:} \quad \frac{1}{|\mathcal{A}|} \sum_{\hat{s}_{h-1}[\mathcal{K}] \neq \hat{s}[\mathcal{K}]} \mathbb{P}_D(\hat{s}_{h-1}[\mathcal{K}] \mid s[\mathcal{K}])\mathbb{P}_D(s[\mathcal{K}])
$$

$$
\leq \frac{1}{|\mathcal{A}|} \left\{ 1 - \mathbb{P}_D(\hat{s}[\mathcal{K}] \mid s[\mathcal{K}]) \right\}
$$

$$
\leq \frac{1}{|\mathcal{A}|} \left\{ 1 - (1 - \varrho)^{|\mathcal{K}|} \right\} \leq \frac{2\kappa\varrho}{|\mathcal{A}|}
$$

This gives us $\varepsilon_2 \leq \frac{2\kappa\varrho}{|\mathcal{A}|}$. Plugging bounds for $\varepsilon_1$ and $\varepsilon_2$ in Equation 27 and using $\mathbb{P}_D(s[\mathcal{K}], a) = \frac{\mathbb{P}_D(s[\mathcal{K}])}{|\mathcal{A}|} \geq \frac{\alpha\eta_{min}}{N|\mathcal{A}|}$ gives us:

$$
|\mathbb{P}_D(s'[i] \mid \hat{s}[\mathcal{K}], a) - \mathbb{P}_D(s'[i] \mid s[\mathcal{K}], a| \leq \frac{4\kappa\varrho}{|\mathcal{A}|\mathbb{P}_D(s[\mathcal{K}], a)} \leq \frac{4\kappa\varrho N}{\alpha\eta_{min}}. \tag{28}
$$

We can use correspondence result to derive a lower bound:

$$
\mathbb{P}_D(\hat{s}'[i] \mid \hat{s}[\mathcal{K}], a) \geq \mathbb{P}_D(\hat{s}'[i] \mid s'[i])\mathbb{P}_D(s'[i] \mid \hat{s}[\mathcal{K}], a)
$$

$$
\geq (1 - \varrho)\mathbb{P}_D(s'[i] \mid \hat{s}[\mathcal{K}], a) \geq \mathbb{P}_D(s'[i] \mid \hat{s}[\mathcal{K}], a) - \varrho
$$

and an upper bound:

$$
\mathbb{P}_D(\hat{s}'[i] \mid \hat{s}[\mathcal{K}], a) = \mathbb{P}_D(\hat{s}'[i] \mid s'[i])\mathbb{P}_D(s'[i] \mid \hat{s}[\mathcal{K}], a) +
$$

$$
\mathbb{P}(\hat{s}'[i] \mid 1 - s'[i])\mathbb{P}_D(1 - s'[i] \mid \hat{s}[\mathcal{K}], a)
$$

$$
\leq \mathbb{P}_D(s'[i] \mid \hat{s}[\mathcal{K}], a) + \varrho
$$

Combing the lower and upper bounds with Equation 28 gives us:

$$
|\mathbb{P}_D(\hat{s}'[i] \mid \hat{s}[\mathcal{K}], a) - \mathbb{P}_D(s'[i] \mid s[\mathcal{K}], a)| \leq \frac{4\kappa\varrho N}{\alpha\eta_{min}} + \varrho \leq \frac{5\kappa\varrho N}{\alpha\eta_{min}}.
$$

which is the desired result. $\qquad\square$

We can merge the estimation error and approximation error to generate the total error.

**Lemma 20** ($\mathcal{K}$-Model Error). *For any $i \in [d]$, $\mathcal{K} \in \mathscr{C}_{\leq 2\kappa}([d])$, $s \in \mathcal{S}_{h-1}$, $a \in \mathcal{A}$, $s' \in \mathcal{S}_h$, let $\hat{s} = \theta_{h-1}^{-1}(s)$ and $\hat{s}' = \theta_h^{-1}(s')$. Then we have:*

$$\left| \widehat{\mathbb{P}}_D(\hat{s}'[i] \mid \hat{s}[\mathcal{K}], a) - \mathbb{P}_D(s'[i] \mid s[\mathcal{K}], a) \right| \leq \Delta_{est}(n_{est}, \delta_{est}) + \Delta_{app}.$$

*with probability at least $1 - \delta_{est}$.*

*Proof.* Follows trivially by combining the estimation error (Lemma 18) and approximation error (Lemma 19) with application of triangle inequality. $\square$

## D.4 DETECTING LATENT PARENT STRUCTURE IN TRANSITION $pt_h$

We are now ready to analyze the performance of learned parent function $\widehat{pt}_h$. Let $\mathcal{K}_1, \mathcal{K}_2 \in \mathscr{C}_{\leq 2\kappa}([2d])$ and $\hat{w}_1 \in \{0,1\}^{|\mathcal{K}_1|}, \hat{w}_2 \in \{0,1\}^{|\mathcal{K}_2|}$. We will assume $\hat{w}_1$ is reachable for $\mathcal{K}_1$ and $\hat{w}_2$ is reachable for $\mathcal{K}_2$. For convenience we will define the following quantity $\Omega$ to measure total variation distance between distributions $\widehat{\mathbb{P}}_D(s'[i] \mid \cdot, \cdot)$ conditioned on setting $\hat{s}[\mathcal{K}_1] = \hat{w}_1$ and $\hat{s}[\mathcal{K}_2] = \hat{w}_2$, and for a fixed action $a \in \mathcal{A}$:

$$\widehat{\Omega}_{ia}(\mathcal{K}_1, \hat{w}_1, \mathcal{K}_2, \hat{w}_2) := \frac{1}{2} \sum_{\hat{v} \in \{0,1\}} \left| \widehat{\mathbb{P}}_D(\hat{s}'[i] = \hat{v} \mid \hat{s}[\mathcal{K}_1] = \hat{w}_1, a) - \widehat{\mathbb{P}}_D(\hat{s}'[i] = \hat{v} \mid \hat{s}[\mathcal{K}_2] = \hat{w}_2, a) \right|.$$

We can compute $\widehat{\Omega}$ for every value of $i, \mathcal{K}_1, \hat{w}_1, \mathcal{K}_2, \hat{w}_2, a$ in computational time of $\mathcal{O}\left((2ed)^{3\kappa+3}|\mathcal{A}|\right)$. We also define a similar metric for the true distribution for any $\mathcal{K}_1, \mathcal{K}_2$ and $v \in \{0,1\}, w_1 \in \{0,1\}^{|\mathcal{K}_1|}, w_2 \in \{0,1\}^{|\mathcal{K}_2|}$ and $a \in \mathcal{A}$:

$$\Omega_{ia}(\mathcal{K}_1, w_1, \mathcal{K}_2, w_2) := \frac{1}{2} \sum_{v \in \{0,1\}} \left| \mathbb{P}_D(s'[i] = v \mid s[\mathcal{K}_1] = w_1, a) - \mathbb{P}_D(s'[i] = v \mid s[\mathcal{K}_2] = w_2, a) \right|.$$

Recall that $[\mathcal{I}; \mathcal{J}]$ denotes concatenation of two ordered sets $\mathcal{I}$ and $\mathcal{J}$. We use this notation to state our next result.

**Lemma 21** (Inclusive Case). *Fix $i \in [d]$ and $\mathcal{I} \in \mathscr{C}_{\leq \kappa}([d])$. If $pt(i) \subseteq \mathcal{I}$ then for all $a \in \mathcal{A}$ and $\hat{u} \in \{0,1\}^{|\mathcal{I}|}$ we get:*

$$\max_{\mathcal{J}_1, \mathcal{J}_2, \hat{w}_1, \hat{w}_2} \widehat{\Omega}_{ia}([\mathcal{I}; \mathcal{J}_1], [\hat{u}; \hat{w}_1], [\mathcal{I}; \mathcal{J}_2], [\hat{u}; \hat{w}_2]) \leq 2\Delta_{est}(n_{est}, \delta_{est}) + 2\Delta_{app},$$

*where max is taken over $\mathcal{J}_1, \mathcal{J}_2 \in \mathscr{C}_{\leq \kappa}([d])$, $\hat{w}_1 \in \{0,1\}^{|\mathcal{J}_1|}$, $\hat{w}_2 \in \{0,1\}^{|\mathcal{J}_2|}$ such that $[\hat{u}; \hat{w}_1]$ is reachable for $[\mathcal{I}; \mathcal{J}_1]$, $[\hat{u}; \hat{w}_2]$ is reachable for $[\mathcal{I}; \mathcal{J}_2]$, and $\mathcal{I} \cap \mathcal{J}_1 = \mathcal{I} \cap \mathcal{J}_2 = \emptyset$.*

*Proof.* We fix $\mathcal{J}_1, \mathcal{J}_2, \hat{u}, \hat{w}_1, \hat{w}_2, a$ and let $\mathcal{K}_1 = [\mathcal{I}; \mathcal{J}_1]$, $\mathcal{K}_2 = [\mathcal{I}; \mathcal{J}_2]$, $v = \theta(\hat{v})$, $u = \theta(\hat{u})$, $w_1 = \theta(\hat{w}_1)$, and $w_2 = \theta(\hat{w}_2)$. As $pt(i) \subseteq \mathcal{I}$, therefore, we have:

$$\mathbb{P}_D(s'[i] = v \mid s[\mathcal{K}_1] = [u; w_1], a) = T_i(s'[i] = v \mid s[\mathcal{I}] = u, a) = \mathbb{P}_D(s'[i] = v \mid s[\mathcal{K}_2] = [u; w_2], a)$$

Using this result along with Lemma 20 and application of triangle inequality we get:

$$\left| \widehat{\mathbb{P}}_D(\hat{s}'[i] = \hat{v} \mid \hat{s}[\mathcal{K}_1] = [\hat{u}; \hat{w}_1], a) - \widehat{\mathbb{P}}_D(\hat{s}'[i] = \hat{v} \mid \hat{s}[\mathcal{K}_2] = [\hat{u}; \hat{w}_2], a) \right| \leq 2\Delta_{est}(n_{est}, \delta_{est}) + 2\Delta_{app}.$$

Summing over $\hat{v}$, dividing by 2, and using the definition of $\widehat{\Omega}$ proves the result. $\square$

The following is a straightforward corollary of Lemma 21.

**Corollary 11.** *Fix $i \in [d]$ then there exists an $\mathcal{I}$ such that for all $a \in \mathcal{A}$ and $\hat{u} \in \{0,1\}^{|\mathcal{I}|}$:*

$$\max_{\mathcal{J}_1, \mathcal{J}_2, \hat{w}_1, \hat{w}_2} \widehat{\Omega}_{ia}([\mathcal{I}; \mathcal{J}_1], [\hat{u}; \hat{w}_1], [\mathcal{I}; \mathcal{J}_2], [\hat{u}; \hat{w}_2]) \leq 2\Delta_{est}(n_{est}, \delta_{est}) + 2\Delta_{app},$$

*where max is taken over $\mathcal{J}_1, \mathcal{J}_2, \hat{w}_1, \hat{w}_2$ satisfy the restrictions stated in Lemma 21.*

*Proof.* Take any $\mathcal{I}$ such that $pt(i) \subseteq \mathcal{I}$ and apply Lemma 21. Note that we are allowed to pick such an $\mathcal{I}$ as $|pt(i)| \leq \kappa$ by our assumption. $\square$

Recall that we define $\widehat{pt}(i)$ as the solution of the following problem:

$$\widehat{pt}(i) := \operatorname*{argmin}_{\mathcal{I}} \max_{a,\hat{u},\mathcal{J}_1,\mathcal{J}_2,\hat{w}_1,\hat{w}_2} \widehat{\Omega}_{ia}([\mathcal{I};\mathcal{J}_1],[\hat{u};\hat{w}_1],[\mathcal{I};\mathcal{J}_2],[\hat{u};\hat{w}_2]), \qquad (29)$$

where $\mathcal{I} \in \mathscr{C}_{\leq\kappa}([d])$, $a \in \mathcal{A}$, and $\hat{u}, \mathcal{J}_1, \mathcal{J}_2, \hat{w}_1, \hat{w}_2$ satisfy the restrictions stated in Lemma 21.

We are now ready to state our main result for $\widehat{pt}$.

**Theorem 12** (Property of $\widehat{pt}$). *For any $s \in \mathcal{S}_{h-1}$, $a \in \mathcal{A}$, $s' \in \mathcal{S}_h$, let $\hat{s} = \theta_{h-1}^{-1}(s)$ and $\hat{s}' = \theta_h^{-1}(s')$. Then the learned parent function $\widehat{pt}$ satisfies:*

$$\forall \in [d], \qquad \left| \widehat{\mathbb{P}}_D(\hat{s}'[i] \mid \hat{s}[\widehat{pt}(i)], a) - T_i(s'[i] \mid s[pt(i)], a) \right| \leq 3\Delta_{est}(n_{est}, \delta_{est}) + 3\Delta_{app}.$$

*Proof.* Fix $i \in [d]$. Let $\mathcal{J} = pt(i) - \widehat{pt}(i)$ and $\mathcal{K} = \widehat{pt}(i) \cup \mathcal{J}$. As $pt(i) \subseteq \mathcal{K}$, therefore, we have:

$$\mathbb{P}_D(s'[i] \mid s[\mathcal{K}], a) = T_i(s'[i] \mid s[pt(i)], a)$$

Combining this result with Lemma 20 we get:

$$\left| \widehat{\mathbb{P}}_D(\hat{s}'[i] \mid \hat{s}[\mathcal{K}], a) - T_i(s'[i] \mid s[pt(i)], a) \right| \leq \Delta_{\text{est}}(n_{\text{est}}, \delta_{\text{est}}) + \Delta_{\text{app}}.$$

From the definition of $\widehat{pt}(i)$ (Equation 29) and Corollary 11 we have:

$$\left| \widehat{\mathbb{P}}_D(\hat{s}'[i] \mid \hat{s}[\widehat{pt}(i); \emptyset], a) - \widehat{\mathbb{P}}_D(\hat{s}'[i] \mid \hat{s}[\widehat{pt}(i); \mathcal{J}], a) \right| \leq 2\Delta_{\text{est}}(n_{\text{est}}, \delta_{\text{est}}) + 2\Delta_{\text{app}}.$$

Note that we are allowed to use Corollary 11 as $\hat{s}[\widehat{pt}(i); \emptyset]$ and $\hat{s}[\widehat{pt}(i); \mathcal{J}]$ are both reachable since they are derived from a reachable real state $s$, $|\widehat{pt}(i)| \leq \kappa$, $|[\widehat{pt}(i); \mathcal{J}]| \leq |[\widehat{pt}(i); pt(i)]| \leq 2\kappa$, and $\widehat{pt}(i) \cap \emptyset = \emptyset = \widehat{pt}(i) \cap \mathcal{J}$. Combining the previous two inequalities using triangle inequality completes the proof. $\square$

## D.5 Bound Total Variation between Estimated Model and True Model

Given the learned transition parent function $\widehat{pt}_h$ we define the transition model as:

$$\widehat{T}_{hi}\left(\hat{s}'[i] \mid \hat{s}[\widehat{pt}_h(i)], a\right) = \widehat{\mathbb{P}}_D\left(\hat{s}'[i] \mid \hat{s}[\widehat{pt}_h(i)], a\right), \qquad \widehat{T}_h\left(\hat{s}' \mid \hat{s}, a\right) = \prod_{i=1}^{d} \widehat{T}_{hi}\left(\hat{s}'[i] \mid \hat{s}[\widehat{pt}_h(i)], a\right).$$

From Theorem 12 we have for any $i \in [d]$, $\hat{s} \in \mathcal{S}_{h-1}$, $a \in \mathcal{A}$, $s' \in \mathcal{S}_h$, and $\hat{s} = \theta^{-1}(s)$, $\hat{s}' = \theta^{-1}(s')$:

$$\left| \widehat{T}_{hi}(\hat{s}'[i] \mid \hat{s}[\widehat{pt}(i)], a) - T(s'[i] \mid s[pt(i)], a) \right| \leq 3\Delta_{\text{est}}(n_{\text{est}}, \delta_{\text{est}}) + 3\Delta_{\text{app}}.$$

**Transition Closure.** A subtle point remains before we prove the model error between $\widehat{T}_h$ and $T$. Theorem 12 only states guarantee for those $\hat{s}$ that are inverse of a reachable state $s$. However, as stated before, due to decoder error we can reach a state $\hat{s}$ which does not have a corresponding reachable state, i.e. $\theta(\hat{s}) \notin \mathcal{S}_h$ (see Figure 3). We cannot get model guarantees for these unreachable states $\hat{s}$ since we may reach them with arbitrarily small probability. However, we can still derive model error if we can simply define the real transition probabilities in terms of the learned probabilities for these states. This will not cause a problem since the real model will never reach these states. We start by defining the closure of the transition model $T^\circ$ for time step $h$ as:

$$\forall \hat{s} \in \widehat{\mathcal{S}}_{h-1}, a \in \mathcal{A}, \hat{s}' \in \mathcal{S}_h, \qquad T_h^\circ(\theta(\hat{s}') \mid \theta(\hat{s}), a) = \begin{cases} T(\theta(\hat{s}') \mid \theta(\hat{s}), a), & \text{if } \theta(\hat{s}) \in \mathcal{S}_{h-1} \\ \widehat{T}_h(\hat{s}' \mid \hat{s}, a), & \text{otherwise} \end{cases}$$

We also define the state space domain of $T_h^\circ$ as $\mathcal{S}_{h-1}^\circ = \{\theta_{h-1}(\hat{s}) \mid \forall \hat{s} \in \widehat{\mathcal{S}}_{h-1}\}$. It is easy to see that $\theta_{h-1}$ represents a bijection between $\widehat{\mathcal{S}}_{h-1}$ and $\mathcal{S}_{h-1}^\circ$.

We will derive our guarantees with respect to $T^\circ$ which will allow us to define a bijection between the domain of $\widehat{T}$ and $T^\circ$, and use important lemmas from the literature. The next result shows that our use of $T^\circ$ is harmless as it assigns the same probability as $T$ to any event.

**Lemma 22** (Closure Result). *Let $T^\circ$ be the closure of transition model with respect to some learned transition model. Then for any policy $\pi \in \Pi$ and any event $\mathcal{E}$ which is a function of an episode sampled using $\pi$, we have $\mathbb{P}_\pi(\mathcal{E}; T) = \mathbb{P}_\pi(\mathcal{E}; T^\circ)$, where $\mathbb{P}_\pi(\mathcal{E}; T')$ denotes the probability of event $\mathcal{E}$ when sampling from $\pi$ and using transition model $T'$.*

*Proof.* The proof follows form observing that when using $T^\circ$ we will never reach a state $s \notin \mathcal{S}_{h-1}$ for any $h-1$ by definition of $\mathcal{S}_{h-1}$. From definition of $T^\circ$ this means that both $T^\circ$ and $T$ will generate the same range of episodes sampled from $\pi$ and will assign the same probabilities to them. As $\mathcal{E}$ is a function of an episode, therefore, its probability remains unchanged. $\qquad\square$

With the definition of closure, we are now ready to state our last result in this section, which bounds the total variation between the estimated model and the transition closure under the bijection map $\theta$.

**Theorem 13** (Model Error). *For any $\hat{s} \in \widehat{\mathcal{S}}_{h-1}$ and $a \in \mathcal{A}$ we have:*

$$\sum_{\hat{s}' \in \widehat{\mathcal{S}}_h} \left| \widehat{T}_h(\hat{s}' \mid \hat{s}, a) - T_h^\circ(\theta(\hat{s}') \mid \theta(\hat{s}), a) \right| \leq 6d \left( \Delta_{est}(n_{est}, \delta_{est}) + \Delta_{app} \right).$$

*Proof.* If $\theta(\hat{s}) \notin \mathcal{S}_{h-1}$ then by definition $T^\circ$ the bound holds trivially. Therefore, we focus on $\theta(\hat{s}) \in \mathcal{S}_{h-1}$ for which $T^\circ = T$. We define the quantity for every $j \in [d]$:

$$S_j = \sum_{\hat{s}'[j]\cdots\hat{s}'[d] \in \{0,1\}} \left| \prod_{i=j}^d \widehat{T}_{hi}(\hat{s}'[i] \mid \hat{s}[\widehat{pt}(i)], a) - \prod_{i=j}^d T_i(\theta(\hat{s}')[i] \mid \theta(\hat{s})[pt(i)], a) \right| \qquad (30)$$

We claim that $S_j \leq 6(d - j + 1)(\Delta_{est} + \Delta_{app})$ for every $j \in [d]$. For base case we have:

$$S_d = \sum_{\hat{s}'[d] \in \{0,1\}} \left| \widehat{T}_{hd}(\hat{s}'[d] \mid \hat{s}[\widehat{pt}(d)], a) - T(\theta(\hat{s}')[d] \mid \theta(\hat{s})[pt(d)], a) \right| \leq 6(\Delta_{est} + \Delta_{app}),$$

from Theorem 12. We will assume the induction hypothesis to be true for $S_k$ for all $k > j$. We handle the inductive below with triangle inequality:

$$S_j \leq \sum_{\hat{s}'[j]\cdots\hat{s}'[d] \in \{0,1\}} \prod_{i=j+1}^d \widehat{T}_{hi}(\hat{s}'[i] \mid \hat{s}[\widehat{pt}(i)], a) | \widehat{T}_{hj}(\hat{s}'[j] \mid \hat{s}[\widehat{pt}(j)], a) -$$

$$T(\theta(\hat{s}')[j] \mid \theta(\hat{s})[pt(j)], a) | +$$

$$\sum_{\hat{s}'[j]\cdots\hat{s}'[d] \in \{0,1\}} T(\theta(\hat{s}')[j] \mid \theta(\hat{s})[pt(j)], a) | \prod_{i=j+1}^d \widehat{T}_{hi}(\hat{s}'[i] \mid \hat{s}[\widehat{pt}(i)], a) -$$

$$\prod_{i=j+1}^d T(\theta(\hat{s}')[i] \mid \theta(\hat{s})[pt(i)], a) |$$

The first term is equivalent to $\sum_{\hat{s}'[j] \in \{0,1\}} |\widehat{T}_{hj}(\hat{s}'[j] \mid \hat{s}[\widehat{pt}(j)], a) - T(\theta(\hat{s}')[j] \mid \theta(\hat{s})[pt(j)], a)|$ which is bounded by $6(\Delta_{est} + \Delta_{app})$ following base case analysis. The second term is equivalent to $S_{j+1}$ which is bounded by $6(d - j)(\Delta_{est} + \Delta_{app})$ by induction hypothesis. Combining these two bounds proves the induction hypothesis and the result then follows from bound for $S_1$. $\qquad\square$

### D.6 LEARNING A POLICY COVER

In this section we show how we learn the policy cover. We start by defining some notation.

**Two MDPs.** After time step $h$, we can define two Markov Decision Processes (MDPs) at this time $\mathcal{M}_h$ and $\widehat{\mathcal{M}}_h$. $\mathcal{M}_h$ is the true MDP consists of state space $(\mathcal{S}_1^\circ, \cdots, \mathcal{S}_h^\circ)$, action space $\mathcal{A}$, horizon $h$, a deterministic start state $s_1 = \{0\}^d$, and transition function $T_t^\circ : \mathcal{S}_{t-1}^\circ \times \mathcal{A} \to \Delta(\mathcal{S}_t)$ for all $t \in [h]$. Recall that the set $\mathcal{S}_h \subseteq \{0,1\}^d$ denote states which are reachable at time step $h$, and the set

$\mathcal{S}_t \subseteq \mathcal{S}_t^\circ \subseteq \{0,1\}^d$ represents the *closure* of state space $\mathcal{S}_h$ based on the learned state space $\widehat{\mathcal{S}}_t$. For any $t \in [h]$, $s \in \mathcal{S}_t$ and $\mathcal{K} \in \mathscr{C}_{\le 2\kappa}([d])$ we know $\sup_{\pi \in \Pi_{\mathrm{NS}}} \mathbb{P}_\pi(s_t[\mathcal{K}] = s[\mathcal{K}]) \ge \eta_{min}$.

The second MDP $\widehat{\mathcal{M}}_h$ consists of the learned state space $(\widehat{\mathcal{S}}_1, \cdots, \widehat{\mathcal{S}}_h)$, action space $\mathcal{A}$, horizon $h$, a deterministic start state $\hat{s}_1 = \{0\}^d$, and transition function $\widehat{T}_t : \widehat{\mathcal{S}}_{t-1} \times \mathcal{A} \to \Delta(\widehat{\mathcal{S}}_t)$.

For every $t \in [h]$, we have $\theta_t : \widehat{\mathcal{S}}_t \to \mathcal{S}_t^\circ$ represent a bijection from the learned state space to the closure of the set of reachable states at time step $t$. The learned decoder $\widehat{\phi}_t$ predict $\theta_t(s)$ given $s \in \mathcal{S}_t$ with high probability for all $t < h$ by IH.2 and for $t = h$ due to Corollary 9.

Lastly, the transition model $T_t^\circ$ and $T_t$ are close in $L_1$ distance for $t < h$ due to IH.3 and for $t = h$ due to Theorem 13.

These results enable us to utilize the analysis of Du et al. (2019) for learning to learn a policy cover.

Let $\hat{\varphi} : \widehat{\mathcal{S}} \to \mathcal{A}$ denote a non-stationary deterministic policy that operates on the learned state space. Similarly, $\varphi : \mathcal{S}^\circ \to \mathcal{A}$ denote a non-stationary deterministic policy that operates on the real state. We denote $\hat{\varphi} = \varphi \circ \theta$ if for every $\hat{s} \in \widehat{\mathcal{S}}$, $\hat{\varphi}(\hat{s}) = \varphi(\theta(\hat{s}))$. Similarly, we denote $\varphi = \hat{\varphi} \circ \theta^{-1}$ if for every $s \in \mathcal{S}^\circ$, $\varphi(s) = \hat{\varphi}(\theta^{-1}(s))$. Let $\pi : \mathcal{X} \to \mathcal{A}$ be a non-stationary deterministic policy operating on the observation space. We say $\pi = \hat{\varphi} \circ \widehat{\phi}$ if for every $x \in \mathcal{X}$ we have $\pi(x) = \hat{\varphi}(\widehat{\phi}(x))$. Similarly, we define $\pi = \varphi \circ \phi^\star$ if for every $x \in \mathcal{X}$ we have $\pi(x) = \varphi(\phi^\star(x))$.

We will use $\mathbb{P}_\pi[\mathcal{E}]$ to denote probability of an event $\mathcal{E}$ when actions are taken according to policy $\pi : \mathcal{X} \to \mathcal{A}$. We will use $\mathbb{P}_\varphi[\mathcal{E}]$ to denote the probability of event $\mathcal{E}$ when we operate directly on the real state and take actions using $\varphi$. Similarly, we define $\mathbb{P}_{\hat{\varphi}}[\mathcal{E}]$ to denote the probability of event $\hat{\mathcal{E}}$ when we operate on the learned state space. Lastly, let $\widehat{\mathbb{P}}_{\hat{\varphi}}[\mathcal{E}]$ denote probability of an event $\mathcal{E}$ when actions are taken according to policy $\hat{\varphi}$ operating directly over the latent state and following our estimated transition dynamics $\widehat{T} : \widehat{\mathcal{S}} \times \mathcal{A} \to \Delta(\widehat{\mathcal{S}})$. Recall that our planner will be optimizing with respect to $\widehat{\mathbb{P}}_{\hat{\varphi}}[\mathcal{E}]$.

**Theorem 14** (Planner Guarantee). *Fix $\Delta_{pl} \ge 0, h \in [H]$. Let $\mathcal{I} \in \mathscr{C}_{\le 2\kappa}([d])$ and $\hat{w} \in \{0,1\}^{|\mathcal{I}|}$. We define a reward function $R : \widehat{\mathcal{S}} \to [0,1]$ where $R(\hat{s}) := \mathbf{1}\{\tau(\hat{s}) = h \wedge \hat{s}[\mathcal{I}] = \hat{w}\}$. Let $\hat{\varphi}_R = \mathtt{planner}(\widehat{T}, R, h, \Delta_{pl})$ be the policy learned by the planner. Let $\hat{\pi} := \hat{\varphi}_R \circ \widehat{\phi}$ then:*

$$\mathbb{P}_{\hat{\pi}}(s_h[\mathcal{I}] = \theta(\hat{w})) \ge \eta(s_h[\mathcal{I}] = \theta(w)) - 2d\varrho H - 12dH\Delta_{est} - 12dH\Delta_{app} - \Delta_{pl}, \tag{31}$$

*further, we have:*

$$\widehat{\mathbb{P}}_{\hat{\varphi}_R}(\{\hat{s}_h[\mathcal{I}] = \hat{w}\}) \ge \eta(s_h[\mathcal{I}] = \theta(w)) - 6dH\Delta_{est} - 6dH\Delta_{app} - \Delta_{pl}, \tag{32}$$

*and if $\{s_h[\mathcal{I}] = \theta(\hat{w})\}$ is unreachable, then*

$$\widehat{\mathbb{P}}_{\hat{\varphi}_R}(\{\hat{s}_h[\mathcal{I}] = \hat{w}\}) \le 6dH\Delta_{est} + 6dH\Delta_{app}. \tag{33}$$

*Proof.* We define two events $\mathcal{E} := \{s_h[\mathcal{I}] = \theta(\hat{w})\}$ and $\hat{\mathcal{E}} := \{\hat{s}_h[\mathcal{I}] = \hat{w}\}$. We define a policy $\varphi_R = \hat{\varphi}_R \circ \theta^{-1}$ where for every $s \in \mathcal{S}$ we have $\varphi_R(s) = \hat{\varphi}_R(\theta^{-1}(s))$. We also define $\bar{\pi} : \mathcal{X} \to \mathcal{A}$ as $\bar{\pi}(x) = \varphi_R \circ \phi^\star(x)$. If for a given $x \in \mathcal{X}$ and $\phi^\star(x) = s$ we have $\widehat{\phi}(x) = \theta^{-1}(s)$ then $\bar{\pi}(x) = \hat{\varphi}_R(\widehat{\phi}(x)) = \hat{\pi}(x)$. Hence, every time our decoder outputs the correct mapped state $\theta(s)$, policies $\bar{\pi}$ and $\hat{\pi}$ take the same action. We use the result of Du et al. (2019) stated in Lemma 30 (setting $\varepsilon$ set to $d\varrho$ using Corollary 9) to write:

$$|\mathbb{P}_{\hat{\pi}}(\mathcal{E}) - \mathbb{P}_{\bar{\pi}}(\mathcal{E})| = |\mathbb{P}_{\hat{\pi}}(\mathcal{E}) - \mathbb{P}_{\varphi_R}(\mathcal{E})| \le 2d\varrho H \tag{34}$$

Let $\varphi : \mathcal{S}^\circ \to \mathcal{A}$ be any policy on real state space and let $\hat{\varphi} : \widehat{\mathcal{S}} \to \mathcal{A}$ be the induced policy on learned state space given by $\hat{\varphi}(\hat{s}) = \varphi \circ \theta(\hat{s}) = \varphi(\theta(\hat{s}))$ for any $\hat{s} \in \widehat{\mathcal{S}}$. We showed in Theorem 13 that $\widehat{T}$ and $T$ have small $L_1$ distance under the bijection $\theta$. Therefore, from the perturbation result of Du et al. (2019) stated in Lemma 31 we have:

$$\sum_{s_h \in \mathcal{S}_h^\circ} \left| \widehat{\mathbb{P}}_{\hat{\varphi}}(\theta^{-1}(\hat{s}_h)) - \mathbb{P}_\varphi(s_h) \right| \le h\varepsilon \le H\varepsilon,$$

where $\varepsilon := 6d \left( \Delta_{\text{est}}(n_{\text{est}}, \delta_{\text{est}}) + \Delta_{\text{app}} \right)$ due to Theorem 13. As $\{s_h[\mathcal{I}] = \theta(\hat{w})\} \Leftrightarrow \{\hat{s}_h[\mathcal{I}] = \hat{w}\}$, therefore, we can derive the following bound:

$$\left| \mathbb{P}_\varphi(\mathcal{E}) - \hat{\mathbb{P}}_{\hat{\varphi}}(\hat{\mathcal{E}}) \right| = \left| \sum_{s_h \in \mathcal{S}_h^\circ; s_h[\mathcal{I}]=\theta(\hat{w})} \mathbb{P}_\varphi(s_h) - \sum_{s_h \in \mathcal{S}_h^\circ; s_h[\mathcal{I}]=\theta(\hat{w})} \hat{\mathbb{P}}_{\hat{\varphi}}(\theta^{-1}(s_h)) \right|$$

$$\leq \sum_{s_h \in \mathcal{S}_h^\circ} \left| \mathbb{P}_\varphi(s_h) - \hat{\mathbb{P}}_{\hat{\varphi}}(\theta^{-1}(s_h)) \right| \leq H\varepsilon \tag{35}$$

Let $\varphi^\star = \arg\max \mathbb{P}_\varphi[\mathcal{E}]$ be the optimal policy to satisfy $\{s_h[\mathcal{I}] = \theta(w)\}$. Note that $\varphi^\star$ is also the latent policy that optimizes the reward function $R$ on the real dynamics. Let $\hat{\varphi}^\star = \varphi^\star \circ \theta^{-1}$ be the induced policy on learned states. We now bound the desired quantity as shown:

$$\begin{aligned}
\mathbb{P}_{\hat{\pi}}(\mathcal{E}) &\geq \mathbb{P}_{\varphi_R}[\mathcal{E}] - 2d\varrho H && \text{(using Equation 34)} \\
&\geq \hat{\mathbb{P}}_{\hat{\varphi}_R}(\hat{\mathcal{E}}) - 2d\varrho H - H\varepsilon && \text{(using Equation 35)} \\
&\geq \hat{\mathbb{P}}_{\hat{\varphi}^\star}(\hat{\mathcal{E}}) - 2d\varrho H - H\varepsilon - \Delta_{\text{pl}} && (\varphi_R \text{ is } \Delta_{\text{pl}}\text{-optimal on } \widehat{T}) \\
&\geq \mathbb{P}_{\varphi^\star}(\mathcal{E}) - 2d\varrho H - 2H\varepsilon - \Delta_{\text{pl}} && \text{(using Equation 35)} \\
&= \eta(s_h[\mathcal{I}] = \theta(w)) - 2d\varrho H - 2H\varepsilon - \Delta_{\text{pl}}.
\end{aligned}$$

This proves Equation 31 and Equation 32. Note that our calculations above show:

$$\hat{\mathbb{P}}_{\hat{\varphi}_R}(\hat{\mathcal{E}}) \leq \mathbb{P}_{\varphi_R}[\mathcal{E}] + H\varepsilon$$

If $\{s_h[\mathcal{I}] = \theta(\hat{w})\}$ is unreachable then $\mathbb{P}_{\varphi_R}[\mathcal{E}] = 0$. Plugging this in the above equation proves Equation 33 and completes the proof. $\square$

## D.7 Wrapping up the proof for FactoRL

We are almost done. All we need to do is to make sure is to set the hyperparameters and verify each induction hypothesis. We first set hyperparameters.

**Setting Hyperparameters.** Let $\{s_h[\mathcal{I}] = \theta(\hat{w})\}$ be reachable for some $\mathcal{I} \in \mathscr{C}_{\leq 2\kappa}([d])$ and $\hat{w} \in \{0,1\}^{|\mathcal{I}|}$. Then applying Theorem 14 and using the definition of $\eta_{min}$ we have:

$$\mathbb{P}_{\hat{\pi}}\left(s_h[\mathcal{I}] = \theta(w)\right) \geq \eta_{min} - 2d\varrho H - 12dH\Delta_{\text{est}} - 12dH\Delta_{\text{app}} - \Delta_{\text{pl}}$$

As we want the right hand side to be at least $\alpha\eta_{min}$ we divide the error equally between the three terms. This gives us:

$$\text{(Planning Error)} \quad \Delta_{\text{pl}} \leq {}^{(1-\alpha)\eta_{min}}/_4 \tag{36}$$

$$\text{(Model Approximation Error)} \quad \Delta_{\text{app}} \leq {}^{(1-\alpha)\eta_{min}}/_{48dH} \Rightarrow \varrho \leq \frac{\alpha(1-\alpha)\eta_{min}^2}{240\kappa dHN} \tag{37}$$

$$\text{(Model Estimation Error)} \quad \Delta_{\text{est}} \leq {}^{(1-\alpha)\eta_{min}}/_{48dH}$$

$$\Rightarrow n_{\text{est}} \geq \frac{18432\,2^\kappa d^2 H^2 N|\mathcal{A}|}{\alpha(1-\alpha)^2\eta_{min}^3} \ln^2\left(\frac{4e|\mathcal{A}|(ed)^{2\kappa}}{\delta_{\text{est}}}\right) \tag{38}$$

$$\text{(Decoding Error)} \quad \varrho \leq {}^{(1-\alpha)\eta_{min}}/_{8dH} \tag{39}$$

The model approximation error places a more stringent requirement on $\varrho$ than the decoding error for planning. However, throughout the proof for FactoRL in this section, we made other requirements on our hyperparameters. For $\varrho$ this is given by $\min\{\beta_{\min}^2/1200, 1/2\} = \beta_{\min}^2/1200$ by combining constraints in Lemma 8 and Theorem 7, and an additional constraint for detecting non-degenerate factors stated in Corollary 9. Due to the inefficiency of the non-degenerate factors detection, we state results separately for the two cases:

$$\varrho \leq \min\left\{\frac{\beta_{\min}^2}{1200}, \frac{\alpha(1-\alpha)\eta_{min}^2}{240\kappa dHN}\right\} \qquad \text{(no non-degenerate factor)}$$

$$\varrho \leq \min\left\{\frac{\beta_{\min}^2}{1200}, \frac{\alpha(1-\alpha)\eta_{min}^2}{240\kappa dHN}, \frac{\alpha^2\eta_{min}^2\delta_{\text{abs}}}{30N^2|\mathcal{A}|^2}\log^{-1}\left(\frac{4}{\delta_{\text{abs}}}\right)\right\} \qquad \text{(general case)}$$

Using the definition of $\varrho$ from Theorem 7, we get a value of $n_{\text{abs}}$ for non-degenerate factor (Equation 40) and general case (Equation 41) given below:

$$n_{\text{abs}} \geq \frac{3840^2 N^4 |\mathcal{A}|^2}{\alpha^4 \eta_{min}^4 \sigma^2} \ln\left(\frac{|\mathcal{G}|}{\delta_{\text{abs}}}\right) \max\left\{\frac{\kappa^2 d^2 H^2 N^2}{\alpha^2 (1-\alpha)^2 \eta_{min}^4}, \frac{25}{\beta_{\min}^4}\right\} \tag{40}$$

$$n_{\text{abs}} \geq \frac{3840^2 N^4 |\mathcal{A}|^2}{\alpha^4 \eta_{min}^4 \sigma^2} \ln\left(\frac{|\mathcal{G}|}{\delta_{\text{abs}}}\right) \max\left\{\frac{\kappa^2 d^2 H^2 N^2}{\alpha^2 (1-\alpha)^2 \eta_{min}^4}, \frac{25}{\beta_{\min}^4}, \frac{N^4 |\mathcal{A}|^4}{\alpha^4 \eta_{min}^4 \delta_{\text{abs}}^2} \ln^2\left(\frac{4}{\delta_{\text{abs}}}\right)\right\} \tag{41}$$

Recall that for detecting degenerate factors we collect $n_{\text{deg}}$ samples. Corollary 9 gives value of this hyperparameter as

$$n_{\text{deg}} = \frac{3 N^2 |\mathcal{A}|^2}{\alpha^2 \eta_{min}^2} \log\left(\frac{4}{\delta_{\text{abs}}}\right),$$

which also satisfies the condition in Lemma 16. Lastly, Theorem 3 gives number of samples for independence testing $n_{\text{ind}}$ and rejection sampling frequency $k$ as:

$$n_{\text{ind}} \geq \mathcal{O}\left(\frac{1}{\beta_{\min}^4} \ln \frac{m^2 |\mathcal{A}||\mathcal{F}| (2ed)^{2\kappa+1}}{\delta_{\text{ind}}}\right), \qquad k \geq \frac{8}{\eta_{min}} \ln\left(\frac{30}{\beta_{\min}}\right)$$

Failure probabilities for a single timestep are bounded by $\delta_{\text{ind}}$ due to identification of emission structure (Theorem 3), $3d\delta_{\text{abs}}$ due to decoding (Corollary 9), and $\delta_{\text{est}}$ due to model estimation (Lemma 17). The total failure probability using union bound for a single step is given by $\delta_{\text{ind}} + 3d\delta_{\text{abs}} + \delta_{\text{est}}$, and for the whole algorithm is given by $\delta_{\text{ind}} H + 3d\delta_{\text{abs}} H + \delta_{\text{est}} H$. Binding $\delta_{\text{ind}} H \mapsto \delta/3$, $3d\delta_{\text{abs}} H \mapsto \delta/3$, $\delta_{\text{est}} H \mapsto \delta/3$, gives us total failure probability of $\delta$ and the right value of hyperparameters.

Sample complexity of `FactoRL` is at most $k H n_{\text{ind}} + H n_{\text{abs}} + H n_{\text{deg}} + H n_{\text{est}}$ episodes which is order of:

$$\texttt{poly}\left\{d^{16\kappa}, |\mathcal{A}|, H, \frac{1}{\eta_{min}}, \frac{1}{\delta}, \frac{1}{\beta_{\min}}, \frac{1}{\sigma}, \ln m, \ln|\mathcal{F}|, \ln|\mathcal{G}|\right\},$$

where use the fact that $N = |\Psi_{h-1}|$ can be at most $2(ed)^{2\kappa}$ from Lemma 23. Note that if we did not have to apply the expensive degeneracy detection step, then we would get logarithmic dependence on $1/\delta_{\text{abs}}$. Cheaper ways of detecting degeneracy can, therefore, significantly improve the sample complexity.

We have not attempted to optimize the degree and exponent in the sample complexity above.

For our choice of two hyperparameters $n_{\text{est}}$ and $n_{\text{abs}}$, we can bound the model error and decoding failure by:

(Model Error) $\quad 6d(\Delta_{\text{mod}} + \Delta_{\text{app}}) \leq \frac{(1-\alpha)\eta_{min}}{4H}$, $\quad$ (Decoding Failure) $\quad \varrho \leq \frac{\alpha(1-\alpha)\eta_{min}^2}{240\kappa d H N}$.

**Verifying Induction Hypothesis.** Finally, we verify the different induction hypothesis below.

1. We already verified IH.1 with Theorem 3. We learn a $\widehat{ch}_h$ that is equivalent to $ch_h$ upto label permutation.

2. We already verified IH.2 with Corollary 9. Given a real state $s \in \mathcal{S}_h$, our decoder outputs the corresponding learned state with high probability. We also derived the form of $\varrho$.

3. We already verified IH.3 with Theorem 13. We also derived the form of $\Delta_{\text{est}}$ and $\Delta_{\text{app}}$.

4. Lastly, Theorem 14 and our subsequent calculations for hyperparameter show that $\Psi_h$ is an $\alpha$-policy cover of $\mathcal{S}_h$ and that the size of $\Psi_h$ is at most $2(ed)^{2\kappa}$ from Lemma 23. Lastly, for all reachable factor values we get the value of learned policy as at least $(1+\alpha)\eta_{min}/2$ using Equation 32 and our choice of hyperparameter values. Similarly, from Equation 33 we get the value of learned policy for all unreachable factor values as at most $(1-\alpha)\eta_{min}/4$. This allows us to filter all unreachable factor values. In the main paper, we focus on the value of $\alpha = 1/2$, which explains why on Algorithm 1, line 8 we only keep those policies with value at least $(1+\alpha)\eta_{min}/2 = 3\eta_{min}/4$. This verifies IH.4.

This completes the analysis for `FactoRL`.

# E  SUPPORTING RESULT

**Lemma 23** (Assignment Counting Lemma). *For a given $k, d \in \mathbb{N}$ and $k \leq d$, the cardinality of the set $\{(\mathcal{K}, u) \mid \mathcal{K} \in \mathscr{C}_{\leq k}([d]), u \in \{0,1\}^{|\mathcal{K}|}\}$ is bounded by $2(ed)^k$.*

*Proof.* Assume $k \geq 2$. The cardinality of this set is given by $\sum_{i=0}^{k} \binom{d}{i} 2^i$ which can be bounded as shown below:

$$\sum_{i=0}^{k} \binom{d}{i} 2^i = 1 + 2d + \sum_{i=2}^{k} \binom{d}{i} 2^i \leq 1 + 2d + \sum_{i=2}^{k} \left(\frac{ed}{i}\right)^i 2^i \leq 1 + 2d + \sum_{i=2}^{k} (ed)^i < \sum_{i=0}^{k} (ed)^i.$$

The first inequality here uses the well-known bound for binomial coefficients $\binom{n}{i} \leq \left(\frac{ed}{i}\right)^i$ for any $n, i \in \mathbb{N}$ and $i \leq n$. Further bounding the above result using $ed - 1 \geq {ed}/{2}$ gives us:

$$\sum_{i=0}^{k} (ed)^i \leq \frac{(ed)^{k+1}}{ed - 1} \leq 2(ed)^k.$$

The proof is completed by checking that inequality holds for $k < 2$. $\qquad\square$

**Lemma 24** (Lemma H.1 in Du et al. (2019)). *Let $u, v \in \mathbb{R}_+^d$ with $\|u\|_1 = \|v\|_1 = 1$ and $\|u-v\|_1 \geq \varepsilon$. Then for any $\alpha > 0$ we have $\|\alpha u - u\|_1 \geq \frac{\varepsilon}{2}$.*

**Lemma 25** (Chernoff Bound). *Let $q$ be the probability of an event occurring. Then given $n$ iid samples with $n \geq \frac{1}{q}$, the probability that the event occurred at least once is at least $1 - 2\exp(\frac{-qn}{3})$.*

*Proof.* Let $X_i$ be a 0-1 indicator denoting if the event occurred or not, and let $X = \sum_{i=1}^{n} X_i$. We have $\mathbb{E}[X_i] = q$ and $\mathbb{E}[X] = qn$. Let $t = 1 - 1/qn$. We will assume that $qn > 1$ and so $t \in (0, 1)$.

Then the probability that the event never occurs is bounded by:

$$\mathbb{P}(X < 1) = \mathbb{P}(X < (1-t)qn) \leq \exp\left(\frac{-qnt^2}{3}\right) < 2\exp\left\{-\frac{qn}{3}\right\}.$$

$\qquad\square$

**Lemma 26** (Hoeffding's Inequality). *Let $X_1, X_2, \cdots, X_n$ be independent random variables bounded by the interval $[0, 1]$. Let empirical mean of these random variables be $\overline{X} = \frac{1}{n} \sum_{i=1}^{n} X_n$, then for any $t > 0$ we have:*

$$\mathbb{P}(\left|\overline{X} - \mathbb{E}[\overline{X}]\right| \geq t) \leq 2\exp(-2nt^2).$$

**Lemma 27** (Theorem 2.1 of Weissman et al. (2003)). *Let $P$ be a probability distribution over a discrete set of size $a$. Let $X^n = X_1, X_2, \cdots, X_n$ be independent identical distributed random variables distributed according to $P$. Let $\hat{P}_{X^n}$ be the empirical probability distribution estimated from sample set $X^n$. Then for all $\epsilon > 0$:*

$$\mathbb{P}(\|P - P_{X^n}\|_1 \geq \epsilon) \leq (2^a - 2)\exp\left(-\frac{n\epsilon^2}{8}\right).$$

The next result is a direct corollary of Lemma 27.

**Corollary 15.** *For any $m \geq \frac{8a}{\epsilon^2} \ln(1/\delta)$ samples, we have $\|P - P_{X^n}\|_1 < \epsilon$ with probability at least $1 - \delta$.*

**Lemma 28.** *Let $X_1, X_2, \cdots, X_n$ be 0-1 independent identically distributed random variables with mean $\mu$. Let $X = \sum_{i=1}^{n} X_i$. Fix $m \in \mathbb{N}$ and $\delta \in (0, 1)$. If $n \geq \frac{2m}{\mu} \ln\left(\frac{e}{\delta}\right)$ then $\mathbb{P}(X < m) \leq \delta$.*

*Proof.* This is a standard Chernoff bound argument. We have $\mathbb{E}[X] = n\mu$. Assuming $n \geq m/\mu$ then from multiplicative Chernoff bound we have:

$$\mathbb{P}(X < m) = \mathbb{P}\left(X \leq \left\{1 - \left\{1 - \frac{m}{n\mu}\right\}\right\} n\mu\right) \leq \exp\left(\frac{-n\mu}{2}\left\{1 - \frac{m}{n\mu}\right\}^2\right) \leq \exp\left(m - \frac{n\mu}{2}\right)$$

Setting right hand side equal to $\delta$ and solving gives us $n \geq \frac{2}{\mu}\left(m + \ln(\frac{1}{\delta})\right)$ which is satisfied whenever $n \geq \frac{2m}{\mu} \ln\left(\frac{e}{\delta}\right)$. $\qquad\square$

**Lemma 29** (Lemma H.3 of Du et al. (2019)). *For any $a, b, c, d > 0$ with $a \leq b$ and $c \leq d$ we have:*

$$\left| \frac{a}{b} - \frac{c}{d} \right| \leq \frac{|a - c| + |b - d|}{\max\{b, d\}}$$

The next Lemma is borrowed from Du et al. (2019). They state their Lemma for a specific event ($\mathcal{E} = \alpha(\hat{s})$ in their notation) but this choice of event is not important and their proof holds for any event.

**Lemma 30** (Lemma G.5 of Du et al. (2019)). *Let $\mathcal{S} = (\mathcal{S}_1, \cdots, \mathcal{S}_H)$ and $\widehat{\mathcal{S}} = (\widehat{\mathcal{S}}_1, \cdots, \widehat{\mathcal{S}}_H)$ be the real and learned state space. We assume access to a decoder $\widehat{\phi} : \mathcal{X} \to \widehat{\mathcal{S}}$ and let $\phi^\star : \mathcal{X} \to \mathcal{S}$ be the oracle decoder. Let $\theta_h : \widehat{\mathcal{S}}_h \to \mathcal{S}_h$ be a bijection for every $h \in [H]$ and $\theta : \widehat{\mathcal{S}} \to \mathcal{S}$ where $\theta(\hat{s}) = \theta_h(\hat{s})$ for $\hat{s} \in \widehat{\mathcal{S}}_h$. For any $h \in [H]$ and $s_h \in \mathcal{S}_h$, we assume $\mathbb{P}(\hat{s}_h = \theta_h^{-1}(s_h) \mid s_h) \geq 1 - \varepsilon$, i.e., given the real state $s_h$, our decoder $(\widehat{\phi})$ will map it to $\theta_h^{-1}(s_h)$ with probability at least $1 - \varepsilon$.*

*Let $\varphi : \mathcal{S} \to \mathcal{A}$ be a deterministic policy on the real state space and $\hat{\varphi} : \widehat{\mathcal{S}} \to \mathcal{A}$ be the induced policy given by $\hat{\varphi}(\hat{s}) = \varphi(\theta(\hat{s}))$ for every $\hat{s} \in \widehat{\mathcal{S}}$. Let $\pi, \hat{\pi} : \mathcal{X} \to \mathcal{A}$ where $\pi(x) = \varphi(\phi^\star(x))$ and $\hat{\pi}(x) = \hat{\varphi}(\widehat{\phi}(x))$ for every $x \in \mathcal{X}$. For every random event $\mathcal{E}$ we have:*

$$|\mathbb{P}_\pi(\mathcal{E}) - \mathbb{P}_{\hat{\pi}}(\mathcal{E})| \leq 2\varepsilon H$$

**Lemma 31** (Lemma H.2. of Du et al. (2019)). *Let there be two tabular MDPs $\mathcal{M}$ and $\widehat{\mathcal{M}}$. Let $\mathcal{S} = (\mathcal{S}_1, \cdots, \mathcal{S}_H)$ be the state space of $\mathcal{M}$ and $\widehat{\mathcal{S}} = (\widehat{\mathcal{S}}_1, \cdots, \widehat{\mathcal{S}}_H)$ be the state space of $\widehat{\mathcal{M}}$. Both MDPs have a $\mathcal{A}$ be the action space of both MDPs and horizon of $H$. For every $h \in [H]$, there exists a bijection $\theta_h : \widehat{\mathcal{S}}_h \to \mathcal{S}_h$. Let $T : \mathcal{S} \times \mathcal{A} \to \Delta(\mathcal{S})$ and $\widehat{T} : \widehat{\mathcal{S}} \times \mathcal{A} \to \Delta(\widehat{\mathcal{S}})$ be transition dynamics for $\mathcal{M}$ and $\widehat{\mathcal{M}}$ satisfying:*

$$\forall h \in [H], a \in \mathcal{A}, \hat{s} \in \widehat{\mathcal{S}}, \qquad \sum_{\hat{s} \sim \widehat{\mathcal{S}}_h} \left| T_h(\theta(\hat{s}') \mid \theta(\hat{s}), a) - \widehat{T}_h(\hat{s}' \mid \hat{s}, a) \right| \leq \varepsilon$$

*Let $\varphi : \mathcal{S} \to \mathcal{A}$ be a policy for $\mathcal{M}$. Let $\widehat{\varphi} : \widehat{\mathcal{S}} \to \mathcal{A}$ be the induced policy on $\widehat{\mathcal{M}}$ such that for any $\hat{s} \in \widehat{\mathcal{S}}$ we have $\widehat{\varphi}(\hat{s}) = \varphi(\theta(\hat{s}))$. Then for any $h \in [H]$ we have:*

$$\sum_{s_h \in \mathcal{S}_h} \left| \widehat{\mathbb{P}}_{\hat{\varphi}}(\theta^{-1}(\hat{s}_h)) - \mathbb{P}_\varphi(s_h) \right| \leq h\varepsilon$$

# F   EXPERIMENT DETAILS

We provide details for our proof of concept experiment below.

**Modeling Details.**   We model $\mathcal{F}$, used for performing independence test, using a single-layer feed-forward network $\theta_\mathcal{F}$ with Leaky ReLu non-linearity (Maas et al., 2013) and a softmax output layer. Give a pair of atoms $x[u]$ and $x[v]$, we concatenate these atoms and map it to a probability distribution over $\{0, 1\}$ by applying $\theta_\mathcal{F}$.

We implement the model class $\mathcal{G}$ for learning state decoder following suggestion of Misra et al. (2020). Recall that a function in $\mathcal{G}$ maps a transition $(x, a, \check{x}) \in \mathcal{X} \times \mathcal{A} \times \mathcal{X}^\star$ to a value in $[0, 1]$. We first map $x$ and $\check{x}$ to vectors $v_1$ and $v_2$ respectively, using two separate linear layers. We map the action $a$ to its one-hot vector representation $\mathbf{1}_a$. We map the vector $v_2$ to a probability distribution using the Gumbel-softmax trick (Jang et al., 2016), by computing $q_i \propto \exp(v_2[i] + \vartheta_i)$ for all $i \in \{1, 2\}$, where $\vartheta_i$ is sampled independently from the Gumbel distribution. We concatenate the vectors $v_1, \mathbf{1}_a$ and $q$ and map it to a probability distribution over $\{0, 1\}$, through a single layer feed-forward network $\theta_\mathcal{G}$ with Leaky ReLu non-linearity. We recover a decoder $\phi$ from the model that maps a set of atoms $\check{x}$ to $\phi(\check{x}) = \arg\max_{i \in \{0, 1\}} q_{i+1}$.

**Learning Details.** We train the two models using cross-entropy loss. Formally, given a dataset $\mathcal{D}_{ind} = \{(x_i[u], x_i[v], y_i)\}_{i=1}^{n_{ind}}$ for performing independence testing, and a dataset $\mathcal{D}_{abs} = \{(x_i, a_i, \check{x}_i, y_i)\}_{i=1}^{n_{abs}}$ for learning a decoder, we optimize the model by minimizing the cross-entropy loss as shown below:

$$\hat{f} = \arg\max_{f \in \mathcal{F}} \frac{1}{n_{ind}} \sum_{i=1}^{n_{ind}} \ln f(y \mid x_i[u], x_i[v]), \qquad \hat{g} = \arg\max_{g \in \mathcal{G}} \frac{1}{n_{abs}} \sum_{i=1}^{n_{abs}} \ln g(y \mid x_i, a_i, \check{x}_i).$$

Here we overload our notation to allow the output of models to be distribution over $\{0, 1\}$ rather than a scalar value in $[0, 1]$ as we assumed before. This is in sync with how we implement these model class and allows us to conveniently perform cross-entropy loss minimization.

**Planner Details.** We use a simple planner based on approximate dynamic programming. Given model estimate, reward function and a set of visited learned states, we perform dynamic programming to compute optimal Q-values for the set of visited states. We assume the Q-values for non-visited states to be 0. This allows us to compute Q-values in a computationally-efficient manner. Later, if the agent visits an unvisited state, then it simply takes random action.

**Hyperparameters.** We set the hidden dimension of $\theta_{\mathcal{F}}$ to 10 and that of $\theta_{\mathcal{G}}$ to 56. We set the threshold $c$ on held-out log-loss value, when performing independence test to be $0.65$. For reference, if one uses a random uniform classifier then its performance is $-\ln(0.5) \approx 0.69$. We train both models for 10 epochs using Adam optimization with learning rate of $0.001$, and a batch size of 32. We remove $0.2\%$ of the training data and use it as a validation set. We evaluate on the validation set after every epoch, and use the model with the best performance on the validation set. We used PyTorch 1.6 to develop the code and used default initialization scheme for all parameters.

