# OpenReview forum: "Provable Rich Observation Reinforcement Learning with Combinatorial Latent States"
_ICLR.cc/2021/Conference — ICLR 2021 Poster_

### Official Review · AnonReviewer3 · 2020-10-26
**Questions about the dynamics assumptions and hyperparameters.**

**Rating:** 7
**Confidence:** 4

**Review:**

This paper proposes a new result for provably efficient exploration in rich observation RL with assuming that the underlying structure is a factored block MDP. Under this setting along with some other dynamics assumptions, e.g., reachability and identifiability, the algorithm achieves a sample complexity independent of the number of latent states |S| but the number of factors d (which can be much smaller than |S|). Although the algorithm uses a similar approach as in Du et al. 2019, the adaptation is not that straightforward. I appreciate the novelty of this result in the direction of combining factored MDP with rich observation RL.

My major concerns are:
1. Since this work takes a similar approach as in Du et al. 2019, i.e., learn decoding functions and transition model level-by-level, it also requires additional dynamics assumptions e.g., reachability and identifiability which are described by hyperparameters such as \eta_{min} and \beta_{min}. And the sample complexity can still be pretty large if \eta_{min} and \beta_{min} are very small even with a small d.
2. In one recent work by Feng et al. (https://arxiv.org/abs/2003.06898), a general framework is proposed to attack the rich observation RL problem under the block MDP setting without any dynamics assumption. It might be good to include a mention of this work and some discussions about whether the approach in Feng et al. can be extended to the factored block MDP setting (if not, what is the main challenge).
3. For the hyperparameters such as \eta_{min}, \beta_{min}, \kappa, and d, how should a practitioner select these values without any prior knowledge? Can this be done in an adaptive manner as we run the algorithm? Or must we choose very conservative values (e.g., very small \eta, \beta, and big \kappa) initially to guarantee the learning accuracy?

Minors:
- Some notations are a little bit confusing for first-time readers. More clarification can be added, e.g., 1) A parent function is a set-valued map from an integer in [d] to a subset of [d]; 2) s[pt(i)] := {s[j] | j \in pt[i]}. A simple example would be even better for readers to quickly understand.
- I appreciate this work as a theoretical result. It would be more promising if some preliminary experiments can be conducted to compare with prior works, e.g. Du et al. 2019 and Feng et al. 2020, to further demonstrate the benefit of a factored MDP setting in practice.

I would like to change my score if the authors can address my concerns.

---

> ### Author Response · Authors · 2020-11-19
> **We will revise the paper to include comparison with Feng et al., and will add experiments and hyperparameter tuning details.**
>
> Thank you for your feedback. Please see the response below:
>
> > _Since this work takes a similar approach as in Du et al. 2019,...it also requires additional dynamics assumptions e.g., reachability and identifiability which are described by hyperparameters such as_ $\eta_{min}$ _and_ $\beta_{min}$. _And the sample complexity can still be pretty large if_ $\eta_{min}$ _and_ $\beta_{min}$ _are very small even with a small d._
>
> **Response:** We want to clarify that Du et al., 2019 make similar assumptions as our reachability assumption ($\eta_{min}$) and the identifiability Assumption 1. However, identifiability Assumption 2 is specific to our algorithm since we deal with the extra challenge of factored MDP.
>
> We have not tried to optimize the polynomial degree in our sample complexity, and feel that dependence on $\eta_{min}$ and $\beta_{min}$ can be improved further. Specifically, for $\eta_{min}$, we think our algorithm may efficiently learn an $\epsilon$-optimal policy even if $\eta_{min}$ was very small. E.g., if one added a factor that can be reached with an extremely small probability $p$ ($p \lt\lt \epsilon$), then we feel our algorithm will simply ignore this factor and still work without any performance issue. Removing any dependence on $\beta_{min}$ appears harder. Since the primary goal of this paper is to initiate the first line of work that provably learns factored MDP with rich observations, we will leave this problem for future work.
>
> $~$
> ### Comparison with Feng et al., 2020
>
> **Response:** Thank you for the reference to Feng et al., 2020. We will cite and discuss this paper in our revision. We think our work and Feng et al., 2020 are very different in terms of the problem setting, thus not directly comparable. See detailed comparison as following:
>
> 1. In Feng et al. 2020, they assume having access to a purely unsupervised learning oracle, and by feeding it with raw observations, the oracle can output a decoder that can decode the latent state accurately enough. Such an assumption may be too good to be true in practice. Moreover, by making this strong assumption, they greatly reduce the difficulty of learning the decoder, which is one of the main challenges we are addressing in this paper.
>
> 2. Feng et al., 2020 study non-factorized block MDPs and the sample complexity of their result has polynomial dependence on the size of latent space. In our paper, we assume combinatorial latent space, the size of which can be exponential in $d$ (the number of latent factors). And importantly, the sample complexity of our algorithm depends only polynomially on $d$. If one wants to extend the approach in Feng et al., 2020 to the factored block MDP setting, one of the main challenges is to improve the dependence of their result on the size of latent space. It is unclear to us how to directly extend their algorithm to accomplish this.
>
> $~$
> ### Setting Hyperparameters
>
> **Response:** Great question! We can use any values of $\eta_{min}$, and $\beta_{min}$ that are lower bound of the actual quantity. One way to tune them is to use a halving trick: we start with $\eta_{min} = 1$ and $\beta_{min}=1$, and then compute the final metric we care about (e.g., the value of the learned policy). We then half the chosen values, and try again. If the metrics improve then we keep halving until they stop improving. This is better than starting with an ultra-conservative value which can be very inefficient. Hyperparameters “d” and “\kappa” can be similarly tuned, except instead of halving the values we will double them.
>
> $~$
> > _Some notations are a little bit confusing for first-time readers. More clarification can be added, e.g., 1) A parent function is a set-valued map from an integer in [d] to a subset of [d]; 2) s[pt(i)] := {s[j] | j \in pt[i]}. A simple example would be even better for readers to quickly understand._
>
> **Response:** Thank you for the great suggestion. We described the first in Definition 1 (line: $pt : [d] → 2^{[d]}$) and the second in the footnote number 1. We will use the extra page to incorporate your suggestion by describing these in detail in the main text with an example.
>
> $~$
> ### Proof-of-concept-experiments
>
> **Response:** We believe that our paper represents an important theoretical contribution and stands on its own merits even without experiments. Nonetheless, we have evaluated FactoRL using a proof-of-experiment and will revise the paper to include the details. Our experiments show that for every time step, FactoRL accurately learns (i) the latent child function, (ii) latent state decoder, (iii) latent transition model, and (iv) a policy cover for exploration. We will release the code along with the paper.

---

### Official Review · AnonReviewer2 · 2020-10-26
**A sound paper with limited novelty and unclear contribution**

**Rating:** 5
**Confidence:** 3

**Review:**

##########################################################################
Summary:
This paper studies reinforcement learning under the setting of factored block MDP, where the observations are generated from latent factors. The paper presents a framework with separate parts for learning the policy, the emission structure, and the model, respectively. The performance of the proposed framework is theoretically analyzed.

##########################################################################
I find the paper theoretically sound but lean towards rejection at this point.
##########################################################################
Major comments:
It is appreciated that the authors provide the complete theoretical analysis of the proposed framework, and the analysis is reasonable and sound to me.

My main concern is that the paper combines several settings together without much technical novelty. The settings are somehow artificial in the sense that a lot of assumptions are made only for obtaining theoretical results, and in addition, no experimental study was provided.

Detailed comments:

“Our goal here is to define a problem setting that …” This sounds to me that the paper aims to find a setting that can be analyzed with proofs but not the settings that are natural and have real applications. In addition, it would be good to inform the reader which settings are inapplicability or intractability.

The discussion on the rationale of using factored transition is not necessary (as it is well-known as the basic part of graphical models).

The paper adopts many assumptions, but not all of them are well-justified. E.g.,
-	“For our purposes, we assume a deterministic start state and assume without loss generality that each state is reachable at exactly one time step” It is not clear to me what the "purposes" are. What is the consequence if the start state is not deterministic? Why do we assume that each state is reachable at exactly one time step?
-	In addition to making the problem trackable, do we have the disjointness property and Assumption 1& 2 in real applications (such as the image example in the paper)?
-	The paper assumes two computational oracles and one planning oracle. Please provide the accessibility of these oracles so one can evaluate the feasibility of the proposed method.



The presentation is often flowery without being clear. E.g.,
-	“There are many possible ways to add rich observations to a factored MDP resulting in inapplicability or intractability.” Please clarify with respect to what the inapplicability or intractability is taken (e.g., learnability or computability).
-	Please be consistent. For example, “the observations are emitted by latent states” and “the observations are emitted by latent factors”; “decode the latent state” and “decode the factors”. They are making the paper hard to follow.
-	In Def 2, ch() is defined to be [d]->2^[m], making x[chi(i)] not well-defined.
-	“An agent is responsible for mapping each observation x ∈ X to individual atom …” What is the difficulty in having such a mapping? From the paper, the atoms are simply given by the observations.

There are almost 30 pages proof in the appendix. Are all the proofs new? If not, please state clearly that which proofs can be found in the existing works and which can be obtained by adapting the proofs in other works.  Otherwise, it is hard to evaluate the contribution of this paper.

---

> ### Author Response · Authors · 2020-11-19
> **Part 1 of Response: Assumption, Example of Intractable Setting, Language Consistency, and Experiments**
>
> Thank you for the feedback. We'll respond to your queries in a couple of comments.
>
> ## Assumptions
>
> All assumptions except Assumption 2 are adapted from prior work. We will use the extra page to discuss these, and briefly do so below:
>
> 1. **Each state is reachable at exactly a one-time step?:**
> This holds true without loss of generality, as the algorithm can simply concatenate the time step information to observation/state.  We do so for algorithmic convenience. As we are learning the model iteratively over time step, this allows us to focus on a time step without worrying about any possible conflict with other time steps.
>
> 2. **Disjointness Assumption:** This assumption was introduced in Du et al., 2019, who argued that for many real-world problems, observations are rich enough, to enable inferring the latent state. Navigation with an overhead camera, such as the example in Figure 1, is the simplest example where this is true. However, many first-person view problems can also satisfy it. E.g., no two different locations in New York street will generate the same image, and the difference can be captured using a good camera.
>
> 3. **Deterministic start state and Assumption 1**: These assumptions were also introduced in Du et al., 2019. Even though our algorithm is significantly different from theirs, we need these assumptions for decoding the state.  Assumption 1 is true for many real-world problems.  Specifically, it is true for all problems with deterministic transitions, which is common in the empirical world. Generally, if one were to create a transition probability matrix $T(s’ [i]  | s[pt(i)], a)$ by uniformly sampling from the space of all transition matrices, then the probability that it gives a 0 margin is 0. As we decode state at a given time step using decoder for previous time steps, we assume a deterministic start state for the base case of our induction to hold. For some problems, these assumptions may be undesirable. Since the primary goal of this paper is to initiate the first line of work that provably learns factored MDP with rich observations, we leave removing this assumption for future work.
>
> 4. **Computational and Planning Oracle:**  Our computational oracle for $\mathcal{F}$ is a standard square loss minimizer which is routinely used. If the model family is convex, then one can provably optimize them with gradient descent. The square loss minimizer for $\mathcal{G}$ is borrowed from Misra et al., 2020 who discuss how to implement it.  Planning oracle is commonly assumed in the Factored MDP literature that focus on statistical guarantees (e.g., Strehl et al., (2007) and Osband and van Roy (2014)). In parallel, there is significant work on developing planning oracles using various approximation strategies, (e.g., linear programming). For reference, see:
>
>   - Efficient solution algorithms for factored mdps, Guestrin et al. 2013
>   - Multiagent Planning with Factored MDPs, Guestrin, et al., 2001
>   - Distributed Planning in Hierarchical Factored MDPs, Guestrin and Gordon 2012
>
> 5. **The only assumption we introduce is Assumption 2.** This assumption says that atoms emitted from the same factor are correlated, which is true for some real-world problems. For example, consider navigation in a grid-world. Each cell in the grid represents a factor ($s[i]$), and can be occupied ($s[i]=1$) or empty ($s[i]=0$). If the cell is occupied then an object sampled from some distribution takes its place (e.g., fridge, tv). Assumption 2 is true in this case. If the object occupying the cell at the moment is a television, then one pixel from the cell dictates what the other pixels would look like.
>
> $~$
> ## Example of an intractable setting
>
> Say we take a factored MDP model and add an emission process $q(x\mid s)$ to generate an observation $x$ from a state $s$. Without the disjointness property, the problem becomes partially-observed and exponential sample complexity lower bounds for POMDPs come into play. Now say we have disjointness but allow $q$ to be arbitrary, then also we cannot learn an accurate decoder with polynomial samples. Since the emission process for one state does not dictate behaviour for another state, and there are exponentially many states.
>
> $~$
> ## Language Consistency
>
> We use the term factor to refers to individual dimensions ($s[i]$) of a state $s$. It is possible to both decode a factor or a state. We’ll carefully review to ensure that we use the right term at the right place.
>
> $~$
> ## Experiments
>
> We believe that our paper represents an important theoretical contribution and stands on its own merits even without experiments. Nonetheless, we have evaluated FactoRL using a proof-of-experiments, and will revise the paper to include the details. Our experiments show that for every time step, FactoRL accurately learns (i) the latent child function, (ii) latent state decoder, (iii) latent transition model, and (iv) a policy cover for exploration. We will release code with the paper.

---

> > ### Comment · AnonReviewer2 · 2020-11-24
> > **Novelty**
> >
> > The author's response is appreciated. I am happy to raise my score to 5 but still have some concerns.
> >
> > ---------------------------------------
> >
> > The paper justifies its novelty by the fact that the considered setting has not been studied in existing works, which is, however, not sufficient.
> >
> > The first point remaining unclear is the importance of the proposed setting. To me, it is a little bit artificial, in the sense that it relies on many assumptions and there is no real-world application. The strength of the paper would significantly increase if we could identify one application satisfying all the assumptions and experiment with it.
> >
> > Second, the paper combines the analysis of several parts together, and mentions that the key challenge is making sure that all arguments work together in tandem and there is no cascading error that leads to intractability. I agree that long proofs are hard to handle, but the length of the proof, by itself, does not justify its technical novelty. The three modules are, respectively, based on Sen et al. (2017), Misra et al., 2020 , and Du et al., 2019. It is true that none of them has addressed the three modules in one paper, but this does not mean that combining them together is a novel contribution.
> >
> > Finally, I wish to note that we can always formulate a very complicated problem by combining a sequence of learning problems, and get certain theoretical results under certain assumptions. But it is not very interesting if the assumptions are added only for getting theoretical results.

---

> > > ### Author Response · Authors · 2020-11-24
> > > **Quick clarification:  Our first module is in Appendix D.1 not Appendix C, and is not based on Sen et al., 2017.**
> > >
> > > Thank you for your comments.
> > >
> > > We want to quickly clarify one thing about Sen et al., 2017. That paper is about how to do oracalized independence test. It has nothing to do with reinforcement learning.  The independence test analysis in Appendix C is similar to Sen et al., *but Appendix D.1, which is the first module of our algorithm, is totally new and fully designed for our task.* This is the most novel part of the paper which shows, how we convert the children function identification problem to performing a set of independence tests using carefully constructed distributions. We are able to create these distributions through the combined use of policy cover, previous decoder, and rejection sampling.
> > >
> > > Please let us know if more clarification is needed. The step on identifying children function (and proof in Appendix D.1) is both the main novelty and a central idea in our paper. We are happy to explain more.

---

> ### Author Response · Authors · 2020-11-19
> **Part 2 of Response: Technical Novelty, Proof Novelty, Notation, and Observation Mapping**
>
> This is a continuation of response from Part 1.
>
> ## Technical Novelty
>
> We want to clarify that our work is the first provably sample efficient algorithm that can solve rich-observation factored MDP and be efficiently implemented in an oracle model. Previous work such as Du et al., 2019 and Misra et al., 2020 cannot get a sample complexity guarantee which is polynomial in the number of state factors.  As both factorization and rich-observation are common in real-world problems, we believe our work makes major inroads into the theoretical study of this important novel setting. To the best of our knowledge, the main claim of the paper (Theorem 1) is new.
>
> We further argue that our approach is non-trivial, and not a simple extension of a previous algorithm. We explain this below:
>
> 1) FactoRL performs a novel step in identifying the latent emission structure through the use of special roll-in distributions that make atoms from different factors to be independent. This enables us to reduce the problem to independent tests that we solve in an oracle model using noise-contrastive learning. Neither Du et al., 2019 nor Misra et al., 2020 (the two closest work) have an analogous step since they do not address factorization. In general, disentangling latent factors is theoretically challenging and empirically appealing.
>
> 2) Even if the latent emission structure was somehow known, the approach of Du et al., 2019 and Misra et al., 2020 cannot decode the state or learn a model. Further, it is unlikely they can be separately adapted to work for the factored setting. The prediction problem in Du et al. 2019 is unsuitable for this task (++). And, as optimal policies for factored MDP do not factorize, therefore, the model-free planning in Misra et al. 2020 is unlikely to succeed (see [Sun et al. 2019](https://arxiv.org/abs/1811.08540) for discussion). Lastly, the prediction problem of Misra et al. 2020, is not tailored to handle factorization, i.e., using it will not give us a state which respects the latent transition factorization (**). We are happy to explain ++ and ** in more detail, if asked, but defer it for the moment for brevity.
>
> FactoRL combines the best ideas from these approaches, in a way that we feel is non-trivial. It adapts the prediction problem in Misra et al., 2020 to address factorization by learning a separate decoder for each factor. And it follows a model-based learning such as in Du et al., 2019, that avoids exponential sample complexity. Recall that this is on top of identifying latent emission structures that neither approaches do.
>
> ## Proof Novelty
>
> We have tried to keep all the results that are used exactly from other papers in Appendix E. The rest of the proof is tailored for FactoRL and contains results which are either totally new or modification of reasoning in previous work. We briefly describe this division below:
>
> Our algorithm consists of three key modules: (1) identifying the latent emission process; (2) learning the decoder function; (3) estimating the model and learning a policy cover. To our knowledge, the analysis for the first two modules is new. We comment that for the second module, we adapt a similar method proposed by Misra et al. (2020)  for estimating the decoder. However, our analysis differs greatly from and is never some easy modification of theirs because we need to carefully utilize the factorized structure otherwise we get an exponential sample complexity. In module (3), we follow the line of reasoning in Du et al., 2019 but the part on learning parent function and factorized models are new, as Du et al., 2019 don’t address factorization.
>
> We note that whole Appendix D is trying to prove Theorem 1 in the main paper which is novel and as we argued earlier non-trivial. A key challenge in such long proofs is making sure that all arguments work together in tandem and there is no cascading error that leads to intractability. E.g., since we cannot learn a perfect decoder,  the estimation error in module (2) can in turn influence identifying the latent emission process in module (1). Therefore, we need to carefully control the error amplification between modules.
>
> $~$
> > _In Def 2, ch() is defined to be [d]->2^[m], making x[chi(i)] not well-defined._
>
> We define this in footnote 1. Formally, for any ordered set $U$ of length $n$ and an ordered set $I \subseteq [n]$ and length $k$, we define $U[I] = (U[I[1]], U[I[2]], …, U[I[k]])$. We will move this definition to the main text, and as suggested by Reviewer 3, we will provide an example.
>
> $~$
>
> > “An agent is responsible for mapping each observation x ∈ X to individual atom …” What is the difficulty in having such a mapping?
>
> This appears doable in realistic settings.  For example, observation can be a string, and atoms can be individual words which can be obtained by using off the shelf tokenizer. In this case, observation is not already presented as an atom sequence. However, tokenization is routinely performed in practice.

---

### Official Review · AnonReviewer1 · 2020-10-28
**Potentially interesting paper; needs better presentation**

**Rating:** 6
**Confidence:** 2

**Review:**

This paper studies reinforcement learning in spaces with a large number of states by modeling the states using a factored / latent representation. This problem has been studied in the non-factored setting by Du et al (2019), and this paper extends to factored settings. This is an interesting problem.

Overall, I found the paper difficult to follow since it is not presented well. It would help to distill the novelty and main ideas in the algorithms early. Section 2 on the setting includes too many technical details that can be deferred to later sections.

This paper also does not verify the feasibility of their algorithms through experiments. Compare this to the cited papers which include experiments to show the validity and feasibility of their algorithms.

---

> ### Author Response · Authors · 2020-11-19
> **Experiments will be added. Request for more specific feedback on presentation.**
>
> Thank you for the feedback. Please see our response below:
>
>  >_“This paper also does not verify the feasibility of their algorithms through experiments. Compare this to the cited papers which include experiments to show the validity and feasibility of their algorithms.”_
>
> **Response:** We believe that our paper represents an important theoretical contribution and stands on its own merits even without experiments. Nonetheless, we have evaluated FactoRL using a proof-of-experiments and will revise the paper to include the details. Our experiments show that for every time step, FactoRL accurately learns (i) the latent child function, (ii) latent state decoder, (iii) latent transition model, and (iv) a policy cover for exploration. We will be releasing the code with the paper.
>
> $~$
> > _“Overall, I found the paper difficult to follow since it is not presented well. It would help to distil the novelty and main ideas in the algorithms early. Section 2 on the setting includes too many technical details that can be deferred to later sections.”_
>
> **Response:** We tried to optimize for providing all theoretical assumptions, notations, and definitions in the paper early on so that it is easy to understand the main claim (Theorem 1) in the paper. We will really benefit if you can provide specific feedback on which details could be moved to later sections. We will revise the paper to add details for the example in Figure 1, that hopefully, will provide a more accessible intuition about the main techniques used by the FactoRL algorithm.

---

### Official Review · AnonReviewer4 · 2020-11-01
**Factorized reinforcement learning**

**Rating:** 7
**Confidence:** 3

**Review:**

The paper considers the problem of partitioning the atoms (e.g., pixels of an image) of a reinforcement learning task to latent states (e.g., a grid that determines whether there exists furniture in each cell). The number of states grows exponentially with the number of cells of the grid. So the algorithms that are polynomial in the number of states are not efficient. The paper considers the factored block Markov decision process (MDP) model and adds a few more assumptions. Generally, this model and assumptions guarantee that the cells of the grid partition the atoms (i.e., each atom depends on only one cell), the atoms in a cell are dependent (in the probabilistic sense), the conditional probability of the parent value of the states and the action given the next state is 0 or 1 is separated (i.e., the difference is bounded away from zero), and the regressor classes that are used are realizable. The paper shows that this is enough to give an algorithm that partitioned the atoms in each step with high probability and its time complexity is polynomial in the number of cells and logarithmic in the number of atoms.

The model and assumptions are very natural in many applications, and the theoretical guarantee is very nice. Moreover, the paper is very well-written and clear. However, I would like to see some experiments regarding this model. Even an experiment on a synthesized dataset with the grid and furniture example would be very interesting.

---

> ### Author Response · Authors · 2020-11-19
> **We will add proof-of-concept experiments**
>
> Thank you for your positive comments! Please see our response below:
>
> >_However, I would like to see some experiments regarding this model. Even an experiment on a synthesized dataset with the grid and furniture example would be very interesting._
>
> **Response:** We believe that our paper represents an important theoretical contribution and stands on its own merits even without experiments. Nonetheless, we have evaluated FactoRL using a proof-of-experiment and will revise the paper to include the details. Our experiments show that for every time step, FactoRL accurately learns (i) the latent child function, (ii) latent state decoder, (iii) latent transition model, and (iv) a policy cover for exploration. We will be releasing the code with the paper.

---

### Author Response · Authors · 2020-11-24
**Submitted a Revision: Added proof of concept experiment, comparison with Feng et al., and examples**

We have uploaded a revision with the following changes (all edits except new citations are in blue color to make it easy to find):

1. Added a proof-of-concept experiment to support our theoretical result, as asked by multiple reviewers. We do want to echo our sentiment, that we feel our paper has an important theoretical contribution that stands on its own merits without experiments.

2. We have cited and added a comparison to Feng et al., as asked by AnonReviewer3. We have also discussed the Du et al., and Misra et al., approaches and highlighted their shortcomings. We have argued why none of them can solve our setting, and therefore, it shows that our approach is novel and non-trivial.

3. We have added short examples for different assumptions as asked by AnonReviewer2. We again want to emphasize, that all assumptions except one are adapted from previous work in RL theory.

4. We have moved details about notations from footnote into a short paragraph.

We would appreciate hearing from reviewers, either on our detailed response below or at the new revision.

---

### Decision · Program_Chairs · 2021-01-07
**Final Decision**

**Decision:**

Accept (Poster)

**Comment:**

This paper considers the reinforcement learning in rich observation setting. Concretely, the authors provide a provable sample efficient algorithm for the rich-observation factored MDP. As the majority of the reviewers commented, although the techniques used in the proof share some similarities to the existing work, the analysis for the whole algorithm is still challenging. As a theoretical oriented paper, I think this paper should have a position in ICLR.

The major concern of the paper is the necessity of the assumptions (R2). The validation and justification of the Assumptions should be stated clearly in main text, even they are adapted from the prior work.